# Effects of *Schistosoma haematobium* infection and treatment on the systemic and mucosal immune phenotype, gene expression and microbiome: A systematic review

**Anna M. Mertelsmann**[1,2]*, **Sheridan F. Bowers**[2], **Drew Wright**[3], **Jane K. Maganga**[4], **Humphrey D. Mazigo**[5], **Lishomwa C. Ndhlovu**[1], **John M. Changalucha**[4], **Jennifer A. Downs**[2,4,6]

**1** Division of Infectious Diseases, Department of Medicine, Weill Cornell Medicine, New York, New York, United States of America, **2** Center for Global Health, Weill Cornell Medicine, New York, New York, United States of America, **3** Samuel J. Wood Library & C.V. Starr Biomedical Information Center, Weill Cornell Medical College, New York, New York, United States of America, **4** Mwanza Intervention Trials Unit/National Institute for Medical Research, Mwanza, Tanzania, **5** Department of Parasitology and Entomology, Catholic University of Health and Allied Sciences, Mwanza, Tanzania, **6** Weill Bugando School of Medicine, Mwanza, Tanzania

* annamertelsmann@gmail.com

## Abstract

### Background

Urogenital schistosomiasis caused by *Schistosoma haematobium* affects approximately 110 million people globally, with the majority of cases in low- and middle-income countries. Schistosome infections have been shown to impact the host immune system, gene expression, and microbiome composition. Studies have demonstrated variations in pathology between schistosome subspecies. In the case of *S. haematobium*, infection has been associated with HIV acquisition and bladder cancer. However, the underlying pathophysiology has been understudied compared to other schistosome species. This systematic review comprehensively investigates and assimilates the effects of *S. haematobium* infection on systemic and local host mucosal immunity, cellular gene expression and microbiome.

### Methods

We conducted a systematic review assessing the reported effects of *S. haematobium* infections and anthelmintic treatment on the immune system, gene expression and microbiome in humans and animal models. This review followed PRISMA guidelines and was registered prospectively in PROSPERO (CRD42022372607). Randomized clinical trials, cohort, cross-sectional, case-control, experimental *ex vivo*, and animal studies were included. Two reviewers performed screening independently.

### Results

We screened 3,177 studies and included 94. *S. haematobium* was reported to lead to: (i) a mixed immune response with a predominant type 2 immune phenotype, increased T and B

**Data Availability Statement:** All extracted data used for this systematic review are available in the tables that accompany this manuscript.

**Funding:** This work was supported by grants for Research Training in Infectious Diseases (5T32 AI 007613; AMM) as well as K24 AI182638 (JAD) and R01 AI 168306 (JAD) from the National Institute of Allergy and Infectious Diseases, the Burroughs Wellcome Fund (BWF 1020043; AMM) and by the Fogarty International Center (D43 TW 011826; JKM). The funders had no role in study design, data collection and analysis, decision to publish, or preparation of the manuscript.

**Competing interests:** The authors have declared that no competing interests exist.

regulatory cells, and select pro-inflammatory cytokines; (ii) distinct molecular alterations that would compromise epithelial integrity, such as increased metalloproteinase expression, and promote immunological changes and cellular transformation, specifically upregulation of genes *p53* and *Bcl-2*; and (iii) microbiome dysbiosis in the urinary, intestinal, and genital tracts.

## Conclusion

*S. haematobium* induces distinct alterations in the host's immune system, molecular profile, and microbiome. This leads to a diverse range of inflammatory and anti-inflammatory responses and impaired integrity of the local mucosal epithelial barrier, elevating the risks of secondary infections. Further, *S. haematobium* promotes cellular transformation with onco-genic potential and disrupts the microbiome, further influencing the immune system and genetic makeup. Understanding the pathophysiology of these interactions can improve outcomes for the sequelae of this devastating parasitic infection.

## Author summary

The parasitic trematode *S. haematobium* affects 110 million people worldwide. Many studies have described the effects of schistosome infections on humans and animals, but data focusing solely on *S. haematobium* infections, which cause urogenital schistosomiasis are scarce. Our goal was to evaluate, in a systematic manner, how *S. haematobium* infection affects the immune system, gene expression and microbiome of the host. These effects are important because they could lead to increased risk of infections, such as HIV, and bladder cancer. We screened 3,177 studies for potential relevance and included 94 of them in this review. Our analysis showed that *S. haematobium* infection profoundly alters the immune system with a mixed pro-inflammatory and anti-inflammatory response, though with a predominant type 2 immune phenotype and increased regulatory cells. We further found consistent evidence that it impairs local mucosal epithelial barrier integrity, promotes cellular transformation with pro-oncogenic changes in the host, and is associated with microbial alterations in urine, stool, and genital tracts. We discuss how these findings might be interpreted, and the additional research needed, to improve our understanding of *S. haematobium* pathophysiology and ameliorate the potential sequelae of *S. haematobium* infection, such as increased viral infections and cancer.

## Introduction

*Schistosoma haematobium* (*Sh*) is a parasitic trematode affecting over 110 million people worldwide, disproportionately infecting the world's poor, who suffer from limited access to clean water and adequate healthcare [1–4]. Adult *Sh* worms reside predominantly in the veins surrounding the genitourinary tract and release eggs that can become entrapped in genitourinary and reproductive tract organs, causing urogenital schistosomiasis. Schistosome eggs are highly immunogenic and provoke local irritation, inflammation, including granulomatous disease, and fibrosis in the affected tissue leading to chronic symptoms, morbidity, and even organ failure and death. *Sh* eggs that become lodged in the genital tract can result in female or

male genital schistosomiasis, while those in the urinary bladder are categorized as Class A carcinogens, contributing to the development of squamous cell carcinoma in the bladder [2].

Overall, *Sh* infection leads to approximately 1.5 million disability-adjusted life years (DALYs) lost annually [5]. Not included in these DALY calculations is the increasing evidence suggesting that *Sh* may additionally contribute to DALYs lost through its association with increased susceptibility to secondary infections, including HIV, HPV, and other sexually transmitted infections [6–12]. Moreover, a previously infected host is not immune against reinfection. Hence, individuals living in endemic areas frequently become reinfected with *Sh* after successful treatment with praziquantel (PZQ) [13] and can experience additional DALYs lost after reinfection. It remains unknown why the human host does not generate a sterilizing immune response upon reencountering *Sh*. To date, there is no active vaccine against schistosome infections.

*Sh* is the leading risk factor for bladder cancer in *Sh* endemic countries with the highest incidence in the Middle East and Africa [14–16]. Carcinogenesis is a complex process resulting from the accumulation of multiple genetic and epigenetic changes including activation of oncogenes, inactivation or loss of tumor suppressor genes, and alterations in apoptotic gene products. These changes can alter cell proliferation of the urothelium and ultimately lead to urothelial transformation [14]. *Sh*-associated bladder cancer is largely thought to be due to the combination of changes in gene expression plus the chronic inflammation caused by *Sh* eggs lodged within the bladder tissue.

Schistosome infection also has been reported to affect composition of the human microbiome across a variety of sites, including genitourinary and gastrointestinal tracts [17]. In addition to infections, other factors that influence the composition of the microbiome include inheritance, mode of birth, diet, including malnutrition and breastfeeding, environment, geographical locations, hormonal changes, antimicrobial exposures, and age-related changes throughout child- and adulthood, among many others [17–23]. A mounting number of studies demonstrate that the microbiome influences the immune system, gene expression, and the development of human pathology including cancer and immune mediated diseases [17,24]. Infections, including parasitic infections such as *Sh*, may directly affect the composition of the microbiome, and thereby then indirectly affect the host immune phenotype [25,26]. Identifying the links between the microbiome and *Sh* infection may provide prophylactic or therapeutic tools to improve human health [17].

Critically, within schistosome studies, most knowledge to date is derived from studies focusing on *S. mansoni* and other *Schistosoma* species despite *Sh* being the most prevalent schistosome species worldwide. The pattern of immune response, alterations in gene expression or microbiome likely differ from that in *Sh* infection because the bladder and reproductive tract represent immunologic, genetic and microbiome environments distinct from that of the gastrointestinal tract or liver, which are affected during infections with other *Schistosoma* spp. A comprehensive analysis of the pathophysiology leading to increased secondary infections or bladder cancer in the setting of *Sh* infection has not been performed. Gaining a deeper comprehension of these factors could provide valuable insights into the fundamental processes involved and could provide novel targets for treatment and prevention.

To fill this gap, we conducted a rigorous systematic review that aimed to summarize concisely the current knowledge on immune, gene, and microbiome alterations of the host in the setting of *Sh* infection. We sought to consolidate data to enhance our understanding of the specific pathology occurring during this parasitic infection, bringing together disparate immunological, molecular, and microbial findings from an array of human and animal studies.

## Results

Titles and abstracts of 3,177 studies were screened after which 398 full-text papers were assessed for inclusion. A total of 105 studies were included for extraction, and 11 of the 105 were excluded based on the quality assessment using the Downs and Black checklist (Fig 1) [27]. The 11 studies that did not meet Downs and Black criteria of "fair" or better quality are reported in S1 Table. The remaining 94 studies examined effects of schistosome in 3 categories: immune system (N = 66), gene expression (N = 26), and microbiome (N = 6). Each category is discussed in the following sections. Of note, 4 studies reported both on effects of *Sh* infection on the immune system and gene expression, and in these instances the studies were each cited in the relevant section.

The host's immune response to *Sh* infection is dynamic with an initial Th1 predominate response to the acute *Sh* infection followed by a shift to a Th2 predominate response that also includes T regulatory cells (Tregs), B regulatory cells (Bregs), and alternatively activated macrophages (AAM) [28]. This shift to a Th2 predominate response is thought to explain why

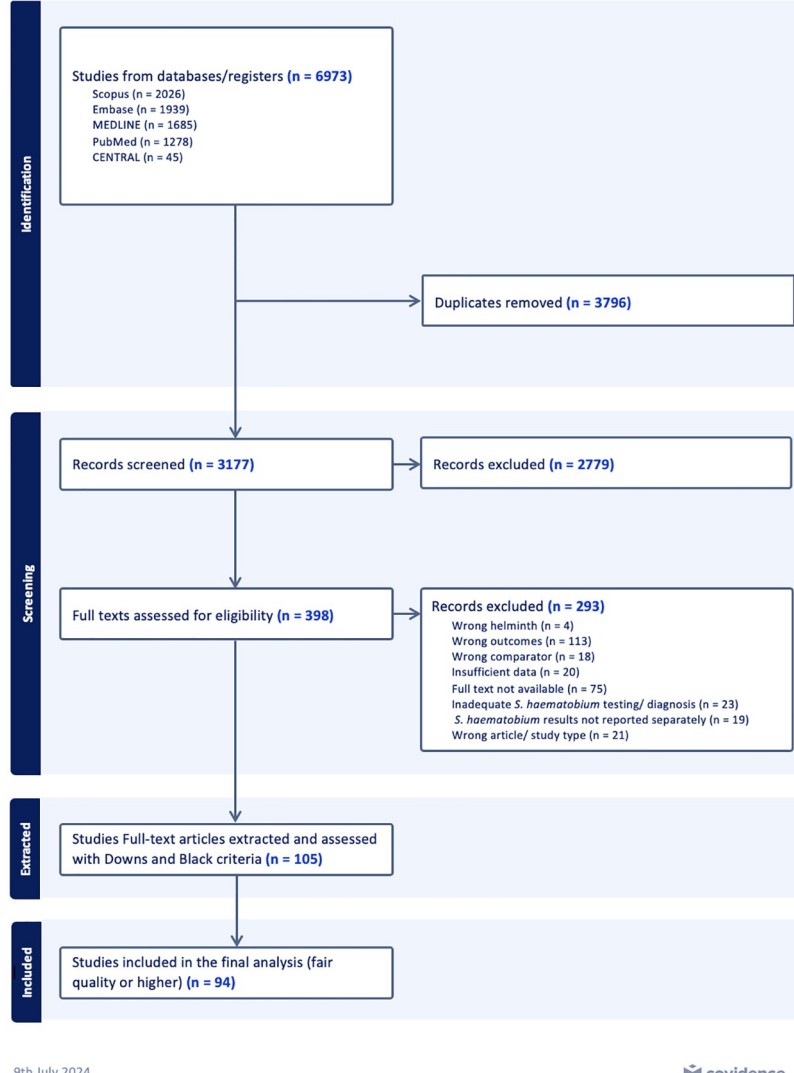

**Fig 1. PRISMA flow diagram of systematic review.**

individuals infected with *Sh* have poorer immune responses to viruses, as well as higher risk of developing bladder cancer [29,30]. Most studies included in this review focused on the consequences of chronic *Sh* infection in humans living in *Sh*-endemic regions. We begin with studies focused on the host's immune system, followed by those on gene expression, and lastly microbiome. Table 1 outlines the organization of the results section.

**Table 1. Overview of the organization of results.**

| Heading | Subheadings |
|---|---|
| **S. haematobium infection and the host immune system** | **Systemic immune responses to Sh infection in humans**<br>*S. haematobium* infection and systemic host chemokine and cytokine immune responses in humans<br>  - Systemic cytokine responses to *Sh* infection<br>  - Variation in cytokine responses to *Sh* by age<br>  - Variation in cytokine responses to *Sh* infection by genitourinary pathology and infection intensity<br>  - Variation in cytokine responses to *Sh* infection by sex<br>  - *S. haematobium* infection and the systemic host cytokine immune response in humans after praziquantel therapy<br>  - Overall synthesis of cytokine studies in *Sh* infection<br>Systemic innate immunity<br>*S. haematobium* infection and the host innate immune response in the systemic circulation in humans<br>  - Dendritic cells<br>  - Monocytes<br>  - Granulocytes<br>  - Innate lymphoid cells<br>Systemic adaptive immunity<br>*S. haematobium* infection and host systemic lymphocytic proliferation in humans<br>*S. haematobium* infection and host T cell immune response in the systemic circulation in humans<br>  - CD4$^+$ and CD8$^+$ T cells<br>  - Memory T cells<br>  - Th1 and Th2 cells<br>  - Th17 cells<br>  - T regulatory cells<br>  - T cell marker expression<br>  - Overall synthesis of T cell studies in *Sh* infection<br>*S. haematobium* infection and host B cell immune response in the systemic circulation in humans<br><br>**S. haematobium infection and the host local genitourinary tissue immune response in humans**<br>  - Overall synthesis of immune studies in *Sh* infection on local tissue<br>*S. haematobium* infection and the local tissue host immune response in patients with bladder cancer<br>  - Overall synthesis of Sh infection and the local tissue host immune response in patients with bladder cancer<br>Experimental models to study effects of *Sh* infection on immune response<br>  - Overall synthesis of experimental immune studies in *Sh* infection |
| **S. haematobium infection and host gene expression** | **S. haematobium infection and host mucosal gene and protein expression in S. haematobium associated bladder cancer**<br>  - Overall synthesis of gene expression studies in *Sh* infection and bladder cancer<br>**S. haematobium infection and mucosal gene expression of the host**<br>**S. haematobium infection and systemic gene expression in blood**<br>  - Overall synthesis of systemic gene expression studies in Sh infection<br>**Experimental models to study S. haematobium infection and host gene, protein expression, and cell cycle dynamics.**<br>  - Overall synthesis of experimental gene expression studies in *Sh* infection |

*(Continued)*

**Table 1.** (Continued)

| Heading | Subheadings |
| --- | --- |
| *S. haematobium* **infection and the host microbiome** | *S. haematobium* **infection and the host microbiome.**<br>- Overall synthesis of microbiome studies in Sh infection |

### *S. haematobium* infection and the host immune system

We begin by examining studies of the systemic immune responses to *Sh* infection in humans categorized into studies focusing on cytokines (N = 27), innate immunity (N = 12), and adaptive immunity (N = 26). We then review studies of local tissue immune responses to *Sh* infection in humans separating studies on humans without (N = 6) and with bladder cancer (N = 3), and end with animal and other experimental models (N = 8), and how these fit with observations from humans. For studies that investigated multiple aspects of the immune response, such as cytokines and cellular changes, each is referenced twice in the relevant section.

### Systemic Immune Responses to *Sh* Infection in Humans

The immune response to schistosome infections is multifaceted and, as with other pathogens, is mediated by both cytokines and chemokines. These molecules orchestrate both innate and adaptive immune responses, serving both as regulators and amplifiers, and may directly or indirectly be involved in pathophysiology (Box 1). CD4$^+$ T helper (Th) cells produce a variety of cytokines that can be grouped based on the cells they are associated with, including Th1, Th2, Th17, and regulatory T cell cytokines. The type 2 immune response associated with chronic schistosome and other helminth infections is marked by the release of Th2-linked cytokines, which promote wound healing and worm expulsion while dampening inflammation [31–35].

References to box 1 right below Box 1: [36–56]

## Box 1 Overview of cytokines and chemokines

Cytokines and chemokines are a diverse group of molecules that play a crucial role in coordinating both innate and adaptive immune responses to pathogens. They can be categorized into two functional groups: pro- or anti-inflammatory cytokines, although some overlap exists. Cytokines primarily originate from immune cells but can also be generated by non-hematopoietic cells including epithelial cells (36).

Th1 cytokine family:

- Mediate pro-inflammatory responses important for immune responses to pathogens.

- Main cytokines are **IFN-γ** and **TNF-α**.

- **IFN-γ** is a key mediator of inflammation in response to invading pathogens, particularly bacteria and viruses, and is produced by Th1 cells and other cell types including Natural Killer (NK) cells.

Th2 cytokine family:

- Mediate host protection during parasitic infections through enhancing tissue repair, control of inflammation, and helminth expulsion.

- Play significant roles in allergic-inflammatory immune responses, inducing eosinophils, basophils, mast cells, and macrophages.

- Can counteract the Th1 cytokines since they are important for anti-inflammatory response and have a role in dampening excessive inflammation which can lead to uncontrolled tissue damage and/ or autoimmune disease.

- A Th2 dominant immune phenotype counteracts Th1 microbicidal action, and therefore a balance between Th1 and Th2 response is critical.

- Main cytokines are **IL-4**, **IL-5**, **IL-10**, **IL-13**, and **IL-33**.

- **IL-4** and **IL-13** can additionally stimulate fibroblasts to produce collagen.

- **IL-13** plays a pro-fibrotic role in the liver during *S. mansoni* infection.

Overlapping Th1 and Th2 cytokine IL-2:

- **IL-2** is predominantly produced by activated CD4$^+$ T cells of either the Th1 or Th2 subset.

- **IL-2** is pertinent for the differentiation of CD4$^+$ and CD8$^+$ T cells into effector and memory T cell subsets following antigen-mediated activation, and it is also crucial for the induction and maintenance of Tregs.

- **IL-2** can have pro- or anti-inflammatory properties depending on the immune milieu.

Immunosuppressive cytokines:

- **IL-10** and **TGF-β** mediate the immune suppressive function of Tregs and can suppress immune responses of any cell type.

Pro-inflammatory cytokines and chemokines, not part of the Th1 family:

- Important for immune activation against invading pathogens.

- The **IL-1** cytokine family, such as **IL-1-α** and **IL-1-β**, regulate inflammation by controlling a variety of innate and adaptive immune processes, including acting as leukocytic endogenous mediator and inducing acute-phase proteins and lymphocyte activation factor.

- **IL-6** is central to host defense and is a potent inducer of inflammation and acute phase proteins.

- **IL-15** is pertinent in anti-pathogen defense by promoting NK and NK T cell development and memory CD8$^+$ T cell function.

- **IL-18** is a pleiotropic cytokine functioning as both a potent inducer of inflammation and an inducer of host defense against pathogens regulating innate and acquired immunity.

- **IL-17** plays a central role in protective immunity against a variety of bacterial and viral pathogens, and promotes pathogenic cytokines in T cell-mediated

autoimmune disease pathology. **IL-17** can be produced by multiple cell types, including Th17 cells, CD4[+] and CD8[+] T cells, NK and NK T cells.

- **IL-21** plays a key pro-inflammatory role in stimulating differentiation of B cells, promoting anti-viral and anti-tumor effects of CD8[+] T cells, and inducing Th17 cells which secrete **IL-17**, recruit neutrophils, and link innate and adaptive immunity. It is produced by T cell subsets and NK T cells.

- **IL-23**, produced by inflammatory myeloid cells, stimulates the development of Th1 cell and Th17 cell responses.

- **Monocyte chemoattractant protein**-1 (**MCP-1**) is a key chemokine that regulates migration and infiltration of monocytes/macrophages.

- **Monocyte chemoattractant protein**-3 (**MCP-3**) is a pluripotent chemokine and can activate all types of leukocytes.

- Macrophages secrete the chemokine **macrophage Inflammatory Protein-1a** (**MIP-1a**) to elicit effects including recruiting inflammatory cells, promoting wound healing, suppressing stem cells, and sustaining effector immune responses.

## *S. haematobium* infection and systemic host chemokine and cytokine immune responses in humans

**Systemic cytokine responses to Sh infection.**   In total, 16 studies assessed systemic cytokine changes in *Sh* infections (Table 2), with an additional 10 studies investigating cytokine responses following PZQ treatment (Table 3). Many of these studies used various antigens (Box 2) to stimulate cells and quantify responses.

In an earlier study examining human peripheral blood mononuclear cells (PBMCs) from two Zimbabwean children with *Sh* eggs in urine, stimulation with schistosome antigens AWA and SEA led to elevated IL-4 and IL-5 levels and higher ratios of these cytokines to IFN-γ, compared to 2 children without *Sh* infection [57].

In a similar study, whole blood of 40 children with *Sh* infection responded to schistosome antigen AWA and SEA stimulation with production of higher levels of Th1 and Th2 cytokines including IFN-γ, IL-2, IL-5, IL-10, and TNF-α, compared to 39 children without *Sh* infection [58]. PZQ treatment increased the production of IFN-γ, IL-2, IL-5, IL-10, and TNF-α in whole blood of previously infected children in response to schistosome antigens AWA and SEA and also increased production of the immunosuppressive cytokine, IL-10, but not the other cytokines, in response to generic antigen stimulation [58].

Circulating IL-10 and INF-γ were elevated in serum of Malian children with *Sh* infection, similarly to the stimulation study above alongside IL-4 and IL-6, compared to individuals without *Sh* infection. In contrast to the stimulation study above, there were no differences seen in IL-2, TNF-α, and IL-5 [59].

A PBMC stimulation study in Gabonese children with *Sh* infection revealed increased macrophage-released chemokines (MCP1-MCAF/MIP-1α) and Th2-related cytokines such as IL-5, IL-10, and IL-13, with a simultaneous decrease of pro-inflammatory cytokines (IFN-γ, TNF-α) compared to children without *Sh* infection [60]. Factors like soil-transmitted

**Table 2.** *S. haematobium* infection and systemic host chemokine and cytokine immune responses in humans.

| Study | Country | Sample size | Sex (female in %) | Age in years | Differences in individuals with *S. haematobium* infection compared to individuals without *S. haematobium* infection | Overall conclusion |
|---|---|---|---|---|---|---|
| Antony et al 2015 (64) | Nigeria | 359 | F (44%) | 2–80 | ↑ IL-6 | IL-6 was elevated in individuals with *Sh* infection. |
| *Ateba-Ngoa et al 2015 (60) | Gabon | 125 | F (49%) | 6–16 | In response to TLR stimulation: ↑ MCP1-MCAF, MIP-1α, IL-5, Il-10, and IL-13 <br><br> ↓ IFN-γ and TNF-α | PBMCs of individuals with *Sh* infection had elevated macrophage-released chemokines (MCP1-MCAF/ MIP-1α), and Th2 cytokines, together with decreased pro-inflammatory cytokines (INF-γ, TNF-α) in response to TLR stimulation, as compared to individuals without infection. |
| Bustinduy et al 2015 (65) | Kenya | 790 | F (50.5%) | 2–19 | ↑ IL-6 at 11- 12yo <br> ↑ IL-10 at 13- 14yo <br><br> = TNF-α | Cytokine response was dependent on age of the individuals with *Sh* infection individuals. <br><br> IL-6 was elevated in individuals with *Sh* infection aged 11–12, while IL-10 was only elevated in individuals with *Sh* infection aged 13–14 compared to individuals without *Sh* infection. |
| Grogan et al 1998 (68) | Gabon | 12 | F (58%) | median 11 | In response to IL-10 blockade: ↑ IFN-γ (in response to AWA) ↑ Lymphocyte proliferation (with and without AWA stimulation) | IL-10 suppressed IFN-γ production and lymphocyte proliferation in individuals with *Sh* infection. |
| Imai et al 2011 (61) | Zimbabwe | 95 | F (% not documented) | 1–5 | ↑ IL-5, IL-10, and IFN-γ | Individuals with *Sh* infection had a mixed Th1 and Th2 cytokine response. |
| Kasambala et al 2023 (66) | Zimbabwe | 136 | F (46%) | Median 51mo | ↑ IL-6 and TNF-α <br><br> ↓ IL-10 <br><br> = IL17A <br> = TGF-β | PSAC with *Sh* infection had higher inflammatory markers, IL-6 and TNF- α, compared to PSAC without infection. |
| *King et al 1996 (69) | Egypt | 38 | F (5%) | 6–30 | In response to SWAP and SEA: ↑ IL-10 <br><br> In response to IL-10 antibody blockage in individuals with *Sh* infection after SWAP/ SEA stimulation: ↑ IFN-γ ↑ Lymphocyte proliferation | IL-10 increased in individuals with *Sh* infection, and blockade of IL-10 led to an increase in IFN-γ and lymphocyte proliferation. |
| King et al 2001 (74) | Kenya | 37 | F (30%) | 7–18 | Individuals with infection with and without bladder pathology: ↑ TNF-α spontaneously, and in response to SEA and PPD <br><br> = IL-4, IL-5, IL-10, and IFN-γ | TNF-α was elevated in individuals with *Sh* infection with bladder pathology. |
| *Labuda et al 2020 (58) | Gabon | 79 | Male and female (%, not documented) | School-children | In response to AWA and SEA: ↑ IFN-γ, IL-2, IL-5, IL-10, and TNF-α <br><br> In response to PHA: = IFN-γ, IL-2, IL-5, IL-10, and TNF-α | Children with *Sh* infection produced elevated systemic Th1 and Th2 cytokines in response to *Sh* antigens, AWA and SEA. |
| *Lyke et al 2006 (59) | Mali | 676 | F (51%) | 4–14 | ↑ IL-6 (only significant in 4–8 yo) ↑ IL-4, IL-10, and IFN-γ <br><br> = IL-2, IL-5, and TNF-α | Children with *Sh* infection had elevated circulating Th1 and Th2 cytokines. |
| Mduluza et al 2001 (57) | Zimbabwe | 4 | Not documented | 10–12 | In response to SEA and AWA: ↑ IL-4:IFN-γ ratio ↑ IL-5:IFN-γ ratio | Participants with *Sh* infection had elevated IL4 to IFN-γ and IL-5 to IFN-γ ratios after stimulation with *Sh* antigens, AWA and SEA. |

*(Continued)*

**Table 2.** (Continued)

| Study | Country | Sample size | Sex (female in %) | Age in years | Differences in individuals with *S. haematobium* infection compared to individuals without *S. haematobium* infection | Overall conclusion |
|---|---|---|---|---|---|---|
| *Meurs et al 2011 (63) | Gabon | 30 | F (42%) | 7–16 | In response to TLR2 stimulation:<br>↑ TNF-α (TLR2 ligand Pam3)<br>↑ TNF-α/ IL-10 ratio (TLR2 ligand Pam3 and FSL-1)<br><br>In response to schistosome SEA and AWA stimulation:<br>↑ TNF-α/ IL-10 ratio (to AWA)<br>↑ IL-10 in response (to AWA/ SEA) | TNF-α response and TNF-α/ IL-10 ratio was higher in individuals with *Sh* infection upon TLR2 stimulation of PBMCs.<br>IL-10 increased only to AWA and SEA stimulation in individuals with *Sh* infection but not to TL2 ligand stimulus. |
| Milner et al 2010 (72) | Zimbabwe | 227 | F (not documented) | 6–60 | ↑ IL-2, IL-10, and IL-23<br><br>↓ IL-4, IL-13, and IL-21<br><br>*Sh* infected with low egg burden:<br>↑ IL-17 and IL-23 | In a population that was *Sh* positive, individuals with *Sh* infection had a mixed Th1/Th2 systemic cytokines at baseline, while Th2 cytokines dominated in individuals without infection. |
| Mutapi et al 2006 (110) | Zimbabwe | 190 | F (54%) | 6–40 | Compared cytokine response among *Sh* infected by age group and infection intensity:<br>↑ IL-4 to SWAP stimulation in 15–16 yo<br>↑ IL-5 to SWAP and ConA stimulation in 15–16 yo<br>↑ IL-10 to SWAP stimulation in < 14 yo<br>↑ IL-10 to ConA stimulation in > 14 yo<br>↑ IFN-γ to ConA stimulation in 13–14 yo<br>High *Sh* egg burden:<br>↑ IL-10 and IL-5<br>In response to IL-10 antibody blockade:<br>↑ IL-5 | Cytokine responses varied by age, infection status, urinary *Sh* egg burden and by the specific antigen stimulus. |
| Remoué et al 2001 (83) | Senegal | 23 | F (48%) | 35–57 | Women with *Sh* infection versus men with *Sh* infection in response to SEA:<br>↓ TNF-α and IFN-γ<br><br>↑ TGF-β and IL-10<br>↑ nuclear matrix protein | Among individuals with *Sh* infection, the cytokine production to SEA stimulation differed between sexes, with women producing less pro-inflammatory cytokines and more IL-10 and TGF-β which can have pro- and anti-inflammatory properties. |
| Van der Kleij et al 2004 (62) | Gabon | 25 | F (40%) | 8–15 | = Monocytes, B cells, and T cells<br><br>In response to LPS:<br>↓ IL-8 and IL-10<br>↓ IL-6 (trend) and TNF-α (trend)<br><br>In response to schistosomal PS:<br>↓ IL-8) and TNF-α<br>↓ IL-10 (trend) and IL-6 (trend)<br><br>In response to schistosomal GL:<br>↑ IL-6<br>↑ IL-8 (trend)<br><br>= TNF-α and IL-10 | Cytokine responses varied in individuals with *Sh* infection and were dependent on the antigen used to stimulate PBMCs.<br>Cytokine response (IL-8, IL-10, IL-6, and TNF-α) was blunted in response to LPS in individuals with *Sh* infection. |

*Sh* = *S. haematobium*, PS = schistosomal phosphatidylserine, GL = Glycolipid, LPS = Lipopolysaccharide, SEA = soluble egg antigen, AWA = adult worm antigen,

Mo = months, PSAC = pre-school aged children, yo = year old

*Studies included participants with additional parasitic infections.

**Table 3. *S. haematobium* infection and the systemic host cytokine immune response in humans after praziquantel therapy.**

| Study | Country | Sample size | Sex (female in %) | Age in years | Differences in individuals with *S. haematobium* infection compared to individuals without *S. haematobium* infection | Effect of praziquantel treatment | Overall conclusion |
|---|---|---|---|---|---|---|---|
| Bourke et al 2013 (87) | Zimbabwe | 72 | F (44%) | 5–17 | N/A | In response to SEA stimulation: ↓ TNF-α, IL-6, IFN-γ, IL-2, IL-8, IL-12p70, IL-21, and IL-23<br><br>In response to CAP stimulation: ↓ IL-8, IL-10, and IL-13<br><br>↓ IL-23<br><br>In response to WWH stimulation: ↑ IL-21<br><br>↓ TNF-α and IL-6<br><br>In response to any schistosome antigen (SEA, CAP, WWH) stimulation: = IL-4, IL-5, and IL-17A<br><br>*6 weeks after treatment* | A pro-inflammatory cytokine pattern was released in response to SEA, a mixed response of pro- and anti-inflammatory cytokines to CAP stimulation and a less inflammatory cytokine response to WWH stimulation in prior individuals with *Sh* infection who underwent PZQ treatment. |
| Grogan et al 1998 (75) | Gabon | 47 | Not documented | Children-adults | N/A | Individuals with prior *Sh* infection who received PZQ 2 years prior comparing individuals with reinfection vs individuals without infection:<br>↓ IL-4 and IL-5 (in response to SEA, AWA)<br>↓ IFN-γ (in response to SEA, AWA)<br>↓ Lymphocyte proliferation (with AWA stimulation only) | Individuals with prior *Sh* infection who became reinfected had decreased Th1 and Th2 cytokines in response to *Sh* antigens, SEA and AWA. |
| Khalil et al 1999 (92) | Egypt | 108 | F (0%) | 10–45 | *Sh*-reinfected:<br>↑ IL-4 | ↓ IL-4<br><br>*12 months after treatment* | IL-4 increased during active *Sh* infection and decreased after treatment. |

*(Continued)*

**Table 3.** (Continued)

| Study | Country | Sample size | Sex (female in %) | Age in years | Differences in individuals with *S. haematobium* infection compared to individuals without *S. haematobium* infection | Effect of praziquantel treatment | Overall conclusion |
|---|---|---|---|---|---|---|---|
| Mduluza at al 2003 (88) | Zimbabwe | 40 | Not documented | 7–16 | Population with high *Sh* endemicity, in response to SEA and SWAP: ↑ IFN-γ, IL-4, and IL-5<br><br>= IL-4 to IFN-γ ratio<br>= IL-5 to IFN-γ ratio<br><br>↓ IL-10 to IFN-γ ratio<br><br><br>Population with low *Sh* endemicity, in response to SEA and SWAP:<br>↑ IFN-γ<br><br>= IL-4 and IL-5<br>= IL-4 to IFN-γ ratio<br>= IL-5 to IFN-γ ratio<br>= IL-10 to IFN-γ ratio | Individuals with prior *Sh* infection with reinfection compared to *Sh* without reinfection after PZQ therapy, in response to SEA and SWAP:<br>↑ IL-4:IFN-γ ratio<br>↑ IL-5:IFN-γ ratio<br>↑ IL-10:IFN-γ ratio<br><br>↑ IL-4 and IL-5<br><br>↓ IFN-γ<br><br>*12 and 18 months after treatment* | Cytokine responses in individuals with *Sh* infection varied between high and low *Sh* endemic areas of residence.<br>Th1-like response dominated during *Sh* infection while the Th2-like responses dominated post treatment in *Sh*-uninfected individuals. |
| Medhat et al 1998 (91) | Egypt | 86 | F (0%) | 9–15 | N/A | Individuals with prior *Sh* infection who became reinfected versus those with prior *Sh* infection who did not become reinfected after PZQ therapy, in response to SWAP stimulation:<br>↓ IL-4 and IL-5<br>↓ IFN-γ (trend)<br><br>*12–18 months after treatment* | Individuals with *Sh* reinfection had lower Th2 cytokines IL-4 and IL-5. |
| Mutapi et al 2001 (71) | Zimbabwe | 602 | Men and women (% F not documented) | 2–86 | ↓ IL-10 (trend) | ↑ IL-10<br><br>*6 weeks after treatment* | IL-10 was inversely associated with *Sh* infection and increased post-treatment. |

(*Continued*)

**Table 3.** (*Continued*)

| Study | Country | Sample size | Sex (female in %) | Age in years | Differences in individuals with *S. haematobium* infection compared to individuals without *S. haematobium* infection | Effect of praziquantel treatment | Overall conclusion |
|---|---|---|---|---|---|---|---|
| Van Den Biggelaar et al 2002 (93) | Gabon | 135 | F (50%) | 5–14 | N/A | Compared cytokine response in individuals with prior *Sh* infection after zero, one, or multiple PZQ treatment courses after 24 months:<br><br>All in response to AWA stimulation:<br><br>Individuals with *Sh* infection versus without infection <u>without PZQ treatment</u>:<br>↑ IL-5<br>↑ IL-13 (trend)<br>= IL-10<br>↓ IFN-γ (trend)<br><br>Individuals with *Sh* infection versus without infection after <u>1 dose of PZQ treatment</u>:<br>↑ IL-5<br>↑ IL-13 (trend)<br>= IL-10 and TGF-β<br>↓ IFN-γ<br><br>Individuals with *Sh* infection versus without infection after <u>multiple doses of PZQ treatment</u>:<br>= IL-5, IL-13, IFN-γ, IL-10, and TGF-β<br><br>Individuals with *Sh* infection and without infection after <u>multiple doses of PZQ treatment</u> in comparison to individuals with *Sh* infection and individuals without infection <u>with no or 1 dose PZQ treatment</u>:<br>↑ IL-5<br><br>*24 months after treatment* | Cytokine production varied depending on current *Sh* infection as well as preceding PZQ treatment.<br><br>In individuals with *Sh* infection at 24 month follow up, Th2 cytokines IL-5 and IL-13 were only elevated in individuals who received 0 or 1 dose of PZQ treatment 24 months before.<br><br>Individuals with *Sh* infection after 0 or 1 dose of PZQ treatment had lower IFN-γ in comparison to the individuals without infection at 24 month follow up.<br><br>Participants who received multiple PZQ courses had the highest IL-5 levels at 24 month follow up regardless of *Sh* status in comparison to the other 2 treatment arms (0 or 1 dose PZQ treatment). |

**Table 3.** (Continued)

| Study | Country | Sample size | Sex (female in %) | Age in years | Differences in individuals with *S. haematobium* infection compared to individuals without *S. haematobium* infection | Effect of praziquantel treatment | Overall conclusion |
|---|---|---|---|---|---|---|---|
| Wilson et al 2013 (84) | Mali | 47 | F (60%) | 5–39 | N/A | *24 hours after treatment*<br><br>↑ IL-5<br><br>*9 weeks after treatment*<br><br>↓ IL-5 and IL-13<br><br>↑ IL-33<br><br>= sST2 (IL-33 receptor)<br><br>↑ Intracellular IL-5 in eosinophils<br>↑ Intracellular IL-13 in eosinophils | IL-5 increased 24 hours post-PZQ treatment.<br><br>In individuals with prior *Sh* infection, IL-33, eosinophilic IL-5 and IL-13 increased 9 weeks post-PZQ. |
| Wilson et al 2013 (85) | Mali | 326 | F (57%) | 5–40 | N/A | ↑ IL-5 and IL-13<br><br>= IL-4<br><br>*24 hours after treatment* | IL-5 increased after 24 hours post-PZQ treatment. |
| *Zinyama-Gutsire et al 2009 (89) | Zimbabwe | 379 | F (80%) | 17–62 | ↑ MIP-1α | ↓ MIP-1α<br><br>*3 months after treatment* | Individuals with *Sh* infection had elevated systemic MIP-1α which decreased post-PZQ. |

*Sh* = *S. haematobium*, SEA = schistosomal egg antigen; CAP = cercariae; WWH = adult worm antigen; MIP-1α = macrophage inflammatory protein-1α

*Studies included participants with additional parasitic infections.

helminths (STH) [59] and filarial co-infections [60] were similarly distributed among the groups. Nevertheless, co-infections in these two studies might have influenced the results.

A factor analysis study in Zimbabwean children similarly found that a composite variable representing IL-5, IL-10, and IFN-γ increased with age and was positively associated with current *Sh* infection as well as with past *Sh* exposure determined by the presence of *Sh*-specific antibodies in serum [61]. Of note, pro-inflammatory IFN-γ results were not reported separately.

A study from an area of Gabon by Van der Kleij et al [62] found varying responses of IL-8 and IL-10 depending on the specific antigen used for stimulation, with lower levels after lipo-polysaccharide (LPS) and schistosomal phosphatidylserine stimulation in individuals with *Sh* infection versus individuals without *Sh* infection, but higher IL-8 in response to the schisto-some glycolipid (GL) antigen. Notably, PBMC stimulation in the presence of a toll-like recep-tor 2 (TLR2)-blocking antibody decreased the cytokine responses induced by schistosomal phosphatidylserine for IL-10, IL-8, and IL-6 [62]. Similarly, Meurs et al [63] found elevated IL-10 and TNF-α levels in Gabonese children with *Sh* infection after stimulation of PBMCs with TLR2-specific and schistosome antigens SEA and AWA, compared to children without *Sh* infection [63]. Meurs et al [63] selected children with *Sh* infection and children without infec-tion from the same area, while the participants without *Sh* infection in van der Kleij's study

## Box 2 Overview of immune cell stimulants

Peripheral blood mononuclear cells (PBMCs) release cytokines through a receptor-mediated cascade, which depends on both the cell type and the specific antigen encountered. Researchers frequently utilize the potential to activate PBMCs using different antigens to examine the cytokine response following antigen stimulation in cells from study participants or in experimental models. This assessment can help ascertain the immune system's functional response to stimulation by specific or general pathogens.

Using different schistosome antigens from different stages of the parasite life cycle, such as worm antigen or egg antigen, is important because each antigen can elicit a different immune response in the host. This variety helps researchers and clinicians better understand the dynamics of the infection and develop more effective diagnostic tools and treatments. Different antigens may also have varying levels of sensitivity and specificity, making them useful in different contexts or stages of the disease.

**Schistosome antigens**

- Soluble egg antigen (SEA)
- Adult worm antigen (AWA)
- Soluble adult worm antigen preparation (SWAP)
- Schistosome phosphatidylserine (PS)

***Mycobacterium tuberculosis* antigen**

- Purified protein derivate (PPD)

**Generic antigens**

- Glycolipid (GL)
- Lipopolysaccharide (LPS)
- Phytohemagglutinin (PHA)
- Pokeweed mitogen (PWM)
- Concanavalin A (ConA)
- TLR-2 stimulants: FSL-1 and Pam3
- TLR-9 stimulants: CpG and R848

were recruited from a nonendemic neighboring semi-urban area and were more likely truly *Sh*-uninfected [62]. Both studies suggest that the immune response to *Sh* operates via TLR ligand signaling [62,63].

Five studies demonstrated higher systemic levels of IL-6, which integrates the immune defense against infections [47], in individuals with *Sh* infection versus individuals without *Sh*

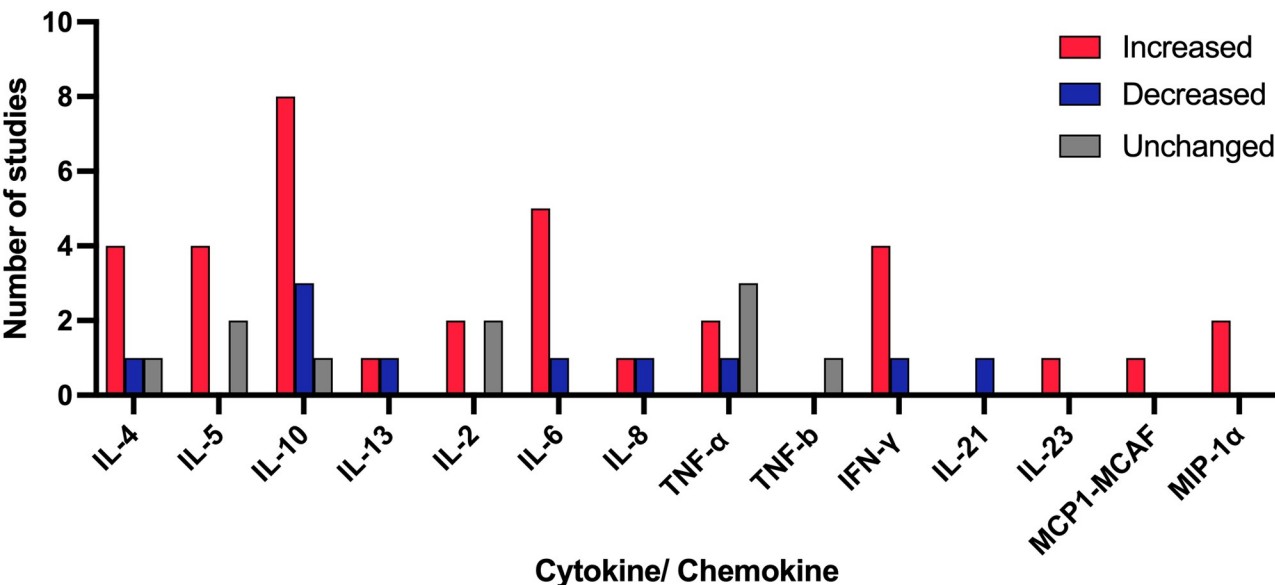

**Fig 2. Systemic cytokine and chemokine changes in the setting of *S. haematobium* infection in humans.** Overview of results of the 14 studies assessing systemic cytokine trends at baseline or after antigen stimulation in individuals with *Sh* infection individuals of the total 27 studies investigating systemic cytokines in this review. Data on counts of the reported cytokines were collected. *Sh* infection was associated with an increased release of systemic Th2 cytokines (IL-4, IL-5, Il-10): Four out of 5 studies (80%) reported an increase in IL-4, 4 out of 5 studies (80%) reported an increase in IL-5 and 8 out of 12 studies (67%) showed an increase in IL-10 in the setting of *Sh* infection. The regulatory cytokine IL-2, which is pertinent for induction and maintenance of T cell subsets including Tregs, was elevated in 2 out of 3 studies (67%) but also unchanged in 2 out of the 3 studies (67%) in individuals with *Sh* infection depending on which specific antigen was used. Increased proinflammatory cytokines (IL-6, IFN-γ, IL-23) were associated with *Sh* infection: Five out of 6 studies (83%) reported increased IL-6 levels, 4 out of 6 (67%) reported an increase in IFN-γ and one study reported an increase in IL-23 in the setting of *Sh* infection. One study (100%) reported an increase in macrophage related inflammatory cytokines MCP1-MCAF and 2 studies (100%) an increase of MIP-1α in individuals with *Sh* infection. Graph developed using GraphPad Prism.

infection [59,62,64–66]. Serum IL-6 levels were also higher among those with active infections than those with positive *Sh* serologies without active infection [64] and in individuals with *Sh* infection with urinary tract pathology [67], supporting a likely role of IL-6 in the pathogenesis of urogenital schistosomiasis.

IL-10 has both immunomodulatory and anti-inflammatory properties [43] that appear to be important during *Sh* infection. Two studies showed that blockade of IL-10 with monoclonal antibodies of isolated PBMCs led to increased IFN-γ production and enhanced lymphocyte proliferation following stimulation with schistosome antigens SEA and AWA in people with *Sh* infection [68,69]. A Zimbabwean study assessing the cytokine response to AWA stimulation of PBMCs revealed that IL-10 was most strongly correlated with *Sh* infection intensity, and that participants with high intensity infection also tended to produce lower IL-5 levels [70]. The addition of IL-10 blocking antibodies to PBMCs isolated from individuals with *Sh* infection led to an increase in IL-5 response [70], suggesting that IL-10 may be exerting its modulatory effect by down-regulating IL-5 production. In contrast, in a large Zimbabwean cohort, IL-10 levels were lower in plasma of those with *Sh* infection, and subsequently increased when measured 6 weeks after PZQ treatment [71]. Similarly, serum of pre-school aged children with *Sh* infection had lower IL-10 levels compared to individuals without *Sh* infection, and had higher pro-inflammatory IL-6 and TNF-α [66]. In conclusion, the diverse findings on IL-10 in *Sh* infection highlight its complex role in the immunopathogenesis of *Sh* infection, with studies indicating both modulatory and anti-inflammatory effects on cytokine responses, and dynamic changes in systemic levels following treatment.

In general, *Sh* infection appears to result in systemic alterations in an array of cytokines and chemokines, with most studies describing an increase in the Th2 cytokines IL-4, IL-5, and IL-10, as well as Th1 cytokines IL-6 and IFN-γ (Fig 2). This mixed pro-inflammatory and anti-inflammatory response observed in *Sh* infection highlights the complexity of immune responses to this parasite. While the prevailing notion of a type 2 immune state is confirmed here, this synthesis further identifies a co-existing pro-inflammatory environment in chronic *Sh* infection.

**Variation in cytokine responses to *Sh* by age.** A Zimbabwean study found elevated circulating cytokines IL-2, IL-10 and IL-23 and lower IL-4, IL-13, and IL-21 in individuals with *Sh* infection aged 6–60 years compared to individuals without *Sh* infection [72]. Cytokine patterns varied with age, with increasing IL-4 and IL-10 levels, and decreasing IL-5 levels in plasma, in those with *Sh* infection as age increased [72]. Notably, all participants had detectable parasite-specific antibodies, particularly IgM reflecting recent schistosome exposure, but only those with *Sh* eggs in urine were considered individuals with *Sh* infection [72]. Given the suboptimal sensitivity of urine microscopy, particularly among females, some infected participants may have been misclassified as individuals without *Sh* infection [73]. In agreement with studies above, this cross-sectional study shows that individuals with *Sh* infection have mixed Th1/Th2 systemic cytokines across the lifespan, while Th2 cytokines dominated in uninfected people who were *Sh*-seropositive, indicating past exposure to *Sh* [72].

In another Zimbabwean study, responses to antigen stimulation varied by age of the person with *Sh* infection. Adolescents with *Sh* infection aged 15–16 years showed the highest IL-4 and IL-5 responses to SWAP stimulation of whole blood, and similarly IL-5 peaked in this age group after ConA stimulation [70]. IFN-γ levels peaked at age 13–14 years in response to generic antigen stimulation, but not SWAP stimulation. IL-10 was highest in serum of children in response to SWAP but was lower in participants older than 14 years and decreased further with age. In contrast, stimulation with a ConA led to IL-10 levels that were lowest in children aged 6–12 years with an increase starting in aged 13–14 years, peak at age 15–16 years, and a decline in older participants though those over age 16 years still had higher levels than children aged 6–12 years [66].

In younger children, TNF-α was higher in serum of pre-school aged children with *Sh* infection, while IL-6 was only found to be higher in children aged 11–12 years in a second study of children aged 2–19 years [65]. Consistent with the Mutapi study above, Bustinduy et al also reported increased IL-10 in 13–14-year-olds with *Sh* infection compared to their age-matched counterparts without *Sh* infection [65]. These studies uniformly emphasize the age-related variability in cytokine response.

**Variation in cytokine responses to *Sh* infection by genitourinary pathology and infection intensity.** A study in Kenyan individuals with *Sh* infection and concomitant bladder pathology assessed by ultrasonography reported a 12-fold higher serum TNF-α production at baseline and 13-fold higher production in response to schistosome and purified protein derivative antigen stimulation than in persons with *Sh* infection without bladder pathology [74]. This implies that pro-inflammatory TNF-α may contribute to the genitourinary damage during *Sh* infection [74]. Conversely, in a Gabonese cohort, *Sh* urinary egg burden correlated inversely with IFN-γ levels in PBMCs after AWA or SEA stimulation from individuals who developed *Sh* reinfection two years after PZQ treatment, suggesting a lower pro-inflammatory immune environment in those with the highest *Sh* burden [75].

**Variation in cytokine responses to *Sh* infection by sex.** Sex hormones affect immune function and lead to differences in the immune phenotype and disease response between women and men [76,77]. Furthermore, sex differences in immune response also vary over the lifespan and with reproductive status, which is particularly pronounced in women with

fluctuations due to the menstrual cycle as well as during pregnancy, breastfeeding, and menopause [78–82]. In a Senegalese cohort, women with *Sh* infection produced lower levels of TNF-α and IFN-γ following SEA stimulation of PBMCs, while they produced more TGF-β and IL-10 and had higher expression of the apoptosis marker nuclear matrix protein, compared to men with *Sh* infection with comparable egg burdens and schistosome circulating anodic antigen (CAA) levels [83]. Although this was a small study consisting of 23 participants, these findings suggest that women with *Sh* infection produce less pro-inflammatory Th1 cytokines, and more anti-inflammatory IL-10 than men. Women's higher production of TGF-β, which is mainly produced by fibroblast and epithelial cells, may have either pro- or anti-inflammatory immune effects [44,45].

**S. haematobium infection and the systemic host cytokine immune response in humans after praziquantel therapy.**  Wilson et al conducted two cohort studies comparing cytokine response in plasma of Malian individuals with *Sh* infection at time-points 24 hours after PZQ treatment and 9 weeks post-treatment [84,85]. In one cohort, systemic IL-5 increased 24 hours post-PZQ, around the time of adult worm death, but decreased at 9 weeks, while IL-33 was higher at 9 weeks post-treatment [84]. Further, intracellular eosinophil IL-5 and -13 both increased at 9 weeks, and sST2 (IL-33 receptor) was associated with intracellular IL-13, suggesting a role of IL-33 in the human response to *Sh* infection [84]. In the second Malian cohort, IL-5 increased 24 hours post-treatment, as did IL-13 in plasma [85]. Eosinophils increased at 9 weeks, associated with an IL-5 boost and pre-treatment infection intensities. Of note, 13.9% of a subset of participants who provided stool were found to have concomitant *S. mansoni* infection [85]. Together, these studies suggest a time dependent cytokine response post-PZQ treatment likely due to an initial surge of *Sh* antigen release from dying adult worms which diminishes over time [86].

Cytokine production after PZQ treatment also appears to differ between children and adults and with specific antigen stimulation. In those with prior *Sh* infection, stimulation of whole blood cultures with egg-specific antigen SEA led to a pronounced pro-inflammatory cytokine pattern with elevated TNF-α, IL-6, IL-8, IFN-γ, IL-12p70, IL-21, and IL-23 6 weeks post-treatment, while cercariae-specific antigen CAP led to a mixed pattern of pro- and anti-inflammatory cytokines with elevated IL-8, IL-13, and IL-10 and lower IL-23 [87]. Stimulation with adult worm-specific antigen WWH evoked the lowest inflammatory responses, reflected by low IL-6 and TNF-α and elevated IL-21 which can have pro- and anti-inflammatory properties [87], likely reflecting the low immunogenic potential of adult worms while they are living in the bloodstream [1]. In the same study, a schistosome egg-specific pro-inflammatory cytokine profile and adult worm-specific Th-2 and Th-17-associated cytokine profiles were associated with lower risk of *Sh* reinfection. Overall, Bourke et al report increased inflammatory, Th1 and Th17 cytokines after PZQ after egg-specific stimulation, while Th2 cytokines were unchanged [87].

Mduluza et al [88] assessed cytokine responses in PBMCs in response to schistosome antigens SEA and SWAP in two Zimbabwean cohorts residing in a high or low *Sh* endemic area and with or without *Sh* infection. Individuals with *Sh* infection from both areas produced higher levels of IFN-γ while IL-4 and IL-5 were only elevated in individuals with *Sh* infection from the high endemic area compared to those without *Sh* infection [88]. PZQ treatment decreased the IFN-γ response, while IL-4 and IL-5 remained elevated in both cohorts compared to persons without *Sh* infection. At enrollment there was no difference in cytokine ratios, while ratios increased towards a Th2 phenotype 12 and 18 months post-PZQ therapy in those who remained uninfected versus those who became reinfected in areas of both high and low endemicity. Specifically, those who became reinfected showed a more pronounced Th1 phenotype [88]. A weakness of this study was lack of data analysis by host sex, despite its

potential impact on the immune response. This study highlights a mixed immune response to *Sh* infection and the putative role of Th2 immunity in *Sh* resistance.

The pro-inflammatory chemokine macrophage inflammatory protein-1 (MIP-1α) was elevated in plasma of individuals with *Sh* infection compared to Zimbabwean adults without infection, correlated with urinary *Sh* egg counts, and decreased 3-months post-PZQ therapy [89]. While MIP-1α can enhance expression of HIV entry molecules CCR5 and CXCR4 on HIV target cells such as CD4+ T cells [90], MIP-1α did not vary by HIV infection status in this Zimbabwean study. This is pertinent since some studies have shown associations between HIV and *Sh* infections [6–8,12].

In a Gabonese cohort, *Sh* reinfection was associated with lower IL-4, IL-5, and IFN-γ levels after schistosome stimulation with AWA and SEA of PBMCs compared to those who remained uninfected 2-years after PZQ treatment, suggesting that these lower cytokine levels could possibly predict risk for reinfection [75]. Of note, this study did not document sex distribution, nor was age clearly defined. Similarly, PBMCs of Egyptian male children with previous *Sh* infection who had become reinfected 12–18 months post-PZQ had lower IL-4 and IL-5 production in response to schistosome antigen SWAP, compared to male children who were not reinfected. Depletion of CD4+ T cells abrogated this response, suggesting that they are the source of these cytokines [91]. These two studies, combined with the Mduluza study discussed above, consistently suggest that an impaired type 2 response post-treatment is associated with susceptibility to reinfection. In contrast, in an Egyptian cohort of male children and adults, higher IL-4 levels were found in individuals who became reinfected compared to those who did not. Further, IL-4 was associated with higher egg burden and remained persistently elevated even after treatment [92].

A study in Gabon showed that plasma cytokine levels varied in individuals with prior *Sh* infection after two years depending on if they received treatment and how many PZQ treatment courses they received [93]. Individuals who were untreated and those who received a single dose of PZQ had higher IL-5 levels and a trend towards higher IL-13 levels in response to schistosome antigen AWA compared to people without infection at the two-year follow up who had not been treated. Individuals who cleared infection after a single dose of PZQ had the highest IFN-γ, suggesting a potential protective effect of this cytokine. Moreover, repeated treatment led to high IL-5 and low IFN-γ production in response to AWA in comparison to no or single treatment, but was not protective against *Sh* reinfection. The authors concluded that overall cytokine response to AWA remained altered in previously persons with *Sh* infection despite treatment [93]. On further analysis, the authors also reported that elevated levels of schistosome-specific IL-10 preceded *Sh* reinfection, and high levels of IL-5 were associated with hematuria.

**Overall synthesis of cytokine studies in *Sh* infection.** Overall, *Sh* appears to evoke aspects of both pro- and anti-inflammatory responses in the host. Nine *Sh* studies align with the prevailing hypothesis that a type 2 immune response predominates in most helminth infections [33–35,94], accompanied by an increase in Th2 related cytokines and decrease in Th1 cytokines [57,60,62,63,70,75,92].

Notably, ten other studies showed evidence of pro-inflammatory responses in individuals with *Sh* infection compared to individuals without *Sh* infection, including higher IL-6 [59,62,64–66,87], IFN-γ [59,70,88], and pro-inflammatory chemokine MIP-1α [89]. Supporting the complexity of this area, seven other studies have documented a hybrid response which includes pro- and anti-inflammatory cytokine responses [58,59,61,63,66,70,72].

Cytokine responses to PZQ treatment are also complex. Taken together, most studies support a conclusion that a strong Th2 immune response is triggered by dying *Sh* worms [85]. In the ensuing time after treatment, most studies indicate further immune changes including a

continued increase of Th2 cytokines [71,84,85,88,93] and decrease in pro-inflammatory cytokines and chemokines including IFN-γ [88,93] and MIP-1α [89]. It remains to be determined how long this Th2 predominant immune phenotype persists and if it ever reverts to a more balanced Th1/ Th2 phenotype in the absence of further helminth infections. Importantly, higher levels of Th2 cytokines have been linked to reduced likelihood of reinfection, while lower levels correlated with reinfection in two separate studies [75,91]. Additional studies to understand the protective role and duration of Th2 cytokines could have high relevance for infection prevention and elimination.

A caveat to these overall cytokine studies in *Sh* is the complexity of assessing immune responses in *Sh* infection. It is not possible to elucidate how long the individual has been infected, nor to account fully for other contributing factors, including age [65,72], sex [83], and infection history both past and current [95]. Further complicating comparisons across studies, varied chemokine and cytokine responses to stimulation by different antigens in different environments may also affect results [62,87]. Indeed, many studies included participants with other parasitic infections, mostly other helminths [59,60,62,69,89] and malaria [63]. These other infectious stimuli likely alter the individual's immune response, and therefore many findings cannot be attributed to *Sh* infection alone.

Finally, there are likely significant differences between systemic and local mucosal cytokine production or secretion during *Sh* infection in the genitourinary tract. It will be critical to increase our understanding of the local tissue immune response to genitourinary tissue damage, which also might increase the risk of acquiring other infections such as HIV [96] or establishing a permissive environment for the development of bladder cancer [30]. The single study that correlated cytokines with organ damage found higher systemic levels of the pro-inflammatory cytokine TNF-α in children with *Sh* infection with genitourinary pathology assessed by ultrasonography [74]. Additional studies to link cytokine activity with genitourinary pathology are critically needed to understand the tissue immune milieu and strategies to prevent mucosal pathology. It remains to be determined whether a pro-inflammatory immune response occurs in reaction to *Sh*-induced tissue damage or if it plays a causative role in tissue damage. Longitudinal studies to assess associations between organ pathology and immune responses, as well as effects post-treatment, would provide much needed additional insight.

Accessing tissue samples is more challenging and costly than obtaining peripheral blood, which is likely why most studies concentrate on systemic cytokine responses while trying to relate them to local mucosal processes in the genitourinary tissue, instead of studying cytokine responses directly at the affected organ. Some of these mucosal studies do exist and are discussed in more detail below.

## Systemic innate immunity

### *S. haematobium* infection and the host innate immune response in the systemic circulation in humans

A total of 12 studies have assessed changes in innate immune cells seen in *Sh* infections (Table 4).

**Dendritic cells.** Antigen-presenting dendritic cells (DCs) are among the first immune cells to contact invading parasites, and they respond by sending specific or generic signals that polarize T cell responses [97,98]. DCs can be divided into CD11c+ myeloid DCs (mDCs) and CD123+ plasmacytoid DCs (pDCs) [99], both of which are immature DCs with distinct roles. MDCs are believed to preferentially induce T cell responses to invading pathogens while pDCs are presumed to contribute to innate antiviral responses and self-antigen tolerance [100,101].

**Table 4. *S. haematobium* infection and the host innate immune response in the systemic circulation in humans.**

| Study | Country | Sample size | Sex (female in %) | Age in years | Differences in individuals with *S. haematobium* infection compared to individuals without *S. haematobium* infection | Effect of praziquantel treatment | Overall conclusion |
|---|---|---|---|---|---|---|---|
| Appleby et al 2014 (108) | Zimbabwe | 100 | F (59%) | Mean 16.4 (5–>16) | = Monocytes<br>= CD14 expression by monocytes<br><br>In > 16 years of age:<br>↓ CD16 expression by monocytes | N/A | Individuals with *Sh* infection had lower expression of the activation marker CD16. |
| Everts et al 2010 (102) | Gabon | 43 | F (21%) | 17–39 | ↓ mDCs and pDCs<br>↓ HLA-DR expression on mDCs and pDCs<br><br>In response to LPS stimulation:<br>↓ IL-6, IL-10, IL-12, and TNF-α by mDCs<br>↓ MAPK by mDCs<br><br>= PDL-1 by T cells<br><br>After co-culture with mDCs from participants with *Sh* infection:<br>↓ IL-4 and IFN-γ production by naïve T cells<br>↓ CD25 expression by naïve T cells | N/A | *Sh* infection led to decrease and functional impairment in DCs subpopulations, mDCs and pDCs, leading to T cell hyporesponsiveness in individuals with *Sh* infection. |
| Hagan et al 1985 (114) | The Gambia | 50 | F (28%) | 8–13 | N/A | ↓ Eosinophils in particpants with reinfection<br><br>*12 months after treatment* | Children with prior *Sh* infection who remained uninfected 1 year after successful PZQ therapy had increased circulating eosinophils compared to reinfected, suggesting a protective effect of eosinophils against *Sh* reinfection. |
| Kasambala et al 2023 (66) | Zimbabwe | 136 | F (46%) | Median 51mo | = Neutrophils | N/A | Neutrophil counts were similar between PSAC with and without *Sh* infection. |
| Kleppa et al 2014 (107) | South Africa | 44 | F (100%) | 15–23 | ↑ Monocytes<br>↑ CCR5$^+$ Monocytes | ↓ CCR5$^+$ Monocytes<br><br>*7–8 months after treatment* | Individuals with *Sh* infection had higher circulating monocytes, and increased expression of the HIV entry molecule CCR5, which decreased post-PZQ. |
| Mbow et al 2013 (113) | Senegal | 26 | F (50%) | 5–14 | ↑ Granulocytes, eosinophils in *Sh* with GU pathology vs without pathology<br><br>= Absolute neutrophil count | N/A | GU pathology in individuals with *Sh* infection was associated with increased granulocytes and eosinophils. |
| Meurs et al 2011 (63) | Gabon | 30 | F (42%) | 7–16 | ↑ Eosinophils (trend) | N/A | *Sh* infection was associated with eosinophilia, although not statisitically signficant. |

(*Continued*)

**Table 4.** (Continued)

| Study | Country | Sample size | Sex (female in %) | Age in years | Differences in individuals with *S. haematobium* infection compared to individuals without *S. haematobium* infection | Effect of praziquantel treatment | Overall conclusion |
|---|---|---|---|---|---|---|---|
| Nausch et al 2012 (103) | Zimbabwe | 61 | F (56%) | 5–45 | In all age groups:<br>= pDCs<br>= HLA-DR andCD86 expression on pDcs<br><br>In 5–10 years of age:<br>↑ mDCs<br>↑ HLA-DR expression on mDCs<br><br>In > 10 years of age:<br>↓ mDCs<br>= HLA-DR and CD86 expression on mDCs | N/A | The response of DC populations to *Sh* infection varied by age with increasd mDC in aged 5–10 and decreased mDC seen only in individuals above age of 10. |
| Nausch et al 2015 (117) | Zimbabwe | 72 | F (54%) | 6–18 | ↓ ILC2<br><br>= TSLP and IL-33<br><br>In 14–18 years of age:<br>↑ IL-4, IL-5, and IL-13 associated with increased *Sh* infection | ↑ ILC2<br>↑ TSLP<br><br>*6 weeks after treatment* | *Sh* infection was associated with lower circulating ILC2 levels which were restored after PZQ therapy. |
| Shariati et al 2001 (207) | Malia, Nigeria | 33 | F (3%) | 19–37 | ↑ Eosinophils | N/A | *Sh* infection was associated with peripherial eosinophilia. |
| Wilson et al 2013 (84) | Mali | 47 | F (60%) | 5–39 | N/A | = Eosinophils<br><br>*9 weeks after treatment* | There was no difference in eosinophils in individuals with prior *Sh* infection d pre- or post-PZQ. |
| Wilson et al 2013 (85) | Mali | 326 | F (57%) | 5–40 | N/A | ↑ Eosinophils<br><br>*9 weeks after treatment* | Eosinophils increased in individuals with prior *Sh* infection 9 weeks post-PZQ. |

*Sh* = *S. haematobium*, DCs = dendritic cells, mDCs = myeloid dendritic cells, pDCs = plasmacytoid dendritic cells, TLR = toll-like receptor, PSAC = pre-school aged children, MAPK = Mitogen-activated protein kinase

Since DCs are the first to encounter the parasite *Sh* in all its forms, including eggs that migrate into tissue, it has been hypothesized that they may play a part in the known T cell hyporesponsiveness occurring in chronic *Sh* infections [102].

Two studies have reported that individuals with *Sh* infection had lower mDCs [102,103] and pDCs [102] than individuals without *Sh* infection. In adults in Gabon, both mDCs and pDCs had lower HLA-DR expression in blood [102], while in children and adults in Zimbabwe no differences were seen in activation marker expression (HLA-DR or CD86) [103]. Interestingly, when the Zimbabwean analysis was restricted only to children aged 5–10 years, those with *Sh* infection had more mDCs and lower levels of HLA-DR expression than children without *Sh* infection [103]. On further analysis, both pDCs and mDCs from the Gabonese adults with *Sh* infection had reduced capacity to respond to toll-like-receptor (TLR) ligands with decreased effector marker expression (CD40, CD80, HLA-DR, and CCR7) compared to adults without *Sh* infection, which was more pronounced in mDCs with decreased cytokine

production (IL-6, IL-12, TNF-α, IL-10) and mitogen-activated protein kinase signaling [102]. Co-culture of mDCs isolated from these adults with *Sh* infection with naïve T cells led to lower production of IL-4 and IFN-γ and expression of the activation marker CD25, emphasizing a potential role of DCs functional impairment in mediating T cell hyporesponsiveness in chronic *Sh* infection [102].

Overall, there appears to be an age-related pattern in DCs immune response to *Sh* infection which may reflect the chronicity of the infection, with lower levels of mDCs particularly in adults with *Sh* infection. These lower circulating mDCs levels might correlate with impaired mDCs response in chronic *Sh* infection, and/or could represent increased recruitment of this cell population to the target sites of *Sh* infection. Such a decrease in systemic mDCs in blood could have further implications such as an impairment of immune response against concurrent bacterial or viral infections, for which the host immune response which relies on antigen presenting cells to induce pertinent T cell responses [104].

**Monocytes.** Monocytes are circulating white blood cells with phagocytotic capabilities. They can differentiate into macrophages or dendritic cells after migration from blood into tissue, and can express the activation marker, CD16 [105], linking humoral and innate cellular immune responses [106]. Three studies in total have examined circulating monocytes in *Sh* infection.

In South African women with *Sh* infection, circulating monocytes were increased and had higher expression of the HIV entry molecule CCR5, compared to women without *Sh* infection. However, no difference was observed in monocyte CCR5 expression in cells collected from the cervix in the same women [107].

In two other studies, no difference was seen in circulating monocytes between individuals with *Sh* infection and participants without *Sh* infection [62,108], but one reported lower CD16 expression in systemic monocytes of participants with *Sh* infection aged 16 and above from Zimbabwe [108]. This lower CD16 expression could lead to weaker innate immune responses in those with *Sh* infection. Further, healthy participants without *Sh* infection had a positive correlation between CD16 monocyte expression and IgG and IgG1 levels against schistosome antigen SEA and WWH, which has been associated with resistance to infection [109,110].

The disparity in monocyte findings of the above studies could be attributed to variations in sex distributions among the two study populations. The study in Zimbabwe consisted of both sexes (60% females) with a mean age 16.4 years, while all 44 female participants were above 14 years of age in the South African study. Given evidence of sex-dependent immune responses to schistosome infection [83], these differences in study populations could explain the higher monocyte levels seen in the Kleppa study.

**Granulocytes.** Eosinophils have been well established to play a critical effector cell role in the immune response against helminth infections and are a cardinal sign of parasitic infection [111]. In *S. mansoni* infections, eosinophils can kill cercariae and adult worms in a degranulation- and IgE-dependent manner [112]. Five studies examined eosinophils in *Sh* infection.

People with *Sh* infection in Senegal who had sonographic genitourinary pathology exhibited higher eosinophil and granulocyte levels compared to individuals with *Sh* infection without pathology and/or people without *Sh* infection [113]. This might indicate the heightened immune responses in the presence of ongoing *Sh*-mediated genitourinary pathology, implying a pathogenic role of eosinophils in chronic inflammation and tissue damage. No differences were reported in neutrophils between Senegalese adults with and without *Sh* infection and children in this study [113] nor in Zimbabwean pre-school aged children [66].

The effect of PZQ treatment on eosinophil counts is inconclusive. One study reported no change 9 weeks post-treatment in 47 Malian children and adults [84] while another study by the same group showed increased eosinophils 9 weeks post-treatment in 326 Malian children and adults [85]. Higher eosinophil counts post-treatment were positively associated with reinfection 2 years after treatment [85]. In contrast, a study from The Gambia reported that children with prior *Sh* infection who remained uninfected one year after PZQ treatment had higher circulating eosinophils than children who became reinfected [114]. Of note, the study in Mali did not report eosinophil counts at the 2-year follow up, and hence used the preceding eosinophilia as a predictive factor for reinfection. In contrast, the study in The Gambia reported the association between *Sh* reinfection and low eosinophil count at 12 months after PZQ-treatment and therefore assessed the association of low eosinophils and *Sh* infection at the same time point.

**Innate lymphoid cells.**   Innate lymphoid cells (ILC) are a newly discovered immune population that mainly reside at mucosal sites and play a critical role in mucosal immunology. ILCs can be categorized into three distinct groups that mirror the Th1, Th2, and Th17 cell subsets. ILC1, mirroring Th1, produce IFN-γ upon activation to promote host defense against pathogens, particularly of viral and bacterial origin. ILC3s, similar to Th17 cells, promote tissue repair, contain commensal bacteria, and protect against pathogenic bacteria [115,116].

Similar to Th2 cells, ILC2s are elicited during parasitic infections and regulate tissue immunity and inflammation by producing Th2 cytokines (IL-4, IL-5- and IL-13) to recruit and activate eosinophils and alternatively activated macrophages. In this way, ILC2s mediate anti-helminth immunity and can also be associated with allergic reactions with increased presence in bronchial secretions of asthmatic patients [116].

Thus far only Nausch et al [117] have investigated the impact of *Sh* infection on ILCs. People with *Sh* infection had lower circulating ILC2 frequencies versus people without *Sh* infection, a difference that was most profound in the youngest aged 6–9 years. ILC2 frequencies also negatively correlated with infection intensity assessed by urinary egg burden. Upon re-evaluation of 12 of the 36 previously participants with *Sh* infection 6 weeks after PZQ therapy, ILC2 frequencies had increased to frequencies comparable to those reported in participants without *Sh* infection. Th2 cytokines (IL-4, IL-5 and IL-13) were only associated with infection intensity in the oldest age group aged 14–18 years of age [117]. This observed decrease in systemic ILC2 in individuals with *Sh* infection may be due to increased migration to affected tissues, initiating a local immune response leading to fewer ILC2s detectable in blood. Conclusions about the effects of PZQ are limited by the follow-up to one third of individuals who were initially infected.

## Systemic adaptive immunity

In total, 25 studies have assessed changes in adaptive immune cells seen in *Sh* infections (Tables 5–7).

## *S. haematobium* infection and host systemic lymphocytic proliferation in humans

In three studies, lymphocyte proliferation was similar among individuals with *Sh* infection and without *Sh* infection in response to stimulation by generic antigens [57,88,118] and by schistosome antigens AWA and SEA before and after PZQ chemotherapy [119]. In contrast, in a Gabonese cohort, lymphocyte proliferation was reduced after schistosome antigen stimulation with AWA but not egg antigen SEA [75] in peripheral blood from those with versus without

**Table 5.** *S. haematobium* **infection and host systemic lymphocytic proliferation in humans.**

| Study | Country | Sample size | Sex (female in %) | Age in years | Differences in individuals with *S. haematobium* infection compared to individuals without *S. haematobium* infection | Effect of praziquantel treatment | Overall conclusion |
|---|---|---|---|---|---|---|---|
| Feldmeier et al 1981 (118) | Sudan, Germany (different African immigrants from *Sh* endemic countries) | 45 | Not documented | 6–44 | In response to antigen stimulation (PHA, PWM, CON-A, PPD, *Sm*-antigen):<br>= Lymphocyte proliferation | N/A | *Sh* infection had no impact on lymphocyte proliferation to various antigen stimuli. |
| Grogan et al 1998 (68) | Gabon | 12 | F (58%) | Median 11 | IL-10 blockade:<br>↑ Lymphocyte proliferation (with and without AWA stimulation) | N/A | IL-10 blockade augmented lymphocyte proliferation in individuals with *Sh* infection. |
| Grogan et al 1998 (75) | Gabon | 47 | N/A | Children-adults | *Sh*-reinfected vs uninfected s/p PZQ 2 years prior:<br>↓ Lymphocyte proliferation (with AWA stimulation only) | N/A | Individuals with prior *Sh* infection who became reinfected had decreased lymphocyte proliferation in response to *Sh* antigens. |
| Mduluza et al 2001 (57) | Zimbabwe | 4 | Not documented | 10–12 | = Lymphocyte proliferation to PHA, SEA and SWAP | N/A | Lymphocyte proliferation remained unchanged in individuals with *Sh* infection to various antigen stimuli. |
| Mduluza at al 2003 (88) | Zimbabwe | 40 | Not documented | 7–16 | In high endemic area:<br>↓ Lymphocyte proliferation to SEA<br>= Lymphocyte proliferation to PHA<br><br>In low endemic area:<br>↑ Lymphocyte proliferation to SEA<br>= Lymphocyte proliferation to PHA | In high endemic area:<br>= Lymphocyte proliferation to SEA and PHA<br><br>In low endemic area:<br>= Lymphocyte proliferation to SEA and PHA | Response to SEA in individuals with *Sh* infection varied among different areas depending on *Sh* endemicity.<br><br>There was no change in lymphocyte proliferation after PZQ treatment. |
| Schmiedel et al 2015 (119) | Gabon | 38 | F (89%) | mean 10 | N/A | = Lymphocyte proliferation pre- and post-PZQ treatment<br><br>*6 weeks after treatment* | *Sh* infection had no impact on lymphocyte proliferation prior and after PZQ therapy. |

*Sh* = *S. haematobium*, *Sm* = *S. mansoni*, PHA = Phytohemagglutinin, PWM = pokeweed mitogen, PPD = purified protein derivate, AWA = adult worm antigen;

SEA = schistosomal egg antigen, SWAP = soluble antigen preparation of adult *schistosomes*

*Sh* infection. Further, a cohort study in Zimbabwe showed changes in lymphocyte proliferation in individuals with *Sh* infection depending on *Sh* endemicity levels in their area of residence. Those living in a low-endemic *Sh* area had higher lymphocyte proliferation in comparison to those without *Sh* infection from the same area, while the opposite was seen in participants living in a *Sh* high-endemic area. These proliferative responses did not change in either population 12 and 18 months after PZQ treatment and no difference was seen in response to generic antigen stimulation with PHA [88]. This study reflects the complexity of assessing immune responses from different populations living in different environments and again highlights the multiple factors that evidently influence an individual's immune response

**Table 6.** *S. haematobium* **infection and host T cell immune response in the systemic circulation in humans.**

| Study | Country | Sample size | Sex (female in %) | Age in years | Differences in individuals with *S. haematobium* infection compared to individuals without *S. haematobium* infection | Effect of praziquantel treatment | Overall conclusion |
|---|---|---|---|---|---|---|---|
| Appleby et al 2015 (136) | South Africa | 94 | F (56%) | 5–54 | In response to SWAP ↓ CD3ζ expression on T cells associated with increased Sh intensity<br><br>↑ CD3ζ expression associated with PBMC Proliferation | N/A | CD3ζ downregulation was associated with increased *Sh* infection intensity without limiting proliferative capacity of PBMCs in response to *Sh* antigen. |
| Caserta et al 2012 (134) | Zimbabwe | 105 | Not documented | 6–16 | ↑ CD200R⁺ CD4⁺ T cells (baseline and with infection intensity) | N/A | Inhibitory receptor CD200R epxression was increased in individuals with *Sh* infection and increased with infectious burden. |
| *DiNardo et al 2018 (128) | Swaziland | 38 | F (59%) | 6 mo– 15 years | In response to ESAT-6 and CFP-10 to elicit an MTB specific immune response:<br><br>↓ Th1:Th2 ratio<br><br>↑ IL-4 producing CD4⁺ T cells<br><br>↓ TNF-α and IFN-γ producing CD4⁺ T cells | In response to ESAT-6 and CFP-10 to elicit an MTB specific immune response:<br><br>↑ Th1:Th2 ratio<br><br>↓ IL-4 producing CD4⁺ T cells<br><br>↑ TNF-α and IFN-γ + producing CD4⁺ T cells<br><br>*6 months after treatment* | Individuals with *Sh* infection had decreased MTB specific Th1 CD4+ T cell function, with associated increases in Th2 cells. |
| Grogan et al 1996 (112) | Gabon | 110 | Male and female (%, not documented) | 5–48 | ↓ T cell proliferation to AWA ↓ IL-4 to AWA and SEA<br><br>= IL-5 and IFN-γ | ↑ T cell proliferation to AWA ↑ IL-4 to AWA and SEA<br><br>= IL-5 and IFN-γ<br><br>*5 weeks after treatment* | *Sh* infection led to low antigen specific T cell proliferation and lL-4 production. |
| Kleppa et al 2014 (107) | South Africa | 44 | F (100%) | 15–23 | Women with FGS vs without FGS (defined by visualization of sandy patches):<br><br>= CD4⁺ T cells<br><br>↑ Monocytes (CD14⁺)<br><br>↑ CCR5⁺ CD4⁺ T cells ↑ CCR5⁺ Monocytes (CD14⁺) | Women with FGS vs without FGS:<br>↓ CCR5⁺ CD4⁺ T cells ↓ CCR5⁺ Monocytes (CD14⁺)<br><br>*7–8 months after treatment* | HIV entry molecule CCR5 was increased on systemic CD4⁺ T cells and monocytes in women with sandy patches, and decreased post-PZQ. |
| Kleppa et al 2015 (124) | South Africa | 765 | F (100%) | mean 18.9 | = CD4⁺ T cells | N/A | *Sh* infection did not impact CD4⁺ T cell counts. |
| Kroidl et al 2019 (125) | Tanzania | 235 | F (42%) | 18–62 | = HLA-DR⁺ T cell | N/A | There were no differences in HLA-DR⁺ T cell frequencies. |

*(Continued)*

**Table 6.** (*Continued*)

| Study | Country | Sample size | Sex (female in %) | Age in years | Differences in individuals with *S. haematobium* infection compared to individuals without *S. haematobium* infection | Effect of praziquantel treatment | Overall conclusion |
|---|---|---|---|---|---|---|---|
| *Labuda et al 2020 (58) | Gabon | 79 | Male and female (%, not documented) | School-children | In response to AWA and SEA<br><br>= CD4$^+$ $T_{CM}$ cells, CD4$^+$ T $_{EMRA}$ cells, and CD4$^+$ $T_{EM}$ cells<br><br>↑ IFN-γ, IL-2, IL-5, IL-10, and TNF-α to AWA, SEA<br><br>= IFN-γ, IL-2, IL-5, IL-10, and TNF-α to PHA<br><br>↑ Tregs | In response to AWA and SEA<br><br>= CD4$^+$ $T_{CM}$ cells and CD4$^+$ T $_{EMRA}$ cells<br><br>↑ $T_{EM}$ cells<br>↑ IFN-γ, IL-2, IL-5, IL-10, and TNF-α to AWA, SEA<br>↑ IL-10 to PHA<br><br>*7 months after treatment*<br><br>↓ Tregs<br><br>*6 weeks and 7 months after treatment* | Tregs were increased in individuals with *Sh* infection. PZQ treatment led to decreased Tregs, increased effector T-cells and antigen-specific cytokine production in previously infected.<br><br>There was no difference in systemic effector CD4$^+$ T cell populations, but $T_{EM}$ cells increased after PZQ treatment. |
| Mbow et al 2013 (113) | Senegal | 26 | F (50%) | 5–14 | ↑ Th17 cells (with GU pathology)<br>↑ IL-17 producing CD4$^+$ T cells<br><br>↑ Ratio: Th17 to Tregs (with GU pathology)<br>↑ Ratio: IL17$^+$ to IL-10$^+$ CD4$^+$ T cells (with GU pathology)<br><br>= Tregs (CD4$^+$CD25$^{high}$Foxp3+ and CD4$^+$Foxp3$^+$)<br>= IL-10$^+$CD4$^+$ T cells<br>= CD4$^+$T-bet$^+$<br>= CD4$^+$IFN-γ$^+$<br>= CD4$^+$GATA3$^+$<br>= CD4$^+$IL-4$^+$ T cells | N/A | Immune dysregulation was more pronounced in individuals with *Sh* infection with GU pathology comapred to without.<br><br>Tregs tended to be lower in individuals with *Sh* infection with GU pathology and the same group had a higher ratio of Th17 cells to Tregs primarily due to the presence of elevated Th17 cells. |
| Nausch et al 2011 (117) | Zimbabwe | 49 | Male and female (%, not documented) | 8–60 | = CD25$^+$Foxp3$^-$ CD4$^+$ T cells<br><br>↑ Tregs (with infection intensity) in < 14yo<br><br>↓ Tregs (with infection intensity) in > 14yo | N/A | There were no differences in activated CD4$^+$ T cells.<br><br>In individuals with *Sh* infection < 14 years of age, Tregs positively correlated with infection intensity, while there was a negative correlation in >14 years of age. |

(*Continued*)

**Table 6.** (Continued)

| Study | Country | Sample size | Sex (female in %) | Age in years | Differences in individuals with *S. haematobium* infection compared to individuals without *S. haematobium* infection | Effect of praziquantel treatment | Overall conclusion |
|---|---|---|---|---|---|---|---|
| Nausch et al 2012 (103) | Zimbabwe | 105 | F (57%) | 6–84 | ↓ CD4$^+$ memory T cells and CD4$^+$ T$_{EM}$ cells<br><br>↓ CTLA-4$^+$ CD4$^+$ T cells<br><br>= CD4$^+$ and CD8$^+$ T cells<br>= CD4$^+$ T$_{CM}$ cells<br>= CD8$^+$ memory T cells<br>= CD31$^+$CD4$^+$ T cells<br><br>= IL-4 after WWH stimulus<br><br>↓ IFN-γ after WWH stimulus<br><br>No association between CD4$^+$ memory T cells and IFN-γ or HLA-DR CD4$^+$ T cells after PHA stimulation | ↓ CD4$^+$ T cells and CD4$^+$ T memory cells<br><br>= CD31$^+$CD4$^+$ T cells (trend)<br><br>↑ CD31$^+$CD4$^+$ T cell proliferation (indicated by shortened telomerase lengths)<br><br><br>*6 weeks after treatment* | Individuals with *Sh* infection had decreased CD4$^+$ memory, EM and CTLA-4$^+$ T cells as well as lower IFN-γ levels after *Sh* antigen stimulus.<br><br>CD4$^+$ memory and EM T cells remained low 6 weeks after PZQ therapy, although CD31$^+$ CD4$^+$ (naïve) memory T cell proliferation increased after treatment in previous individuals with *Sh* infection. |
| Nmorsi et al 2005 (123) | Nigeria | 250 | Male and female (%, not documented) | 5–60 | ↓ CD4<br><br>↑ CD8<br><br>↓ CD4:CD8 ratio<br><br>↑ CD4:CD8 ration with infection intensity | N/A | Individuals with *Sh* infection had lower CD4$^+$ T cells and CD4:CD8 T cell ratio compared to uninfected.<br><br>CD4:CD8 T cell ratio increased with *Sh* infection intensity. |
| Schmiedel et al 2015 (119) | Gabon | 38 | F (89%) | mean 10 | ↑ Tregs<br><br>↓ Th1 and Th2 cytokines to AWA, SEA and BCG<br><br>= PBMC proliferation | ↓ Tregs<br><br>↑ Th1 and Th2 cytokines to AWA, SEA and BCG<br><br>= PBMC proliferation<br><br>*6 weeks after treatment* | Tregs were upregulated in individuals with *Sh* infection and were associated with lower Th1 and Th2 cytokine production, which reversed post-PZQ. |
| Van der Kleij et al 2004 (62) | Gabon | 25 | F (40%) | 8–15 | = Monocytes, B cells, and T cells | N/A | *Sh* did not impact systemic T cell, B cell or monocyte frequencies. |

*Sh* = *S. haematobium*, SWAP = Soluble worm antigen preparation, CM = central memory, EM = effector memory, EMRA = terminally differentiated effector memory T cells, MTB = Mycobacterium tuberculosis, mo = months, PHA = Phytohemagglutinin, BCG: bacille Calmette-Guerin, yo = year old; WWH = whole worm homogenate
*Studies included participants with additional parasitic infections.

to *Sh* infection. These factors may include the individual's exposure history beyond parasitic infections which can impact their immune status and memory, as well as overall health, microbiome, sex, and age.

## *S. haematobium* infection and host T cell immune response in the systemic circulation in humans

T cells play a crucial role in the immune response against invading pathogens, including schistosome infections [28]. Both CD4$^+$ and CD8$^+$ T cells are critical in protecting the host against

**Table 7. *S. haematobium* infection and host B cell immune response in the systemic circulation in humans.**

| Study | Country | Sample size | Sex (female in %) | Age in years | Differences in individuals with *S. haematobium* infection compared to individuals without *S. haematobium* infection | Effect of praziquantel treatment | Overall conclusion |
|---|---|---|---|---|---|---|---|
| *Labuda et al 2013 (139) | Gabon | 56 | 43% F | 8–16 | ↑ MBC<br>↑ CD23 expression on B cells (after BCR stimulation)<br><br>↓ Ki-67$^+$ B cells, and naïve B cells<br>↓ MBC proliferation (after BCR stimulation)<br>↓ TNF-α MBCs (after BCR stimulation) | ↑ Naïve B cells<br><br>↓ MBC (exception: TNF-α producing MBCs)<br><br>↑ B cell responsiveness restored<br><br>*6 months after at least 3 rounds of treatment* | *Sh* infection led to an increase of MBC and B cell subsets which was at least partially resorted 6 months after PZQ treatment. |
| *Lyke et al 2012 (138) | Mali | 84 | 51% F | 4–14 | ↑ MBC (after SEA and SWAP stimulation) | N/A | Stimulation with *Sh* antigen led to an increase in MBC proportions in individuals with *Sh* infection. |
| Rujeni et al 2013 (140) | Zimbabwe | 434 | 60% F | mean 8.2–39 | ↑ Soluble CD23 with infection intensity | N/A | Soluble CD23 increased with infection intensity. |
| *Van der Kleij et al 2004 (62) | Gabon | 25 | F (40%) | 8–15 | = Monocytes, B cells, T cells | N/A | *Sh* infection did not impact systemic T cell, B cell or monocyte frequencies. |
| *van der Vlugt et al 2012 (141) | Gabon | 20 | 55% F | 8–14 | ↑ Breg subpopulation (CD1d$^{hi}$ B cells)<br><br>BCR stimulation:<br>↑ IL-10 production by B cells | BCR stimulation:<br>↓ IL-10 production by B cells<br><br>*6 months after at least 3 rounds of treatment* | *Sh* infection led to an expansion of a Breg subpopulation (CD1d$^{hi}$ B cells) and increase in IL-10 production by B cells.<br><br>IL-10 production decreased in B cells post-PZQ. |
| *van der Vlugt et al 2014 (142) | Gabon | 42 | 50% F | 16–28 | After stimulation with generic antigens:<br><br>↑ Breg subpopulations:<br>  CD1d$^{hi}$ B cells<br>  IL-10$^+$ B cells<br>  IL-10$^+$ CD1d$^{hi}$ B cells<br>  CD24$^{hi}$CD27$^+$ B cells<br>  LAB$^+$ B cells<br><br>Co-culture of healthy CD4$^+$ T cells with CD1d$^{hi}$ B cells of individuals with *Sh* infection subjects:<br><br>↓ IFN-γ production by CD4$^+$ T cells<br>↓ IL-4 production by CD4$^+$ T cells<br>↓ Il-17 production by CD4$^+$ T cells<br><br>↑ IL-10$^+$ T cells<br>↑ Tregs<br><br>→ Depletion of CD1d$^{hi}$ B cells decreased IL-10$^+$ T cells production of CD4$^+$ T cells | N/A | *Sh* infection led to an expansion of Breg subpopulations which supressed CD4$^+$ T cell function *in vitro* and increased regulatory T cell popualtions (Tregs and IL-10$^+$ T cells). |

*Sh* = *S. haematobium*, MBC = memory B cells, Bregs = B regulatory cells BCR = B cell receptor, PZQ = Praziquantel, SEA = Soluble egg antigen, SWAP = Soluble worm antigen

*Studies included participants with additional parasitic infections.

pathogens, with CD8$^+$ T cells being particularly important in fighting intracellular pathogens including viruses and bacteria, as well as in eliminating malignant cells in cancer [120].

Some effector CD4$^+$ and CD8$^+$ T cells will develop into memory T-cells, which after antigen encounter can rapidly expand upon re-encountering the pathogen. Memory T cells can be further subdivided into multiple subpopulations including central memory and effector memory T cells [121]. Central memory T cells reside mainly in lymphoid tissue and blood, while effector memory T cells are present in blood and traffic through peripheral organs [122].

   

CD4[+] T cells have a wide array of functions that rely on specialization through functional polarization. CD4[+] T cell subsets addressed in manuscripts included in this review include Th1, Th2, Th17 and Tregs. The characteristic immune alteration induced by helminth infections is an increase in Th2 cells and a decrease in Th1 cells [33]. Th17 cells provide protection and immunity against extracellular pathogens, especially at mucosal tissue, and play a role in the pathogenesis of multiple chronic inflammatory diseases [52]. Tregs, characterized by high expression of IL-2 receptor α chain, CD25, and Foxp3[+], are important in the maintenance of peripheral immune tolerance [36]. Each of these subsets will be discussed in detail in this section.

**CD4[+] and CD8[+] T cells.**   Overall frequencies of T cells largely do not appear to differ between individuals with and without *Sh* infection [62], though multiple studies have reported altered composition of T cell subsets. Analysis of T cell subsets in people with *Sh* infection compared to individuals without infection showed lower CD4[+] (257 versus 681) and higher CD8[+] (210 versus 114) T cells/μL in a Nigerian cohort without HIV infection [123]. Notably, no information was given on where the 44 negative control subjects were recruited from, which might shed light on the major differences in CD4[+] T cell numbers between the groups. Further, decreased CD4[+] to CD8[+] T cell ratios were associated with heavy *Sh* infection in children and adults (urine eggs >50/mL) compared to ratios in those with light infections. The authors reported that lowest ratios were found in infected adults, followed by children with *Sh* infection, as compared to the control group without infection, demonstrating an age dependent pattern in host immune response to *Sh* [123]. Contrastingly, Nausch et al [103] did not find a difference in circulating CD4[+] or CD8[+] T cell frequencies in persons with or without *Sh* infection in their HIV-uninfected Zimbabwean study population which at least is partially attributable to lower infection intensity in comparison to the Nigerian cohort. A South African study also did not find a difference in mean CD4 cell count in women with or without *Sh* infection or clinical evidence of FGS, regardless of HIV status [124].

In a cohort of 235 men and women who did not have HIV infection in Tanzania, 82.1% were helminth infected and only 9.3% were individuals with *Sh* infection [125]. No difference was seen in HLA-DR[+] T cell frequencies comparing individuals without *Sh* infection, which included participants with and without other helminth infections, to individuals with *Sh* infection. Notably, the small number of individuals with *Sh* infection in this cohort warrants caution when interpreting these results.

**Memory T cells.**   Observed associations of *Sh* infection with CD4[+] and CD8[+] T cell memory subsets vary. In Gabonese schoolchildren, the frequency of CD4[+] effector memory T cells increased in children with *Sh* infection post-PZQ treatment, but there was no difference at baseline between CD4[+] effector or memory T cells [58]. In contrast, Nausch et al found that children and adults with *Sh* infection at baseline had lower CD4[+] memory and effector memory T cells than uninfected [126]. Stimulation of CD4[+] memory T cells with generic antigen led to increased activation, assessed by HLA-DR[+] expression, and increased IFN-γ secretion in individuals without but not in individuals with *Sh* infection, highlighting possible functional impairment of CD4[+] memory T cells in the setting of *Sh* infection. Neither group produced Th2 cytokines (IL-4, IL-5) in response to stimulation [126]. In the same study, PZQ treatment of those with *Sh* infection led to decreased CD4[+] T cells and CD4[+] memory T cells, increased proliferation of naïve thymic CD31[+]CD4[+] T cells, and no changes in CD8[+] T cell and CD8[+] memory T cell frequencies at 6 weeks [126]. From these studies together, it appears that CD4[+] memory T cells are an important source of IFN-γ production that are lower in persons with *Sh* infection and may further decrease at 6 weeks after treatment [126]. Follow-up beyond 6 weeks was not performed, rendering longer-term effects of PZQ on CD4[+] memory T cells unknown.

**Th1 and Th2 cells.**   Two studies examined Th1 and Th2 CD4$^+$ T cell populations in *Sh* infection. PBMCs in response to stimulation with AWA or SEA of individuals with *Sh* infection in Gabon had lower T cell proliferation and lower IL-4 production, both of which were reversed after PZQ treatment [127]. Notably, IL-4 production was abolished by CD4$^+$ T cell depletion, but not by CD8$^+$ T cell depletion [127].

Further supporting these findings, PBMCs of children with *Sh* infection from Swaziland stimulated with *Mycobacterium tuberculosis* (MTB) antigens developed a lower Th1:Th2 cytokine ratio, determined by the ratio of IFN-γ and TNF-α to IL-4, than children without *Sh* infection, and their CD4$^+$ T cells produced greater MTB specific IL-4, with a decrease in MTB specific IFN-γ and TNF-α [128]. Six months after chemotherapy, the Th1:Th2 ratio increased and MTB specific IFN-γ and TNF-α producing CD4$^+$ T cells were detected, but all of these remained lower than in children without *Sh* infection [128]. Overall, this study suggested an *Sh*-associated decrease in MTB specific Th1 CD4$^+$ T cell function, with associated increases in Th2 cells, raising concern for a compromised immune response to MTB in individuals with *Sh* infection which may not be entirely reversed after treatment.

**Th17 cells.**   Only one study to date reported on Th17 cells in the setting of *Sh* infection. In Senegalese children, Th17 cells (CD3$^+$CD4$^+$RORγt$^+$) were elevated in individuals with *Sh* infection with sonographic evidence of genitourinary pathology than individuals with *Sh* infection without pathology and matched individuals without *Sh* infection [113]. Although IL-17 producing CD4$^+$ T cells tended to be higher overall in individuals with versus without *Sh* infection, the greatest differences was noted between individuals with *Sh* infection with genitourinary pathology compared to those with *Sh* infection but without pathology [113]. Investigators also reported that both individuals with and without *Sh* infection with genitourinary pathology had higher ratios of IL17$^+$ to IL-10$^+$ CD4$^+$ T cells compared to those individuals with *Sh* infection without genitourinary pathology. These cytokine findings accord with their finding that participants with pathology had a higher ratio of Th17 cells to Tregs, primarily due to the presence of higher levels of Th17 cells [113]. Though a single study, this work importantly highlights the potential role of Th17 cells in *Sh*-induced tissue pathology, a role that has been suggested in *S. mansoni* and *S. japonicu*m infection as well [129–131].

**T regulatory cells.**   A total of 4 studies investigated Treg populations in *Sh* infection. Most studies (3 out of 4) demonstrated an increase in systemic Tregs individuals with *Sh* infection compared to uninfected [58,119,132], and both studies that assessed the effect of PZQ treatment reported that Tregs decreased after anti-helminthic treatment [58,119].

Mbow et al, described above, found no differences in Tregs between those with *Sh* and genitourinary pathology, *Sh* without pathology, and without *Sh* infection [113].

In addition to finding that Tregs normalized after PZQ treatment in Gabonese children infected with *Sh*, Schmiedel and colleagues also reported that both Th1 (IL-17, IFN-γ, TNF) and Th2 (IL-5, -10, -13) cytokine responses to schistosome and MTB antigen stimulation increased post-treatment [119]. Furthermore, *ex vivo* depletion of Tregs led to an increase in both Th1 and Th2 cytokine responses to both schistosome and MTB antigen stimulations in samples of individuals with *Sh* infection before treatment and after treatment. This demonstrates the persistent suppressive capacity of Tregs, even following a decrease in their numbers post-PZQ [119].

Of note, Nausch et al found that Tregs increased with infection intensity in those below 14 years of age, but the opposite was seen in individuals above age 14 with decreasing Tregs in those with higher *Sh* burden [132]. Overall children with *Sh* infection (< 14 years of age) had higher Treg proportions than children without infection, while Treg proportions were lower in the people with versus without infection older than 14 years. Notably, in these older patients,

the burden of *Sh* was lower, possibly reflecting the development of acquired anti-schistosome immunity with age that could contribute to a diminished regulatory response [132].

Together, these studies suggest that Tregs play a major and potentially age-dependent role in the immune response to *Sh* infection. During youth, when individuals have less acquired immunity, higher Treg proportions may be crucial in dampening excessive inflammation and downstream pathology. As host protective acquired immunity develops with age, the necessity for an enhanced regulatory phenotype that was seen in children may decrease.

**T cell marker expression.** A total of four studies examined T cell surface markers in *Sh* infection. CD4$^+$ T cells that expressed the inhibitory protein Cytotoxic T-lymphocyte-associated protein 4 (CTLA-4) were decreased in individuals with *Sh* infection compared to persons without infection. Notably, CTLA-4$^+$CD4$^+$ T cells increased with age in those individuals with *Sh* infection but did not change with age in those without *Sh* infection [126].

In a study in Zimbabwe, expression of the regulatory molecule CD200 receptor (CD200R) [133], was higher on CD4$^+$ T cells of persons with *Sh* infection. Further, CD200R-expressing cells appeared to belong predominantly to the Th2 lineage, with IL-4 secreting CD4$^+$ T cells almost all expressing CD200R while IFN-γ secreting CD4$^+$ T cells expressed significantly less CD200R [134]. CD200R expression on CD4$^+$ T cells also correlated positively with infection intensity [134]. These studies indicating increased expression of immune regulatory proteins CTLA-4 and CD200R suggest a state of immune suppression, potentially influencing the effectiveness of the immune response to other pathogens, particularly viral.

A study in South Africa reported a negative correlation between CD3ζ expression on T cells, which promotes downstream immune responses including cell proliferation [135], and infection intensity, while noting a positive correlation between CD3ζ expression and PBMC proliferation in response to schistosome antigen stimulation with SWAP [136]. The authors hypothesized that CD3ζ downregulation could be a mechanism for immune dysregulation that is observed during chronic *Sh* infection in humans without changes in the absolute numbers of T cells. It is thought that immune dysregulation in addition to local mucosal damage may contribute to the increased risk of HIV acquisition seen in some studies of women with *Sh* infection [8,12,137]. In accordance with this, women with FGS (defined as presence of sandy patches on colposcopy exam) were found to have increased expression of the HIV entry molecule CCR5 on both systemic circulating and local cervical CD4$^+$ T cells and CD14$^+$ monocytes, which decreased on both cell populations post-PZQ treatment in both compartments [107].

**Overall synthesis of T cell studies in Sh infection.** Together, these 14 studies indicate that systemic T cells play a crucial role in immunity against *Sh* infections. It appears that *Sh* infection leads to increases in T cell populations which augment suppressive and regenerative functions on the immune system, particularly upregulating Th2 immune responses, Tregs, and Th17 cells. These increases appear to be accompanied by decreases in CD4$^+$ memory and effector, Th1, and IFN-γ producing T cell populations. These systemic immune changes may further explain why individuals with *Sh* infection may be more susceptible to a variety of different pathogens [29].

## *S. haematobium* infection and host B cell immune response in the systemic circulation in humans

In contrast to the abundance of data examining T cells in *Sh* infection, only 6 studies have investigated B cells. One study showed no difference in B cell frequencies between individuals with and without *Sh* infection [62]. Two others reported that activated B cells, memory B cells, and Bregs are higher during *Sh* infection. *Sh* infection has been associated with an increase in

memory B cells in response to schistosome antigens SWAP and SEA in a Malian cohort [138] and at baseline in a Gabonese cohort [139] compared to the uninfected individuals in each cohort. At a functional level, B cells were shown to have increased expression of the activation marker CD23, similar to the rising soluble CD23 levels with *Sh* infection intensity described by Rujeni et al [140]. B cells from children with *Sh* infection further had lower proliferation and intracellular B cell TNF-α production compared to children without *Sh* infection [139]. Notably, PZQ treatment led to an increase in naive B cells, decrease in memory B cells apart from TNF-α producing memory B cells, and restoration of B cell responsiveness [139].

In a different study in Gabon, children with *Sh* infection had elevated CD1dhi B cells, a Breg subpopulation, compared to children without *Sh* infection, with elevated IL-10 production by B cells after stimulation, which was reduced post-PZQ treatment [141]. Stimulation of B cells with antigens PMA, ionomycin, brefeldin A (PIB) and LPS from individuals with *Sh* infection led to induction of multiple different Breg subpopulations compared to individuals without *Sh* infection. Co-culture of CD1hi Bregs from persons with *Sh* infection together with CD4$^+$ T cells led to a decrease in IFN-γ, IL-4 and IL-17 production by CD4$^+$ T cells, and increased frequencies of Tregs and IL-10$^+$ T cells [142]. Depletion of CD1hi Bregs in the CD4$^+$ T cell co-culture restored IL-10$^+$ T cell numbers, while Tregs remained the same [142].

Together, these data suggest that *Sh* infection induces changes in the B cell compartment including an increase in Bregs, which may play a key role in inducing T cell hyporesponsiveness and altering memory B cell subsets. Bregs have been shown to suppress Th1 immune response [143] and suppress viral specific CD8$^+$ T cell responses in individuals with HIV, hepatitis B, and hepatitis C [144,145]. Increases in Bregs and alterations in memory B cells subsets might offer at least one mechanism by which *Sh* infection is associated with increased acquisition of certain viral infections and worse outcomes in individuals with *Sh* and viral co-infections.

## *S. haematobium* infection and the host local genitourinary tissue immune response in humans

Six studies have examined local mucosal immune responses in *Sh* infection in humans (Table 8).

Schistosome eggs are highly immunogenic and cause a strong local immune response [94]. Eggs migrating through tissue can become entrapped in organs of the genitourinary system, including not only the bladder but also reproductive organs, causing female and male genital schistosomiasis (FGS, MGS) [35,146] with symptoms dependent on egg location [147]. In MGS eggs become entrapped in male genital organs including seminal vesicles, prostate gland, and vas deferens causing local inflammatory responses with subsequent tissue damage and scarring [146]. Similarly, in FGS eggs become entrapped in female reproductive organs, including the cervix and vagina, leading to tissue damage which can cause a wide array of symptoms including discharge, pain, and infertility [35].

A recent study in Zambia by Sturt et al [148] evaluated cervical immune changes in women with definite FGS (positive vaginal or cervical *Sh* PCR), probable FGS (urinary *Sh* eggs or CAA with one of four typical cervical findings), and those without FGS (absence of these findings) [148]. In the primary analysis, cervical lavage (CVL) specimens of women with definite FGS demonstrated elevated IL-5 compared to women without FGS, without differences in other genital cytokines. In exploratory analyses, women with high *Sh* FGS burden, defined as two or more positive genital *Sh* PCRs, had higher concentrations of Th2 cytokines IL-4, IL-5, and IL-13 as well as inflammatory cytokines IL-1 α and IL-15 compared to women without FGS [148]. In a second exploratory analysis, after adjustment for multiple comparisons, cervical

**Table 8. *S. haematobium* infection and the local tissue host immune response in humans.**

| Study | Country | Sample size | Sex (female in %) | Age in years | Compartment | Differences in individuals with *S. haematobium* infection compared to individuals without *S. haematobium* infection | Overall conclusion |
|---|---|---|---|---|---|---|---|
| Dupnik et al 2019 (149) | Tanzania | 57 | F (100%) | 20–35 | Genital compartment (CVL) | ↓ IL-15 | Women with *Sh* infection had lower IL-15 concentrations in CVL samples. |
| Jourdan et al 2011 (156) | Malawi | 61 | F (100%) | 15–45 | Genital compartment (cervical tissue biopsies) | Women with *Sh* ova in cervical biopsies vs no ova (Malawian and Norwegian controls): ↑ CD4$^+$ T cells<br><br>= Langerhans cells and CD8$^+$ T cells<br><br>Women with *Sh* ova in cervical biopsies with viable ova vs calcified ova: ↑ CD68$^+$ macrophages in tissue with viable ova vs calcified ova | *Sh* ova in cervical tissue were associated with a higher CD4$^+$ T cell response.<br><br>Only viable *Sh* ova in cervical tissue, not calcified ova, led to an influx of CD68 + macrophages. |
| Kleppa et al 2014 (107) | South Africa | 44 | F (100%) | 15–23 | Genital compartment (Cytobrush samples) | Women with versus without genital sandy patches: = Monocytes (CD3⁻D56⁻CD14$^+$) = CCR5$^+$ Monocytes = CD4$^+$ T cells = CCR5$^+$CD4$^+$ T cells<br><br>Praziquantel treatment of *Sh* infection led to: ↓ CCR5$^+$ Monocytes ↓ CCR5$^+$ CD4$^+$ T cells | Women with sandy patches did not have altered cervical CCR5 expression on monocytes or CD4+ T cells, but expression decreased after PZQ treatment. |
| Leutscher et al 2005 (157) | Madagascar | 240 | F (0%) | 15–49 | Genital compartment (Seminal) | ↑ Leukocytospermia<br><br>Moderate to high egg burden vs uninfected: ↑ Eosinophils ↑ IL-10 | Men with *Sh* infection had increased leukocytospermia.<br><br>Other seminal changes including increased eosinophils and IL-10 were only observed in men with high seminal *Sh* egg burden. |
| Njaanake et al 2014 (67) | Kenya | 158 | F (55%) | 5–12 | Genitourinary compartment (Urine) | High intensity *Sh* infection vs low infection intensity: ↑ IL-6<br><br>↓ IL-10<br><br>Individuals with *Sh* infection with urinary tract pathology seen on US versus no pathology seen: ↑ IL-6<br><br>↓ TNF-α<br><br>Hematuria associated with: ↑ IL-6<br><br>↓ IL-10 | Elevation of pro-inflammatory cytokine IL-6 and lower levels of Th-2 associated cytokine IL-10 were associated with *Sh* egg burden and GU pathology in *Sh* infected children. |

*(Continued)*

**Table 8.** (Continued)

| Study | Country | Sample size | Sex (female in %) | Age in years | Compartment | Differences in individuals with *S. haematobium* infection compared to individuals without *S. haematobium* infection | Overall conclusion |
|---|---|---|---|---|---|---|---|
| Sturt et al 2021 (148) | Zambia | 212 | F (100%) | 18–31 | Genital compartment (CVL) | Women with FGS vs without FGS: ↑ IL-5 = IL-1β, IL-6, IFN-γ, IL-8, MIP-1 α /b, MCP-1, IP-10, and eotaxin<br><br>Women with FGS and probable FGS vs FGS uninfected: ↑ TNF-α<br><br>High FGS burden vs without FGS: ↑ IL-4, IL-5, IL-13, IL-1α, and IL-15<br><br>Elevated *Sh* DNA genital concentration vs without FGS: ↑ IL-4, IL-5, IL-13, IL-1, IL-15, and TNF-α | FGS altered the local cervical mucosal immune environment, increasing Th2 associated and inflammatory cytokines. |

*Sh* = *S. haematobium*, FGS = Female genital schistosomiasis, CVL = cervical vaginal lavage, US = ultrasound

Th2 cytokines IL-5 and pro-inflammatory cytokine TNF-α were associated with FGS. Of note, a study in Tanzania found decreased IL-15 levels in CVL samples of women infected with *Sh* as diagnosed by presence of elevated CAA plus *Sh* eggs in urine, but no difference in any of the other 26 measured cytokines [149].

The diverse results of local cervical cytokine expressions might reflect the different study designs among women with *Sh* infection. While the Tanzania study evaluated cervical cytokines in all women with *Sh* infection (therefore possibly including some women who had parasitic infection but without genital tract involvement), the Zambia study's strict requirement of PCR-confirmed *Sh* infection in the genital tract likely restricted that study population to a younger age group and/or only those with the greatest genital tract abnormalities. It is also possible that the presence of FGS plus STIs, particularly trichomoniasis, which tended to be higher in women with *Sh* infection (p = 0.08), could explain why women with FGS in Zambia had both type 2 and pro-inflammatory immune responses since trichomoniasis can elicit a pro-inflammatory response [150]. Furthermore, analysis of the local mucosal cervical immune phenotype is challenging since there are numerous additional potential confounders, including physiologic changes occurring throughout the menstrual cycle [78,151,152], effect of recent pregnancy [153] or current breastfeeding [79], recent sexual contact, and vaginal cleansing practices [154,155], which were not consistently assessed in the above studies. Nevertheless, studies investigating local mucosal changes are critically important to enhance our understanding of the local cervical immune changes occurring during *Sh* and why women with FGS or *Sh* infections may have impaired ability to control genital viral infections such as HIV and HPV [6,11].

In a study of blood and cervical samples of South African women, monocytes were increased in the systemic circulation of women with FGS defined as sandy patches seen on the cervix [107]. While no difference was found in cervical CD4+ T cells or monocytes or their

CCR5 expression of the same cohort of women between women with and without FGS, PZQ treatment led to a significant decrease in CD14[+] monocytes in both blood and cervical samples. PZQ treatment also led to a decrease in CCR5 expression on CD4[+] T cells in blood and cervical samples [107]. The findings of this study provide evidence in favor of the hypothesis that FGS may raise the risk of HIV acquisition, potentially by altering local mucosal cell populations to increase HIV target cell populations.

Jourdan et al [156] performed cervical biopsies to evaluate HIV target cell populations in the female genital tract of Malawian women with *Sh* infection who had *Sh* eggs in urine [156]. This study found that the density of CD4[+] T cells was higher in cervical biopsies containing viable or calcified *Sh* ova, compared to cervical biopsies without ova from both endemic and non-endemic Norwegian controls. No difference was observed in the density of CD4[+] T cells in tissue comparing viable ova (N = 2) and calcified ova (N = 11), or in Malawian women without presence of *Sh* ova on biopsies. Importantly, there was no difference in densities of CD4[+] T cells in tissue with calcified ova and in biopsies from Norwegian women with chronic cervicitis. The density of CD68[+] macrophages was higher in specimens with viable ova than specimens with calcified ova, specimens from Malawian women without ova, and specimens from Norwegian women with healthy cervical tissue [156].

Relatively complementary findings have been reported from one study in men. Men with *Sh* infection, determined by presence of *Sh* eggs in urine or semen, were found to have increased leukocytospermia compared to men without *Sh* infection, which was more pronounced in men with moderate to high egg burden. Similarly, increased seminal eosinophils and IL-10 were seen in men with high egg burden when compared to men without *Sh* infection [157]. Even though only 5 of the 240 individuals in this study had detectable IL-4 in semen, all had *Sh* infection with high burden of infection, as measured by concentration of seminal eggs. In the same study, younger men had higher egg secretion in both urine and semen compared to men above age 39 [157], possibly reflecting partial immunity to the helminth with increasing age.

Similarly, *Sh* disease burden was reported to impact urinary cytokine expression in children with higher *Sh* urinary egg burden, who had higher urinary levels of IL-6 and lower IL-10, compared to children with lower disease burden [67]. Both these cytokine changes were also associated with hematuria and were not seen in peripheral blood. Furthermore, when the same children were assessed by ultrasound for possible genitourinary pathology, those found to have genitourinary pathology had increased geometric mean intensity (GMI) of IL-6 and lower GMI of TNF-α [67]. This study by Njaanake et al complements genital tract findings and demonstrates that the burden of disease, measured here by concentration of urinary eggs, appears to be a key mediator of extent of tissue pathology and the local mucosal immune milieu.

**Overall synthesis of immune studies in *Sh* infection on local tissue.** Collectively, the above studies demonstrate that chronic *Sh* infection with ova in genitourinary tissues leads to a local inflammatory reaction consisting of Th2 associated cytokines, pro-inflammatory cytokines, and an increased cellular immune response to viable ova comprised of CD4[+] T cells and CD68[+] macrophages. The intensity of *Sh* infection appears to be important, with a higher burden of disease triggering a stronger local immune response. Furthermore, individuals with local *Sh* mediated damage, such as hematuria or pathology seen on genitourinary ultrasound or cervical exam, appear to have higher rates of immune changes compared to individuals without such tissue damage.

**Table 9.** *S. haematobium* infection and the local tissue immune response in bladder cancer patients.

| Study | Country | Sample size | Sex (female in %) | Age in years | Compartment | Differences in individuals with *S. haematobium* infection compared to individuals without *S. haematobium* infection | Overall conclusion |
|---|---|---|---|---|---|---|---|
| El-Aal et al 2015 (161) | Egypt | 46 | F (17%) in *Sh* group | 38–84 | bladder | Compared to *Sh* negative patients with bladder cancer:<br>↑ GATA3<br>↑ Foxp3<br><br>Compared to *Sh* negative patients with bladder cancer and healthy controls:<br>↓ STAT4<br>↓ CD8$^+$ T cells | *Sh*-associated bladder cancer was associated with an increase in Th2 and Treg immune phenotype, while Th1 phenotype decreased. |
| El-Salahy et al 2002 (163) | Egypt | 70 | Not documented | Not documented | bladder | In *Sh*-associated bladder cancer or cystitis:<br>= IL-6 | IL-6 was not associated with *Sh*-bladder cancer or cystitis compared to *Sh*-uninfected with bladder cancer or cystitis. |
| Mohammed et al 2023 (162) | Egypt | 20–35 (15 with *Sh* vs 20 healthy PBMCs and 5 healthy bladder tissues) | F (~8%) | 35–73 | bladder<br><br>Blood | Compared to patients with *Sh* negative bladder cancer:<br>↓ CD3$^+$ T cells<br>↓ CD8$^+$ T cells<br><br>Compared to healthy controls:<br>↑ CD4$^+$ T cells<br>↑ Tregs<br>↑ Bregs<br><br>= B cells (overall)<br><br>↓ CD45$^+$ cells<br>↓ CD8$^+$ T cells | *Sh*-associated bladder cancer was associated with a shift towards predominant Th2 versus Th1 immune phenotype and an increase in regulatory immune cell. |

*Sh* = *S. haematobium*, T regulatory cells = Tregs, B regulatory cells = Bregs

### *S. haematobium* infection and the local tissue host immune response in patients with bladder cancer

Three studies focused on tissue immune responses in people with bladder cancer and *Sh* infection (Table 9).

There are many studies investigating the local gene changes that occur in patients suffering from *Sh*-associated bladder cancer [158], but only three analyzing changes in the immune phenotype.

In one study, immunohistochemical analysis of bladder biopsies showed increased numbers of cells expressing: (1) signal transducer and activator of transcription 4 (STAT4), which plays a key role in the differentiation of Th1 cells [159], and (2) Foxp3, a marker for Tregs, in patients with *Sh*-associated bladder cancer, compared to patients with bladder cancer without *Sh* infection. Cancer patients with *Sh* infection also had fewer cells expressing GATA3, which is important to generate an optimal Th2 immune response [160]. Two other studies similarly

reported that *Sh*-associated bladder cancer was associated with fewer CD8$^+$ T cells on bladder biopsy specimens, compared to patients with bladder cancer without *Sh* infection [161,162].

PBMCs from individuals with *Sh*-associated bladder cancer had lower CD3$^+$ T cells compared to *Sh*-unassociated bladder cancer, while CD4$^+$ T cells, Tregs, and Bregs were higher in *Sh*-unassociated bladder cancer compared to healthy controls who did not have *Sh* infection or bladder cancer [162]. Moreover, investigators reported fewer total CD3$^+$ T cells and T helper cells in these individuals and further demonstrated that individuals with *Sh*-associated bladder cancer had higher bladder tissue levels of IL-1β, IL-6, and TNF-α, which correlated positively with egg burden, compared to individuals with bladder cancer but without *Sh* infection [162]. In contrast to this, El-Salahy et al did not find a difference in IL-6 between bladder specimens of patients with bladder cancer or chronic cystitis with or without *Sh* infection, although they did not report the findings in *Sh*-associated bladder cancer versus cystitis separately, which could obscure potential differences between the two groups [163].

**Overall synthesis of *Sh* infection and the local tissue host immune response in patients with bladder cancer.** The finding that two of these three studies that individuals with *Sh* infection with bladder cancer have a shift towards a Th2 predominant phenotype, even compared to individuals with bladder cancer without *Sh* infection, suggests a possible role for *Sh*-induced immune alterations the carcinogenesis of *Sh*. Parasite-induced immune dysregulation that leads to a compromise in Th1 immunity might enable malignancy to thrive more easily, and explain at least partially why *Sh* infection is associated with bladder cancer, since a Th1 immune response is required for cancer control [164].

## Experimental models to study effects of *Sh* infection on immune response

Eight studies have assessed host immune response to *Sh* using experimental *ex vivo* organ culture and animal models (Table 10).

Animal models that recapitulate natural *Sh* egg deposition during infection must be achieved by injection of eggs and/or cercariae into tissue, which is at least a partial limitation, given that this technique disrupts the normal epithelium in a way that does not occur during natural human infection. To date, no animal model exists that mimics the human cycle of *Sh* infection, which is likely due to its specificity to humans, unlike *S. mansoni* which can infect rodents to provide a useful animal model [165]. *Sh* infection models in mice have shown that the response to *Sh* is dynamic and involves multiple different immune cell populations as well as a broad range of cytokines and chemokines. All support the well-established type 2 hypothesis in helminth infections, marked by an increase in Th2 type cytokines such as IL-4, IL-5, and IL-13 in mucosal tissue of the bladder in response to *Sh* eggs [166–168] as well as systemically and in lymphoid tissue [166,168,169].

IFN-γ levels appear to vary depending on tissue type and time since infection. Specific differences between murine models of *Sh* likely affect the observed immunological consequences. For example, introduction of *Sh* cercariae via intraabdominal injection only leads to egg disposition in the liver and intestines, as opposed to human infections where egg disposition is mainly in the genitourinary system [169]. Studies have reported an elevation of IFN-γ in lymphoid tissue [169] in the first weeks after intraabdominal cercarial injection, followed by a decline, and an increase in systemic IFN-γ [168] when *Sh* eggs were directly injected into the bladder wall in combination with intraabdominal cercarial injection. In contrast, Fu et al found decreased systemic IFN-γ levels after *Sh* bladder egg injection alone without additional cercarial infection [170]. Interestingly, leptin was increased in the local bladder tissue in mice injected with both *Sh* eggs and cercariae, which has immunomodulatory properties to shift the

**Table 10. Experimental models to study effects of *S. haematobium* infection on immune response.**

| Study | Study design | Compartment | Differences in *S. haematobium* infected compared to uninfected | Overall conclusion |
|---|---|---|---|---|
| Fu et al 2012 (166) | Mice were injected with *Sh* eggs or control vehicle directly into the bladder wall. | Bladder Serum | ↑ Eotaxin, IL-4, IL-13, TNF-α, KC, MIP-1a, and TGF-β<br>↑ Eosinophils, neutrophils, macrophages and B cells<br><br>= IFN-γ and IL-17<br><br>↑ IL-1a, IL-5, VEGF, and IL-17 | Th2 cytokine responses were increased in *Sh*-infected mice, particularly IL-4. |
| Fu et al 2015 (170) | Mice were injected with *Sh* eggs or control vehicle directly into the bladder wall, with or without experimental macrophage depletion. | Bladder Serum | Macrophage depletion:<br><br>↓ AAMs, macrophages, eosinophils, T- and B cells<br>↓ Eotaxin<br><br>↑ IL-1α, IL-3, MCP-1, MCP-3, and MIP-1α<br><br>↓ IL-1α, IL-3, IL-4, IL-13, IL-17, IL-23, IFN-γ, Eotaxin, RANTES, MIP-1a, and VEGF | Macrophage depletion in *Sh*-infected mice led to multiple downstream effects including change in cellular and cytokine response. |
| He et al 2002 (176) | Organ cultures of healthy human foreskin, obtained by circumcisions, were infected by direct intradermal injection with *Sh* cercariae. | Foreskin | ↑ IL-1α and IL-10 mRNA<br>↑ IL-1β, IL-2, IL-10, and TNF-α (macro gene array analysis)<br><br>= IL-1α (macro gene array analysis)<br><br>↓ IL-6, IL-8, IL-15, and IL-18 | Cercariae induced a local inflammatory immune response in the infected skin tissue *ex vivo*. |
| Hsieh et al 2014 (167) | Mice with induced E. coli UTI were either injected with *Sh* eggs or control vehicle directly into the bladder wall. | Bladder | Mice with E. coli UTI with *Sh* infection versus without:<br>↑ Macrophages, B cells, and neutrophils<br>↑ IL-4<br>↑ NKT cells specific IL-4 production<br><br>↓ T cells (trend)<br>↓ NKT cells<br>↓ NKT cell specific IFN-γ production<br>↓ CD1d expression on macrophages<br>↓ CD1d expression on dendritic cells | *Sh*-infected mice had an elevated Th2 cytokine response, particularly IL-4.<br><br>IFN-γ, T cells and NKT cells were reduced in *Sh*-infected mice. |
| Lane et al 1998 (169) | Mice were infected with *Sh* cercariae via abdominal injections and assessed cytokine response throughout 30 weeks post infection. | Spleen, Mesenteric LNs | At 14–20 weeks post *Sh* infection in response to SWAP and SEA stimulation:<br>↑ IFN-γ, IL-10, and IL-4<br><br>After week 20 post *Sh* infection in response to SWAP and SEA stimulation:<br>↓ IFN-γ, IL-10, and IL-4 | *Sh*-infected mice exhibited a dynamic Th1 and Th2 cytokine response with changing levels over time. |

*(Continued)*

**Table 10.**  (Continued)

| Study | Study design | Compartment | Differences in *S. haematobium* infected compared to uninfected | Overall conclusion |
|---|---|---|---|---|
| Loc et al 2019 (168) | Mice were infected with or without injected *Sh* cercariae (systemic) and with *Sh* eggs or control vehicle directly into the bladder wall (local). | Bladder Serum | In the combined infection model versus *Sh*-uninfected:<br><br>↑ Eotaxin, IFN-γ, IL-4, IL-13, LIF, M-CSF, RANTES, leptin, IL-1α, and TGF-β<br><br>↑ leptin<br><br>↑ IFN-α, IFN-γ, IL1A, IL-5, IL-13, IL-15, IL-18, IL-23, IP10, M-CSF, and TGF-β | Systemic and local *Sh* infection in mice appeared to play a role in the local and systemic immune modulatory effects. |
| Nair et al 2011 (174) | WT or Mta1 KO mice were infected with *Sh* cercariae via direct percutaneous inoculation. | Liver | *Sh*-infected Mta1 KO mice versus WT:<br><br>2 and 12 weeks after *Sh* infection:<br>↑ mRNA of IFN-γ, IL-2, IL-12, IL-10, and IL-6<br><br>↓ mRNA of CD4 T cells (trend)<br>8 weeks after *Sh* infection:<br>↑ IL-12p70 and IL-10<br><br>12 weeks after *Sh* infection:<br>↑ IFN-γ, IL-2, IL-12p70, IL-4, and IL-5<br>↑ mRNA of IFN-γ, IL-2, IL-12, IL-10, and IL-6<br><br>↓ TNF-α<br>↓ Eosinophils | Mta1 appeared to play an important role in the inflammatory responses to *Sh* egg induced hepatic granulomata in *Sh*-infected mice. |
| Richardson et al 2014 (172) | Mice were injected with *Sh* eggs or control vehicle directly into the vaginal wall. | Vaginal | In the first 2 weeks after infection:<br><br>↑ T cells and Macrophages<br>↑ RANTES<br><br>2 weeks after infection:<br><br>↑ T cells<br>↑ RANTES<br><br>4 weeks after infection:<br><br>↑ Macrophages<br>↑ CXCR4 and CCR5 expressing CD4 T cells<br>↑ CXCR4 expressing macrophages | *Sh* infection in mice led to increased expression of HIV entry molecules (CXCR4, CCR5) on local mucosal HIV target cells. |

*Sh* = *S. haematobium*, VEGF = Vascular endothelial growth factor, AAM = Alternatively activated macrophages, KO = knockout, WT = Wildtype

immune response towards a Th1 phenotype, which is observed during the first weeks of infection before cercariae have developed into adult worms [168,171].

Local *Sh* egg injection into the bladder or vaginal walls of mice leads to a change in local immune cell populations, with subsequent rises in macrophages, B cells, neutrophils, and

eosinophils [166,167,172]. Accompanying these rises, overall local tissue reductions of T cells as well as NKT cells have been reported, with NKT cell specific IFN-γ production and decreased expression of CD1d on antigen presenting cells [167]. These findings align with the decrease in IFN-γ levels seen by Fu et al [170] after direct injections of *Sh* eggs into the bladder wall.

In this experimental mouse model, Fu et al used intraperitoneal injections to cause a local depletion of macrophages and thereby demonstrated their profound role in the immune response to *Sh* eggs [170]. After macrophage depletion, mice that underwent direct injection of *Sh* eggs into the bladder wall developed disrupted cellular and tissue architecture of the resulting granulomata and decreased percentages of macrophages, AAMs, eosinophils, B cells, and T cells overall compared to mice whose macrophages were not disrupted. These mice also developed lower bladder tissue levels of eotaxin, and higher levels of IL-1a, IL-3, MCP-1, MCP-3, and MIP-1a. Furthermore, macrophage depletion led to decreases in systemic levels of numerous cytokines, including cytokines associated with leukocyte recruitment (RANTES, MIP-1a, and eotaxin), Th1 (IFN-γ), Th2 (IL-4 and IL-13), and Th17 (IL-17 and IL-23) T cell responses, and vascular endothelial growth factor (VEGF). Of note, this increase in VEGF in *Sh*-infected mice was consistent with findings from a second study [166]. Overall, this work demonstrated the wide-ranging role of macrophages in inducing a breadth of local and systemic immune responses during *Sh* infection in mice [170]. This is not unexpected since different macrophage subtypes are involved distinct types of immune responses. Notably, this study of macrophage depletion depleted all subtypes simultaneously and did not examine differential effects of various subtype depletions [173].

In another murine model of FGS, mice were injected with *Sh* eggs into the vaginal wall which led to an influx of different cell populations at different time points after *Sh* egg injections. In the first two weeks post-injection, vaginal mucosa exhibited a rise of T cells as well as RANTES. This was followed by a decline in T cells and RANTES, while macrophages became elevated starting 4 weeks after infection and persisting through week 8. Both T cells and macrophages in the vaginal wall were found to have increased expression of HIV entry molecules including CXCR4 and CCR5 from week 2 onward [172].

Nair et al used a murine knockout (KO) model to demonstrate a role of Metastasis-associated protein (MTA1) in *Sh* infection after direct percutaneous inoculation by immersion of the tail in water containing cercariae which typically leads to egg disposition in the liver [174]. MTA1 is a chromatin-bound protein that plays a role in mediating host inflammatory responses to components of virus and bacterial infectious products [175]. Mta1 KO mice had lower *Sh* infectious burden with fewer worms and eggs in the liver compared to wild-type mice and lower circulating CD4 mRNA, TNF-α and eosinophils. Mta1 KO mice also had a distinct cytokine profile in the liver, with an aberrant Th1/ Th2 cytokine profile marked by elevations in both Th1 and Th2 cytokines (IFN-γ, IL-2, IL-4, IL-5, IL-10) [174]. According to these findings, MTA1 appears to play a significant role in the egg mediated inflammatory response to murine *Sh* infection in the liver.

Acute infection with *Sh* cercariae was found to cause local immune response within hours of cercarial skin penetration, as evident in organ cultures of circumcised foreskin of healthy men infected with *Sh* cercariae by direct injection [176]. The analyzed cytokine expression in the foreskin from 1 to 8 hours after infection demonstrated an increase of IL-1a and IL-10 mRNA levels 4 and 8hrs after *Sh* infection versus controls, while macro gene array analysis showed decreased IL-6, IL-8, IL-15 and IL-18, elevated IL-1β, IL-2, IL-10 and TNF-α, and no change in IL-1 α in infected foreskin tissue.

**Overall synthesis of experimental immune studies in Sh infection.** Together, these experimental models demonstrate that there is a complex and dynamic immune response to

*Sh* infection, consisting of an initial mixed Th1 and Th2 response followed by a dominant Th2 response in the local tissue and systemically [168,170]. These animal findings largely cohere with human data and provide greater insight into the immune response. These studies extend insights into the response over time after initial infection and allow for an enhanced understanding of the biological mechanisms through *in vivo* experiments, such as the involvement of Mta1 and macrophages in the immunopathology of *Sh*, and through heightened control in comparison to human studies. However, it is important to highlight that there is still no animal model of *Sh* infection that completely resembles human *Sh* infection. To mimic human disease in which worms reside in the genitourinary tract and eggs are deposited in the genitourinary system, eggs must be directly injected into the genitourinary tissue since infection with cercariae only leads to egg disposition in liver and intestines in animals. Despite the importance of animal studies, it is highly likely that the presence of adult worms in humans with egg secretion and proteolytic passage of the eggs through tissue to reach their destination in the bladder cause a far more complex immune response that so far cannot be replicated in animal models.

## S. haematobium infection and host gene expression

In the next section, we turn to described effects of *Sh* infection on gene and protein expression. A total of 26 studies are included in this section. Twelve studies investigated the local tissue host gene and protein expression changes in *Sh*-associated bladder cancer and three studies investigated noncancerous *Sh* infection. Four studies focused on changes in systemic gene expression in individuals with *Sh* infection and seven studies focused on *Sh* experimental techniques in cell cultures and animal models.

## S. haematobium infection and host mucosal gene and protein expression in S. haematobium associated bladder cancer

In total, 12 studies have compared gene expression in people with bladder cancer with and without *Sh* infection (Table 11).

Bladder carcinoma typically involves alterations in several genes including the tumor suppressor genes *p53*, *p63*, *retinoblastoma* (*Rb*), and *p16*, as well as oncogenes *c-myc* and *epithelial growth factor* (*EGFR*), which are all associated with aggressive and invasive bladder carcinomas. Antiapoptotic protein *BCL-2* [177] and the proliferative protein ki-67 [178] have also been linked to aggressive bladder carcinoma.

Three studies have quantified *p53* expression in bladder carcinomas. Two studies reported higher expression of *p53* in individuals with *Sh*-associated bladder cancer compared to *Sh*-unassociated bladder cancer, in bladder specimens [179] and urine [180], suggesting tumor aggressiveness [177,181,182]. In contrast, *p53* was not more overexpressed in *Sh*-associated bladder cancer specimens than in *Sh*-unassociated bladder cancer specimens in patients from Egypt [183].

In addition to increased *p53* expression, Abdulamir et al reported more frequent expression of *BCL-2*, *c-myc*, and *Rb* in bladder biopsy specimens in *Sh*-associated bladder cancer in Jordan and Syria [179]. Further, *Sh*-associated cancer exhibited higher grade and increased invasiveness compared to *Sh*-unassociated bladder cancer [179]. Bernardo et al found, in addition to increased *p53* expression, increased estrogen receptor (ER) α expression in Angolese *Sh*-associated bladder cancer specimens, which was absent in *Sh*-unassociated bladder cancer specimens from Angolese and Portuguese patients without *Sh* exposure. Furthermore, ERα expression was positively correlated with numbers of *Sh* eggs found in bladder cancer specimens [180]. This increased ERα expression in *Sh*-associated bladder cancer appears to be unique to the association with *Sh* infection.

**Table 11. *S. haematobium* infection and mucosal gene and protein expression in *S. haematobium* associated bladder cancer.**

| Study | Country | Sample size | Sex (female in %) | Age in years | Compartment | Differences in individuals with bladder cancer and *S. haematobium* infection compared to individuals with bladder cancer without *S. haematobium* infection | Overall conclusion |
|---|---|---|---|---|---|---|---|
| Abdulamir et al 2009 (179) | Jordan, Syria | 148 | F (14%) | 38–72 | Bladder | In bladder cancer:<br>↑ *p53*, *Bcl-2*, *c*-myc, and *Rb*<br>= *p16*, *Ki-67*, and *EGFR*<br><br>In chronic cystitis (no bladder cancer):<br>= *p53*, *Bcl-2*, *c-myc*, *Rb*, *p16*, *Ki-67*, and *EGFR* | *Sh*-associated bladder cancer had a distinct genetic profile compared to *Sh*-negative bladder cancer with elevated *p53* expression, suggesting tumor aggressiveness.<br><br>There was no difference in gene expression in patients with chronic cystitis with or without *Sh* infection. |
| Bernardo et al 2016 (196) | Portugal, Angola | 9 | F (67%) | 18–82 | Urine | Proteins associated with:<br>↑ immune response and inflammation<br>↑ negative regulation of endopeptidase activity<br><br>*Sh* infection (with and without cancer):<br>↑ oxidative stress<br>↑ immune defense systems responsible for microbicide activity | *Sh* infection with or without cancer led to increased oxidative stress and immune system activation in the GU system.<br><br>*Sh*-associated cancer led to increased immune and inflammatory response. |
| Bernardo et al 2020 (180) | Portugal, Angola | 142 | F (30%) | 54–75 | Bladder | ↑ ERα<br>↑ *p53*<br>↑ proliferation<br><br>↓ ERβ<br><br>*In comparison to Sh negative bladder cancer and Sh positive non-malignant bladder lesions* | Increased *p53* expression in *Sh*-associated bladder cancer suggests aggressive tumor behavior.<br><br>ERα expression was increased in *Sh*-associated bladder cancer. |
| El-Salahy et al 2002 (163) | Egypt | 70 | Not documented | Not documented | Bladder | = cytokeratin 19<br>= cytokeratin-20 | There was no difference in cytokeratin-19 and -20 expression, which have been associated with bladder cancer. |
| Hassan et al 2013 (193) | Sudan | 194 | F (39%) | 50–84 | Bladder | ↑ COX-2<br>↑ iNOS | *Sh* infection, including cancer and cystitis, were both associated with increased COX-2 and iNOS. |
| Elmansy et al 2012 (187) | Egypt | 75 | F (16%) | 20–80 | Bladder | ↑ Fas and FasL<br><br>= cytokeratin-19 and -20 | *Sh*-associated bladder cancer had increased Fas and FasL expression. This may lead to apoptosis of infiltrating anti-tumor lymphocytes promoting cancer survival.<br><br>There was no difference in expression of cytokeratin-19 and -20, which have been associated with bladder cancer. |
| Khayoon et al 2021 (192) | Iraq | 50 | F (22%) | 21–70 | Bladder | ↑ COX-2 | Inflammatory protein COX-2 was increased *Sh*-associated bladder cancer. |
| Mursi et al 2013 (185) | Egypt | 70 | Not documented | Mean 45–60 | Bladder | ↑ *p63* expression | *Sh* infection was associated with increased expression of *p63* in bladder tissue of chronic cystitis and cancer. |
| Pycha et al 1993 (183) | Egypt | 220 | F (23%) | 28–83 | Bladder | = *p53* | *P53* was expressed similarly among *Sh*-associated and non-associated bladder cancer. |

(*Continued*)

**Table 11.** (Continued)

| Study | Country | Sample size | Sex (female in %) | Age in years | Compartment | Differences in individuals with bladder cancer and *S. haematobium* infection compared to individuals with bladder cancer without *S. haematobium* infection | Overall conclusion |
|---|---|---|---|---|---|---|---|
| Salim et al 2008 (194) | Egypt | 36 | F (78%) | 39–72 | Bladder | ↑ iNOS<br>↑ 8-OHdG, OGG1, APE1, and ssDNA<br><br>= eNOS<br><br>↑ eNOS in *Sh* SCC compared to *Sh* UC | *Sh*-associated cancers had increased oxidative stress and DNA damage. |
| Shams et al 2013 (184) | Egypt | 125 | F (20%) | 49–68 | Bladder | ↑ *c-KIT* | *C-KIT* expression was increased *Sh*-associated bladder cancer. |
| Sheweita et al 2001 (189) | Egypt | 41 | F (5%) | 42–70 | Bladder | ↑ carcinogen-metabolizing enzymes (Cyp450, Cyp b3, Aryl hydrocarbon hydroxylase, DMN-N-demethylase I activity, 12, Ethoxyresorufin-O-deethylase, Pentoxyresorufin-O-depentylase)<br>↑ level of free radicals<br>↑ GST<br>↑ thiobarbituric acid-reactive substances<br><br>↓ GR and GSH | Bladder cancer associated with *Sh* infection exhibited distinctive characteristics when compared *Sh* negative cancer with multiple changes associated with a malignant environment. |

*Sh* = *S. haematobium*, SCC = squamous cell carcinoma, UC = urothelial carcinoma, ER = estrogen receptor, COX-2 = Cyclooxygenase-2, GST = glutathione S-transferase, GR = glutathione reductase, GSH = Glutathione, 8-OHdG = 8-hydroxy-2-deoxyguanosine, eNOS = endothelial NOS, OGG1 = 8-oxoguanine-DNA-glycosylase, APE1 = apurinic/apyrimidinic endonuclease, ssDNA = single strand DNA, CK = Cytokeratin

Bladder tissue obtained from radical cystectomies of patients with squamous cell carcinoma in Egypt showed higher expression of the proto-oncogene *c-KIT* in carcinomas associated with *Sh*, compared to those not associated with *Sh* [184]. Furthermore, in the same study, *c-KIT* expression was also associated with increased tumor size and distant metastasis, arguing that *Sh*-associated squamous cell carcinoma is more aggressive than *Sh*-unassociated squamous cell carcinoma [184]. Another Egyptian study found increased *p63* in bladder specimens from people with *Sh* infection with either chronic cystitis or bladder cancer, suggesting that *p63* may be linked more generally to *Sh*-associated tissue disease rather than specifically to cancer [185]. The above four studies highlight that there are distinct differences in the genetic alterations occurring in *Sh*-related and unrelated bladder cancer [179,180,184,185].

The death receptor Fas (Apo1/CD95) and FasL system is recognized as a major pathway for the induction of apoptosis *in vivo*, and its blockade plays a critical role in carcinogenesis and progression in several malignancies [186]. Expression of both Fas and FasL was higher in *Sh*-associated bladder cancer samples than in non-*Sh*-associated bladder cancer [187]. A potential mechanism is that the acquisition of the functional FasL may induce apoptosis of anti-tumor T lymphocytes during tumor progression. Neither this study nor another found a change in cytokeratin-19 or -20 expression, which have also been associated with bladder cancer, in *Sh*-associated versus non-associated bladder cancers [163,187].

The enzymes glutathione S-transferases (GST) and glutathione reductase (GR) are a group of inducible enzymes that support detoxification of many different xenobiotics in mammals. Decrease in GST activity and depletion of glutathione (GSH) levels might potentiate the

harmful effects of environmental toxins and carcinogens. Consequently, inducers of GSTs are generally considered to be protective compounds against malignancy [188]. A study conducted in Egypt showed that bladder biopsies of individuals with *Sh*-associated bladder cancer had higher levels of GST as well as decrease in GR and GSH levels. All of these changes would contribute to poorer ability to attenuate environmental toxins and carcinogens, thereby increasing multiple carcinogen-metabolizing enzymes and free radicals and enhancing carcinogenic potential [189].

Cyclooxygenase-2 (COX-2) is an important pro-inflammatory mediator that is present only at low levels during normal conditions but is robustly induced after an insult such as infection or injury [190]. COX-2 expression, together with inducible nitric oxide synthetase (iNOS) which is associated with oxidative stress [191], was higher in bladder biopsies of *Sh*-associated versus non-*Sh*-associated bladder cancer patients and in *Sh*-associated versus non-*Sh*-associated benign cystitis [192]. A second study similarly reported higher iNOS in *Sh*-associated bladder cancer tissue [193,194]. Further, this second study found that levels of 8-hydroxy-2-deoxyguanosine (8-OHdG), a marker for DNA damage [195], were increased in tissue of *Sh*-associated bladder cancer [194]. Biopsies from *Sh*-associated cancers also had higher expression of DNA-repair genes, including *8-OHdG* and *apurinic/apyrimidinic endonuclease*, and higher formation levels of single strand DNA [194]. Together, these two studies suggest a strong correlation between *Sh* infection and increased levels of oxidative stress accompanied by continuous DNA damage, necessitating increased DNA repair.

Proteome profiling of urine samples of 9 participants with either *Sh*-associated bladder inflammation or cancer, or *Sh*-unassociated bladder cancer, showed an increase in proteins involved in immunity and inflammation as well as negative regulation of endopeptidase activity in *Sh*-associated bladder cancer. Samples from participants with *Sh* infection, whether inflammation or cancer, had higher urinary proteins associated with oxidative stress and immune defense systems responsible for microbicide activity, in comparison to *Sh*-uninfected participants with bladder cancer [196].

**Overall synthesis of gene expression studies in *Sh* infection and bladder cancer.**
Together, the above studies highlight the unique genetic changes associated with immune activation, chronic inflammation, and fibrosis in *Sh* infection. These changes in gene expression may both predispose to carcinogenesis, and then be further modulated by carcinogenesis, resulting in high-grade and aggressive bladder cancer.

## *S. haematobium* infection and mucosal gene expression of the host

A total of three studies have compared local mucosal gene expression in people with and without *Sh* infection, without cancer (Table 12).

110 genes were differently expressed in cervical cytobrush samples from Tanzanian women with or without *Sh* infection. In this study, the transcript count showed that 7 out of 8 genes had lower expression in women with *Sh* infection. The top altered canonical pathway was "inhibition of matrix metalloproteinases", with increased expression of metalloproteinases (MMP) MMP-2 and MMP-16 as well as decreased expression of tissue inhibitor MMP-3 (TIMP-3), followed by: "colorectal cancer metastasis signaling", "glioma invasiveness signaling", and "bladder cancer signaling." Together, these results suggest that *Sh* infection is associated with compromised integrity of the genital mucosal barrier in women with *Sh* infection, and with genetic changes that could promote malignancy [149].

In a study of genetic changes in liver biopsies of patients with *Sh* and/or *S. mansoni* infection, biopsies with fibrosis were compared to normal liver biopsies without schistosomiasis. People with *Sh* mono-infection or *Sh*/ *S. mansoni* co-infection had elevated aflatoxin B1

**Table 12.** **S. haematobium** infection and mucosal gene and protein expression.

| Study | Country | Sample size | Sex (female in %) | Age in years | Compartment | Differences in individuals with *S. haematobium* infection compared to individuals without *S. haematobium* infection | Overall conclusion |
|---|---|---|---|---|---|---|---|
| Dupnik et al 2019 (149) | Tanzania | 57 | F (100%) | 20–35 | Cervical | ↑ *MMP-2*, *MMP-16*, *ABP4* (encoding fatty acid binding protein 4, *TGM3* (transglutaminase 3;), *MB* (myoglobin), *SERPINB11* (serpin family B member 11), *TEPP* (testis, prostate and placenta expressed), *SLC26A2* (solute carrier family 26 member 2), *SERPINA3* (serpin family A member 3; fold change), and *AKR1B15* (aldo-keto reductase family 1 member B15)<br><br>↓ *MMP-3* (TIMP-3) and *SLC26A2* (solute carrier family) | *Sh* infection led to gene alterations in the cervix of infected women which are associated with loss of epithelial barrier integrity and malignancy. |
| Habib et al 2006 (198) | Egypt | 26 | F (27%) | 20–46 | Liver | ↑ *AFB1* and *p53* | *Sh* infection led to pro-oncogenic genetic changes in the liver. |
| Zhong et al 2013 (199) | Ghana | 57 | F (47%) | 18–80 | Urine | Individuals with *Sh* infection with bladder damage on US:<br>*RASSF1A* methylation and *TIMP-3* methylation | *Sh* infection led to methylation and hence to a decreased expression of tumor suppressor genes followed by tissue damage. |

*Sh* = *S. haematobium*, MMP = metalloproteinase, TIMP = tissue inhibitor metalloproteinase, AFB1 = aflatoxin B1, US = ultrasound

adducts, known hepatic carcinogens [197], and those with *Sh* mono-infection had more *p53* mutations than *S. mansoni* mono-infected or healthy controls [198]. Together, this suggests that entrapped *Sh* eggs can lead to pro-carcinogenic changes in the liver in addition to the bladder, while *S. mansoni* eggs appear to be less carcinogenic.

Methylated *RASSF1A* and *TIMP3* genes, both tumor suppressor genes that have been documented to be elevated in bladder cancer and damage [199], were found to be increased in the urine of individuals with *Sh* infection who had ultrasound-confirmed bladder damage compared to individuals without *Sh* infection from the same endemic region in Ghana. DNA methylation of two tumor suppressor genes in urine samples that correlated with *Sh* induced bladder damage suggested that these might be early signs of bladder transformation that occurred in advance of bladder cancer. If this is correct, these methylated genes could serve as predictive biomarkers in individuals with *Sh* infection for genitourinary damage and cancer risk.

## S. haematobium infection and systemic gene expression in blood

Four studies compared systemic gene expression changes by examining whole blood transcriptional profiles in people with and without *Sh* infection (Table 13).

In a study conducted in Ghana, mRNA expression of *TLR2* was downregulated in whole blood of children with *Sh* infection compared to those without infection [200]. *TLR2* recognizes a wide range of pathogenic molecules and is able to induce a Th1 or Th2 immune response [201] and suppressor of cytokine signaling 3 (SOCS-3), which leads to increased Th2 differentiation and altered IL-2 and IFN signaling. The children infected with helminths, including *Sh*, had lower frequencies of reactions to dust hermits tested by a mite allergen skin prick test [200] in line with prior studies describing decreased allergic reactions in *S. mansoni* [202] and helminth infected individuals [203]. Further, the lower allergic reactions in those with *Sh* were accompanied by lower expression of genes associated with the innate immune system, particularly *TLR2* and *SOCS-3*, which trigger allergic inflammation and IFN response.

**Table 13. *S. haematobium* infection and systemic gene expression in blood.**

| Study | Country | Sample size | Sex (female in %) | Age in years | Differences in individuals with *S. haematobium* infection compared to individuals without *S. haematobium* infection | Effect of praziquantel treatment | Overall conclusion |
|---|---|---|---|---|---|---|---|
| Dupnik et al 2018 (205) | Tanzania | 33 | F (60%) | 21–33 | Different gene expression:<br>• 383 genes overall (including genes regarding development, cell death and survival, cell signaling, and immunologic disease pathways)<br>• 7 genes associated with *p53* signaling (including *BCL-2*, *caspase 6*, *histone deacetylase 9*)<br>• 2142 genes differently expressed in women<br>• 270 genes differently expressed in men<br>histone transcription (including *histone deacetylase 9*, *ctr9*, *hsf2*, *elp2*, *wdr2*)<br><br>↑ Notch signaling genes (including mindbomb ubiquitin ligase, neuralized E3 ubiquitin protein ligase 1) | N/A | *Sh* infection led to unique genetic changes including cell signaling and survival and immune responses. Women with *Sh* infection had more changes in gene expression in comparison to men with *Sh* infection. |
| *Hartgers et al 2008 (200) | Ghana | 120 | F (46%) | 5–14 | ↓ TLR2 mRNA and SOSC-3 mRNA | N/A | Individuals with *Sh* infection had decreased expression of TLR2 and SOCS-3 mRNA, identifying a molecular link between *Sh* infection and altered immune response. |
| *Labuda et al 2020 (58) | Gabon | 17 | Male and female (% not documented) | School-children | ↑ 140 genes<br>↑ cell adhesion genes<br>↑ oxidative phosphorylation<br>↑ citrate/TCA cycle<br><br>↓ 180 genes<br>↓ NK cells, DCs and monocytes | ↑ 34 genes<br>↑ immune response<br>↑ T cell anergy<br>↑ EGR2 and EGR3<br>↑ cell cycle<br><br>oxidative phosphorylation<br>↓ 158 genes<br>↓ citrate/TCA cycle<br><br>*7 months after treatment* | *Sh* infection led to molecular changes, including increased genetic pathways important for Th2 immunity, cell adhesion and decrease in pertinent immune cell populations.<br><br>PZQ treatment led to multiple changes in gene expression, but did not completely resolve these. |
| Shariati et al 2001 (207) | Malia, Nigeria | 33 | F (3%) | 19–37 | ↑ *VEGF* | N/A | *Sh* infection led to increased *VEGF* expression which is associated with bladder cancer. |

*Sh* = *S. haematobium*, TLR = toll like receptor, SOCS = suppressor of cytokine signaling, EGR = early growth response genes
*Studies included participants with additional parasitic infections.

The impaired IFN response in *Sh* infection may connect *Sh* infections not only to a reduced risk of atopy but also to an increased risk of viral infections. This research underscores the unique genetic modulation in *Sh* infection compared to other helminth infections, emphasizing the importance of understanding the unique characteristics of *Sh* infection apart from other helminths.

In the previously mentioned study by Labuda et al [58], analysis of the whole blood transcriptome identified 320 differentially expressed genes between 8 Gabonese schoolchildren with *Sh* infection and 9 without *Sh* infection, while after PZQ treatment only 4 genes remained differently expressed between the two groups. Gene ontology analysis showed increase in processes associated with cell adhesion in those with *Sh* infection, whereas PZQ treatment resulted in downregulation of immune responses and cell cycle genes [58]. Furthermore, children with *Sh* infection had reduced gene signatures for NK cells, DCs, and monocytes, while T cell activation, miotic cell cycles, and pathways involved in oxidative phosphorylation and the citrate/TCA cycle were upregulated. All these oncologic changes suggest induction and maintenance of type 2 immunity [204]. The differences in NK cell signatures and mitotic cell cycle were reversed, and immune activation was overall reduced post-PZQ therapy [58]. Chemotherapy with PZQ also led to upregulation of oxidative phosphorylation and downregulated citrate/TCA cycle, and transcriptional factors for the induction of T cell anergy, including early growth response genes 2 and 3 [58]. Overall, this study indicated multiple changes occurring on a molecular level during *Sh* infection, consistent with increased cell adhesion and type 2 immunity, which was only partly reversed by PZQ treatment. A limitation of this study is that schoolchildren in both groups also had different prevalence of co-infections with other parasites, which included *Plasmodium falciparum*, *Ascaris lumbricoides*, *Trichuris trichiura*, and hookworm. These co-infections may have influenced the transcriptome profiling of the individuals.

Transcriptional profiles of PBMCs in a Tanzanian cohort showed 383 differentially expressed genes between those with and those without *Sh* infection, of which 270 genes correlated with CAA values. Interestingly, 29 genes were differentially expressed in men with *Sh* infection, and 2,142 genes women with *Sh* infection compared to men and women without infection, respectively [205]. Ingenuity pathway analysis identified altered networks associated with development, cell death and survival, cell signaling, and immunologic disease pathways. Genes which were differentially expressed were associated with *p53* signaling, including *BCL-2*, as well as with histone transcription. Furthermore, expression of multiple genes involved in Notch signaling were increased. This study emphasizes not only changes in genetic expression occurring during *Sh* infection, but furthermore the importance of host factors, such as sex, that impact this host response to *Sh*.

Individuals with *Sh* infection (n = 15) who had lived in Spain for 6 months after immigrating from West Africa had increased *VEGF* expression on whole blood ELISA testing, which has been associated with bladder cancer, compared to those without *Sh* infection (n = 18) [206]. While the study's size limits interpretability, the reported rise in *VEGF* aligns with findings by Fu et al in *Sh*-infected mice [166], despite the two prior studies not reporting changes in *VEGF* expression.

**Overall synthesis of systemic gene expression studies in *Sh* infection.** Together, these gene expression data triangulate and substantiate immune changes observed in studies described above. Both groups of studies indicate changes in type 2 immune phenotype, cell signaling, and survival. Besides the known role of sex differences in the immune response, this literature demonstrates that they also affect genetic modulation in response to *Sh* infection [205]. This underscores the importance of evaluating host response to infection, including host gene expression, by sex.

## Experimental models to study *S. haematobium* infection and host gene, protein expression, and cell cycle dynamics

Seven studies assessed changes in gene expression associated with *Sh* utilizing experimental models (Table 14).

**Table 14. Experimental models to study effects of *S. haematobium* infection on gene, protein expression and cell cycle dynamics.**

| Study | Study design | Compartment | Differences in *S. haematobium* infected compared to uninfected | Overall conclusion |
|---|---|---|---|---|
| Botelho et al 2009 (215) | Chinese hamster ovary cells co-cultured with *Sh* total antigen | Ovarian cell line | ↑ S phase and G2/ M phase<br><br>↓ G02/ G1 phase<br><br>↑ Bcl-2 gene (anti-apoptotic)<br><br>↓ p27 gene (tumor suppressor) | *Sh* total antigen led to increased proliferation and decreased apoptosis through changes in cell cycle and gene expression. |
| Botelho et al 2013 (210) | Mice were administered *Sh* total antigen versus saline as control transurethrally into the bladder. | Bladder | ↑ Dysplastic bladder lesions<br>↑ *KRAS* (20% of mice with dysplastic lesions)<br><br>↑ Inflammatory and lymphocyte infiltrate<br>↑ Monocytes | Most bladders of mice exposed to *Sh* total antigen developed inflammatory infiltrates. Half developed dysplastic lesions and 20% of these with KRAS mutation |
| Chala et al 2017 (209) | Mice were either injected with *Sh* eggs into the bladder wall or received chemical carcinogen (BBN) administered through drinking water, or received both. | Bladder | Mice injected with *Sh* eggs alone:<br>↑ *Ki-67* in lymphocytes (week 4 and 12)<br>↑ p53 in urothelial cells (week 4)<br><br>Mice injected with *Sh* eggs and BBN:<br>↑ *Ki-67* in urothelial cells (week 20)<br>↑ *Vimentin* urothelial cells (week 20)<br>↑ *p53* urothelial cells (week 20)<br>↓ *E-Cadherin* in urothelial cells (week 20)<br><br>Mice with BBN alone:<br>↑ *Vimentin* urothelial cells (week 20) | Mice injected with *Sh* eggs had increased Ki-67 in lymphocytes and elevated *p53* in urothelial cells.<br><br>Mice injected with eggs and carcinogen had the most evident changes concerning for possible malignant transformation. |
| Mbanefo et al 2020 (214) | WT or IL-4R KO Mice were either injected with *Sh* eggs or a sham injection into the bladder wall. | Bladder | WT mice with *Sh* eggs:<br>↑ S phase and G2/ M phase<br><br>IL-4R KO mice with *Sh* eggs:<br>↓ S phase | *Sh* eggs induced cell cycle changes in the bladder in an IL-4 dependent fashion. |

(*Continued*)

**Table 14.**  (Continued)

| Study | Study design | Compartment | Differences in *S. haematobium* infected compared to uninfected | Overall conclusion |
|---|---|---|---|---|
| Osakunor et al 2022 (212) | Mice were injected with *Sh* eggs or control extract directly into the bladder wall. | Bladder | Proteins associated with: ↑ carcinogenesis (STEAP4, Pla2g7, Dab2, Fkbp14, Stab1, Loxl2, Prpf3, Ltbp3, Nob1, Raly, Gpx8) immune and inflammatory responses (STEAP4, Serpina3n, Dab2, C8a, Pecam1, Prpf3, Dnajc7) protein translation or turnover (Rps9, Rpl5, Rpl26, Rps13, Rps3a, Rpl23) oxidative stress responses (Pla2g7) ↑ epithelial barrier integrity (Ptpn2) glucose metabolism (Tbc1d4) ↓ carcinogenesis (Cryl1, Usp11, Gnao1, Gnao1, Grhpr, Ppp1r14a, Hopx) tumor suppression (Usp11, Gas1, Cbr1) ↓ cell survival (Bcl-2) ↓ structural integrity and cell adhesion (Itgb3, Itga1, Itga7, Armc1, Usp11, Dmd, Mapre3) | Multiple changes in proteome expression were identified in mice with *Sh* eggs associated with carcinogenesis, inflammation, and loss of epithelial barrier integrity. |

(*Continued*)

**Table 14.** (Continued)

| Study | Study design | Compartment | Differences in *S. haematobium* infected compared to uninfected | Overall conclusion |
|---|---|---|---|---|
| Ray et al 2012 (213) | Mice were injected with *Sh* eggs or control vehicle directly into the bladder wall. | Bladder | By week 1:<br>↑ 279 genes ↓ 22 genes<br><br>By week 3:<br>↑ 1001 genes ↓ 570 genes<br><br>By week 5:<br>↑ 794 genes ↓ 308 genes<br><br>Genes related to:<br>↑ Inflammation (IL-4, IL-4-induced 1, IL-1β, IL-6, IFN- inducible proteins (IFI30 and IFI47), TGF-β, IL-13 receptor alpha 2, IL-10 receptor alpha, and cytokine inducible SH2- containing protein)<br>↑ Markers of macrophages (*macrophage-expressed gene 1* (general macrophage marker) Markers of AAM (*arginase, Ym1 (CHI3L3), and mannose receptor C type 1*)<br>↑ Chemokines (*CCL4 (MIP-1β), CCL5 (RANTES), CCL11 (eotaxin) and CXCL1 (KC)*)<br>↑ Carcinogenesis (growth factor-, oncogene-, and mammary tumor-related genes)<br><br>↓ claudins and junctional adhesion molecules<br><br>Changes over time:<br>↑ cytokine-cytokine receptor pathways, peaked at week 1 and 3<br>↑ B cell receptor signaling pathway, peaked at week 5<br><br>↓ claudins and junctional adhesion molecules, week 3 | *Sh* egg disposition in the bladder led to many genetic changes in the bladder including differential transcription of immune response-, fibrosis-, cancer-, and epithelial barrier-related genes. |

(*Continued*)

**Table 14.** (Continued)

| Study | Study design | Compartment | Differences in *S. haematobium* infected compared to uninfected | Overall conclusion |
|---|---|---|---|---|
| Sheweita et al 2003 (208) | Organs of hamsters with and without transdermal *Sh* cercarial infection were assessed at different timepoints after *Sh* infection over the course of 10 weeks. | Bladder | ↑ GST (week 2, 4, 6) <br> ↓ GST (week 8, 10) <br> ↑ GSH (week 2, 6) <br> ↓ GSH (week 4, 8, 10) <br> ↑ GR (week 8) <br> ↓ GR (week 10) <br> ↑ Free radicals (week 2, 4, 6, 8, 10) | Decreased GST activity and increased free radicals were seen in most organs, but most pronounced in bladder which could indicate an early mechanism of *Sh* induced organ damage. |
| | | Kidney | ↑ GST (week 6, 10) <br> ↓ GSH (week 4, 8, 10) <br> ↑ GR (week 2, 4, 6, 8) <br> Free radicals (week 2, 4, 6, 8, 10) | |
| | | Liver | = GST <br> ↓ GSH (week 6, 8, 10) <br> ↑ GR (week 2, 4) <br> ↓ GR (week 6) | |
| | | Spleen | ↑ GSH (week 4, 6, 10) <br> ↓ GST (week 2, 6, 10) <br> ↑ GR (week 4, 8) <br> ↓ Free radicals (week 6, 8, 10) <br> ↑ Free radicals (week 2, 4) | |
| | | Lung | ↑ GST (week 8, 10) <br> ↓ GSH (week 8, 10) <br> ↑ GSH (week 2, 4, 6) <br> ↑ GR (week 2, 4, 6, 8, 10) | |

*Sh* = *S. haematobium*, GST = glutathione S-transferase, GR = glutathione reductase, GSH = Glutathione, BBN = N-butyl n-(4hydroxybutyl) nitrosamine,

AAM = alternatively activated macrophage, WT = wild-type, IL-4R = IL-4 receptor alpha, KO = knockout

A study that assessed activity of GST and GR, and levels of GSH, in multiple organs of transdermally infected hamsters with *Sh* cercariae with successive sampling for up to 10 weeks post infection showed substantial variation depending on the specific organ as well as the time point after infection [207]. There was no change in uninfected hamsters used as a control at each time point. Activity of GST and GR, as well as GSH levels, increased in the studied organs (liver, bladder, spleen, and lung) at some time point post-infection followed by a partial decrease in the liver and bladder. Only the bladder was found to have a decrease in all three (GST and GR activity, and GSH levels) at the last two stages post-infection. Free radicals remained elevated throughout only in the bladder and kidneys. This demonstrated reduction, particularly of GST activity in the bladder during the later stages of *Sh* infection, and the associated high levels of free radicals, could provide new insights on *Sh* induced damage to the bladder and other organs [207].

A mouse study investigated effects of injection of *Sh* eggs directly into the bladder, versus administration of carcinogen N-butyl n-(4-hydroxylbutyl) nitrosamine through drinking water, which is a known carcinogen, versus eggs plus carcinogen. Mice injected with eggs alone had higher expression of activation marker *Ki-67* in lymphocytes and *p53* after 4 weeks, and *Ki-67* remained elevated at 12 weeks. The eggs plus carcinogen group was the only group with increased expression of *Ki-67* in urothelial cells, which was found at week 20. Additionally, this group had elevated *p53* expression and vimentin at week 20, while *E-cadherin* was decreased [208]. This suggests that *Sh* eggs in bladder tissue led to upregulation of lymphocyte

proliferation, yet led to genetic changes worrisome for urothelial transformation only when combined with a known carcinogen.

In a study by Botelho and colleagues [209], half of the mice that were exposed to *Sh* total antigen, infused directly into the bladder through the urethra, developed histologically dysplastic bladder lesions. Almost all *Sh*-exposed mice were found to have inflammatory infiltrates and elevated lymphocytes in the bladder tissue, compared to control mice. Furthermore, 20% of the dysplastic bladder lesions were found to have a *KRAS* gene mutation [209], which is commonly found in bladder cancers [210]. This implies that *Sh* presence may upregulate *KRAS*, increasing the risk of developing bladder cancer. It is unclear if this is specific to *Sh* or a general response in bladder epithelium to carcinogens, including *Sh* antigens. A control group with a non-*Sh* carcinogen-induced bladder transformation would have added insight into differences between *Sh*-mediated and other carcinogenic mediated bladder lesions.

Proteome profiling of bladders of mice that underwent local injection of *Sh* eggs into the bladder walls showed differential expression of proteins important for immune and inflammatory responses, increased protein translation or turnover, oxidative stress responses, reduced epithelial barrier integrity, and carcinogenic pathways 7 days after eggs were administered [211]. Overall, the bladders of mice with *Sh* egg injection were found to have 45 differentially expressed proteins (25 upregulated and 20 downregulated) compared to mice that were injected with uninfected hamster liver and intestine tissue extract into their bladder walls as controls.

Microarray-based comparisons of *Sh* egg versus control vehicle-injected bladders demonstrated differential gene transcription changes over time with 301 changes at 1 week, a peak of 1571 changes at 3 weeks, and 1012 genetic changes by 5 weeks post egg injection [212]. The presence of *Sh* eggs in the bladder of mice prompted an increase of genes related to inflammation, particularly cytokine-cytokine receptor pathways dominated by Th2 cytokines, and macrophage function-associated gene transcription. Simultaneously, extracellular matrix remodeling-related gene transcription was differentially modulated over time, and pathways analysis pointed to differential transcription of multiple genes implicated in carcinogenesis [212]. *Sh* eggs also dampened transcription of epithelial barrier genes, such as tight junctions and uroplakins, which may cause compromises in the urothelial barrier [212]. These findings indicate that the presence of *Sh* eggs in bladder tissue not only provoke a strong local, largely Th2 immune response, but also enhance genes associated with extracellular matrix remodeling suggesting fibrosis development, compromised epithelial barrier integrity, and carcinogenesis.

*Sh* egg bladder injection in wild-type mice increased mitotic cells, with a higher proportion of urothelial cells in S phase and G2/M phase [213], which was also seen after *Sh* total antigen administration to a Chinese hamster ovary cell line [214]. This pro-miotic effect was diminished in IL-4 receptor-deficient mice [213], indicating an IL-4-dependent mechanism. In addition to alterations in cell cycle phases, *Sh* total antigen administration to the Chinese hamster ovary cell line resulted in decreased apoptosis with increases in anti-apoptotic gene Bcl-2, and in tumor suppressor gene p27. These studies suggest that *Sh* may lead to bladder cell cycle skewing in an IL-4-dependent fashion.

**Overall synthesis of experimental gene expression studies in *Sh* infection.** Taken together, these experimental studies conducted in animal models and cell lines demonstrate that *Sh* infection mimicked by the presence of *Sh* eggs or antigens can lead to changes in enzyme and protein expression, and cell cycle changes, which are linked to an environment favorable for inflammation, compromised epithelial barrier, and carcinogenesis. These studies emphasize not only the role of *Sh* infection in the development of bladder cancer, but also provide insight into the pathophysiologic changes that underlie *Sh*-linked bladder cancer, as well as possibly facilitate entry of additional pathogens through a compromised epithelial barrier.

## *S. haematobium* infection and the host microbiome

A total of six studies compared microbiome changes in gut, urine, and cervicovaginal microbiota during *Sh* infection (Table 15).

In four of these, *Firmicutes*, *Proteobacteria*, and *Bacteroidetes* were the most abundant bacteria phyla in both gut and urine microbiota in individuals with and without *Sh* infection [26,215–217].

Three studies focused on differences in the gut microbiome through analysis of stool samples of individuals with and without *Sh* infection. *Sh* adult worms reside in the vasculature of the genitourinary system, and eggs migrate preferentially into the genitourinary system to be excreted in the urine. Though not common, *Sh* eggs can be occasionally identified in stool of infected individuals as well [218,219]. *Sh* could affect the fecal microbiome through the presence of eggs in the gastrointestinal tract or through a systemic effect. Ajibola et al [26] showed higher *Proteobacteria* and lower *Firmicutes* proportions in the gut microbiota by analysis of stool samples of adolescents with and without *Sh* infection. Furthermore, beta diversity, the measure of similarity or dissimilarity between microbiome groups, differed between people with and without *Sh* infection, while alpha diversity, the diversity within a microbiome, remained the same [26]. This study also emphasized how changes in the gut microbiome of adolescents with *Sh* infection led to a dysbiotic state. Kay et al [216] found a difference in the abundance of 5 *Prevotella* operational taxonomic units (OTUs) in the gut microbiome in fecal samples of children with *Sh* infection ages 6 months to 13 years in comparison to children without infection [216]. Osakunor et al [217] studied children with *Sh* infection ages 1–5 and reported that *Sh* infection status and intensity affects bacteria and fungi genera in stool [217]. These studies consistently demonstrate associations of *Sh* infection with altered microbial composition in different compartments outside of the genitourinary tract in infected adolescents and children. Changes in fecal microbiota in individuals with *Sh* infection may have multiple downstream consequences for the host, including altering the host's mucosal immune system, which could impact local response to intruding pathogens as well as cancer. Furthermore, these changes in the fecal microbiota appeared to be refractory to praziquantel treatment [216,217,220].

Studying the urine microbiota in *Sh* infection is pertinent since the genitourinary system is the site where *Sh* eggs mainly become entrapped, subsequently inducing local inflammation, fibrosis, and granuloma formation [212,221]. Changes in the urine microbiome have been postulated to contribute to the pathogenesis of the development of bladder cancer in individuals with *Sh* infection [211,212]. Despite this, Adebayo and colleagues are the only study thus far to report urine microbial composition in individuals with *Sh* infection. Their study found higher *Firmicutes* and lower *Proteobacteria* proportions in the urine of individuals with *Sh* infection, compared to individuals without *Sh* infection [215]. The presence of distinct microbial species and varying levels of *Firmicutes* and *Proteobacteria* in individuals with *Sh* infection could induce inflammation, with less protection for the infected host. Altered *Firmicutes* and *Proteobacteria* levels have been connected to gastrointestinal dysbiosis, which can alter bile acid composition, leading to gastrointestinal tract inflammation and increased gut permeability [222–225].

Two studies investigated the cervicovaginal microbiome using swabs collected from cervicovaginal mucosa, the site of *Sh* egg deposition. Changes in the cervicovaginal microbiome, particularly increased diversity and loss of *Lactobacilli*, are associated with increased susceptibility to some infections, including HIV [226,227]. Women with high intensity *Sh* infections, defined as a CAA value of above 3000 pg/ ml, were found to have higher alpha diversity compared to women with lower intensity infections or women without infection [228]. After PZQ

**Table 15.  Effects of *S. haematobium* infection on the host microbiome.**

| Study | Country | Sample size | Sex (female in %) | Age in years | Microbiome compartment | Differences in individuals with *S. haematobium* infection compared to individuals without *S. haematobium* infection | Effect of praziquantel treatment | Overall conclusion |
|---|---|---|---|---|---|---|---|---|
| Adebayo et al 2017 (216) | Nigeria | 70 | F (49%) | 15–65 | Urine | ↑ OTUs in advanced *Sh* cases<br>↑ *Firmicutes* proportion<br><br>↓ Diversity index in advanced *Sh* cases<br>↓ *Proteobacteria* proportion | N/A | There were unique differences in the urinary microbiome between individuals with *Sh* infection alone. |
| Ajibola et al 2019 (26) | Nigeria | 49 | F (16%) | 11–15 | Gut (fecal) | ↑ Abundance of *Proteobacteria, Moraxellaceae, Veillonellaceae, Pasteurellaceae,* and *Desulfovibrionaceae*<br><br>↓ Abundance of *Firmicutes* and *Clostridiales*<br><br>Differences in beta but not alpha diversity | N/A | Adolescents with *Sh* infection had a shift in the gut microbiome consistent with dysbiosis and gut inflammation. |
| Bullington et al 2021 (229) | Tanzania | 43 | F (100%) | Median = 34 | Cervicovaginal | ↑ Alpha diversity with infection intensity<br>↑ Beta diversity at follow-up<br><br>↓ Trend of abundance of *Lactobacillus* with infection intensity | ↑ Alpha diversity if *Sh* positive vs. *Sh* negative (trend)<br><br>*3 months after treatment* | *Sh* infection with high intensity was associated with more diverse cervicovaginal bacterial communities than women without *Sh* infection. |
| Kay et al 2015 (128) | Zimbabwe | 139 | F (47%) | 6 mo– 13 yo | Gut (fecal) | ↑ OTU species diversity<br>↑ Abundance in 5 *Prevotella* OTU<br>↑ Nonsignificant in 22 OTU | No effect on prior seen microbiome changes in *Sh* infected<br><br>*3 months after treatment* | Children with *Sh* infection had significant gut microbiome changes which were refractory to PZQ treatment. |
| Osakunor et al 2020 (218) | Zimbabwe | 113 | F (50%) | 1–5 | Gut (fecal) | ↑ *Pseudomonas, Stenotrophomonas, Derxia, Thalassospira Aspergillus,* Tricholoma, and *Periglandula*<br><br>↓ *Azospirillum* | No difference was seen in individuals with a history of PZQ treatment | Children with *Sh* infection had distinct differences in their fecal microbiome compared to children without *Sh* infection. |
| Sturt et al 2021 (230) | Zambia | 188 | F (100%) | 18–31 | Cervicovaginal | No differences in key cervicovaginal species (*Lactobacillus crispatus, Lactobacillus iners, Atopobium vaginae,* and *Candida albicans*)<br><br>Women with FGS and moderate/ high genital *Sh* burden (≥2 *Sh* qPCR-positive genital specimens):<br>↑ *Trichomonas vaginalis* and *Garderella vaginalis* | N/A | Women with *Sh* infection with moderate/ high genital *Sh* burden were more commonly infected *T. vaginalis* and *G. vaginalis*. |

*Sh* = *S. haematobium*, PZQ = Praziquantel, OTU = operational taxonomic units, FGS = female genital schistosomiasis

treatment, an increase in alpha diversity in participants that continued to test positive for *Sh* was found [228]. In contrast, Sturt et al [229] did not find differences in presence or concentration of key cervicovaginal species in women with and without FGS. Interestingly, individuals in whom two or more genital samples had detectable *Sh* DNA had a significantly higher risk of

*T. vaginalis* [229]. *T. vaginalis* has been shown to influence HIV-1 acquisition and transmission [230,231], with some studies showing a decline in HIV-1 shedding rates after metronidazole treatment [232–234]. These two studies underscore how alterations in the cervicovaginal microbiome due to *Sh* infection could be a mechanism that increases host susceptibility to HIV and other infectious diseases.

**Overall synthesis of microbiome studies in *Sh* infection.** Together, the above studies demonstrate that *Sh* infection can lead to changes of the microbiome both locally in the genitourinary tracts where egg disposition mainly occurs, as well as in other mucosal compartments such as the gastrointestinal tract. These changes in microbial populations may have far reaching consequences on both immune responses and acquisition and transmission of diseases. To date, no study has examined the relationship between microbiota and mucosal immune populations in *Sh* infection, which would be an important area for future investigation.

## Discussion

In this systematic and comprehensive review, a total of 89 of 94 studies (95%) demonstrated associations of *Sh* infection with considerable changes in the host immune response, gene expression, and/or microbiome composition (Fig 3). These changes underlie several major public health impacts of *Sh* infection. This chronic parasitic infection, which lasts for an average of 3–5 years when untreated [235], shifts the host's immune phenotype to a predominate type 2 immune phenotype, provokes increases in regulatory cell populations, such as Tregs and Bregs, and inflammatory cytokines, and compromises the epithelial barrier and microbiome diversity. Each of these major changes could contribute to the impaired defenses against

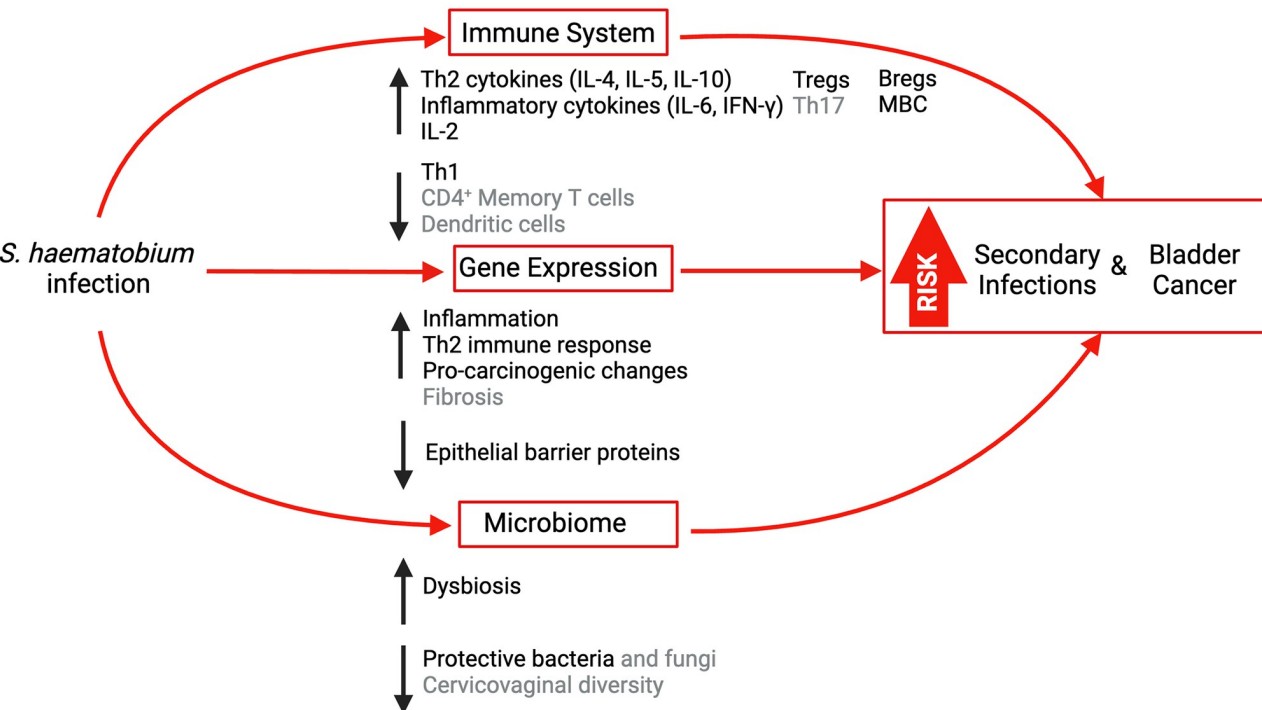

**Fig 3. Changes in the host's immune response, gene and protein expression and microbiome composition during *S. haematobium* infection.** Overview of the main changes in immune, gene and microbiome with *S. haematobium* infection reported in the 94 studies included in this review. Tregs = T regulatory cells; MBC = Memory B cells; Bregs = B regulatory cells; Th = T helper cell. Black font indicates two or more studies have demonstrated this relationship. Light grey indicates a finding that has only been reported by a single study. Image created with BioRender.com.

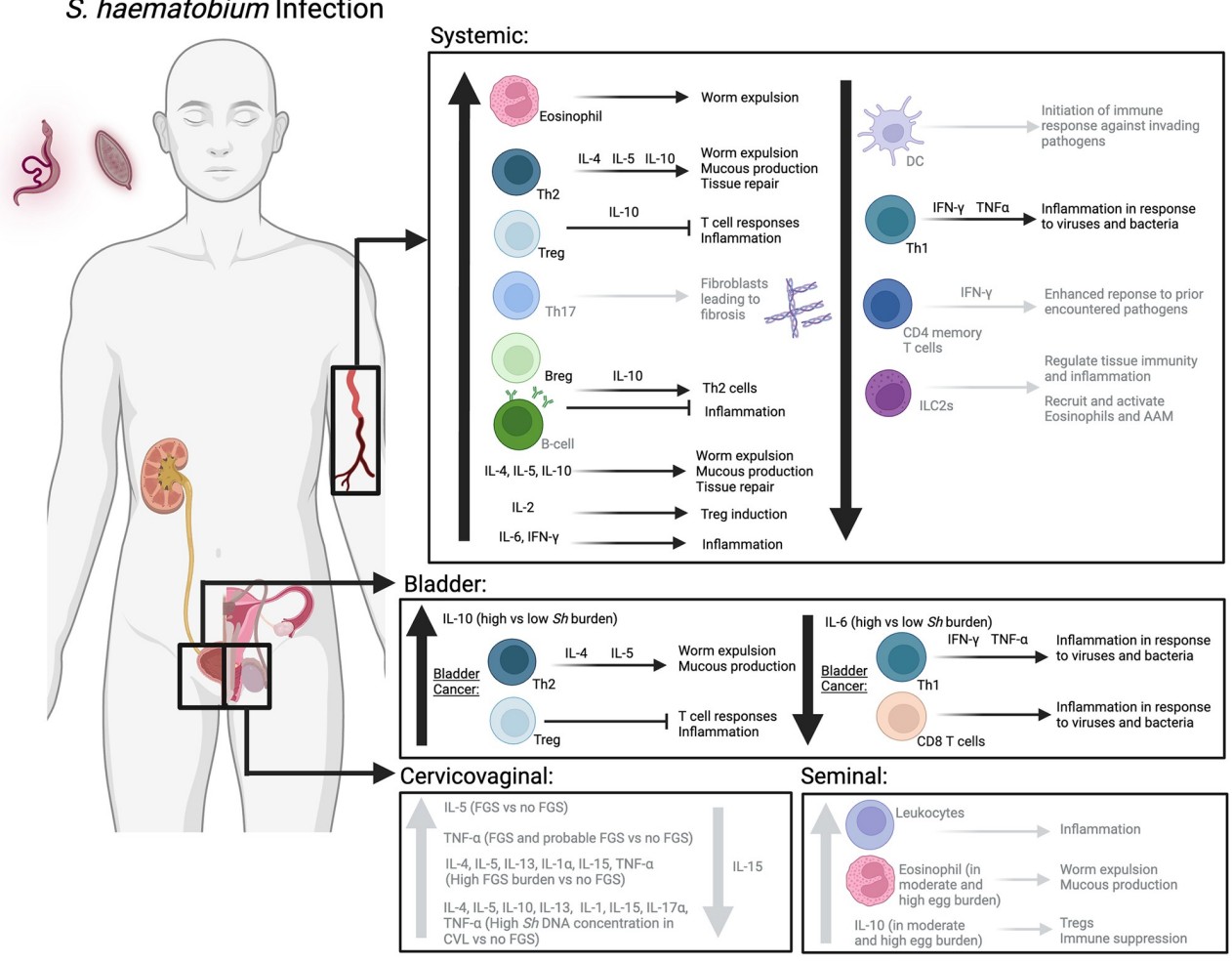

**Fig 4. Changes in the host's immune system seen during *S. haematobium* infection.** Overview of immune changes associated with *S. haematobium* infection. Th = T helper cells; Tregs = T regulatory cells; Bregs = B regulatory cells; DCs = dendritic cells; ILCs = innate lymphoid cells; CVL = cervicovaginal lavage; FGS = female genital schistosomiasis. Black font indicates two or more studies have demonstrated this relationship. Light grey indicates a finding that has only been reported by a single study. Image created with BioRender.com.

viruses, the increased risk of carcinogenesis, or both that are seen in *Sh* infection. Of further importance, the concordance of many studies that treatment with praziquantel only partially resolves these immune, gene expression, and microbial changes emphasizes the urgent need to prevent and aggressively manage this infection in the first place.

The studies assessed reported multiple immune alterations in *Sh* infection, which are summarized in Fig 4. In addition to the type 2 and regulatory phenotype mentioned above, numerous studies provided evidence that *Sh* infection hinders an adequate cellular immune response. This includes functional impairment of various immune cell subsets crucial for host defense against invading pathogens, such as DCs, B cell subsets, CD4$^+$ memory, and EM T cells. Interestingly, the role of CD8$^+$ in infection remains to be determined. One study showed no difference, while another indicated an increase in this pertinent T cell population systemically [103,123]. A number of studies also associated *Sh* infection with increased pro-inflammatory cytokines, such as IFN-γ, IL-6, and MIP1-α, in both blood and the genital tract [236].

When applied to the reproductive tract, where the tissue burden of *Sh* eggs is frequently high, these immune changes may increase the risk of acquisition and transmission of viral infections, particularly in women. It is well established that female genital tract inflammation caused by other infections creates a conducive environment for HIV acquisition, both via recruitment of HIV target cells to mucosal surfaces and via damage to epithelial integrity. Only one study in this review assessed HIV target cells in the female genital tract, and found decreases in CD4$^+$ T cells expressing CCR5 in the genital tract following PZQ treatment [107]. Two others identified genital tract cytokine changes consistent with increased genital tract inflammation and/or decreased antiviral immunity [148,149]. It is also possible that IL-6, which one study reported to be elevated in the urinary bladder in *Sh* infection and which is associated with increased HIV viral shedding in the genital tract, could play a role [237].

A second putative mechanism for increased viral susceptibility in the genital tract could be *Sh*-induced alterations in mucosal gene expression and microbiota. The first line of host defense against pathogens is the epithelial barrier. Alterations in epithelial barrier proteins have been associated with impaired epithelial barrier function, allowing easier pathogen translocation [227,238]. In this review, many studies demonstrated lower expression of epithelial barrier proteins and higher expression of proteases, which can cause epithelial barrier destruction, in *Sh* infection. Interestingly, changes in the epithelial barrier can also impact the microbiome and vice-versa [239]. The reported microbial shifts associated with *Sh* infection, such as altered *Firmicutes* and *Proteobacteria* levels, may further inflammation. Furthermore, reported cervicovaginal findings of increased microbial diversity, abundance of taxa, and low *Lactobacillus* in *Sh* infection have all been associated with increased risk of HIV acquisition [226]. Together, these *Sh* infection gene expression and microbiome data corroborate and may potentiate the *Sh*-associated immune alterations, which collectively align with enhanced potential for increased viral susceptibility.

Of note, it has been reported that that schistosome infection might alleviate disease severity in some chronic viral infections such as HIV, HTLV-1, and respiratory viruses [29]. These observations underscore the importance of a thorough examination of immune effects of *Sh* infection in endemic settings where viral and *Sh*, as well as other helminth infections, intersect. It is possible that the mixed picture of type 2, regulatory, and pro-inflammatory immune responses in *Sh* infection may contribute to these sometimes seemingly paradoxical findings. Moreover, a strong Treg environment in schistosome infection has been linked to reduced hepatic damage from HBV [240], improved HTLV outcome [241] and decreased susceptibility to HIV-1 R5 virus [242]. Further research is essential to understand the reasons that underlie disparate viral presentations during *Sh* infection, to explore effects of *Sh* on other important viruses including HPV, and potentially to guide future immunoregulatory interventions that harness beneficial effects of *Sh* infection.

In *Sh* endemic countries, bladder cancer is commonly linked to past or current infections with *Sh* [14,158]. Across an array of human and animal studies of *Sh* infection, articles in this review report a Th2-skewed environment with concomitant downregulation of Th1 response, which may favor carcinogenesis. Furthermore, chronic inflammation is a well-known risk factor for the development of malignancy [243] and elements of a pro-inflammatory immune environment, in addition to type 2 findings, were identified in a substantial portion of studies included in this review. Other studies demonstrated distinctive genetic, epigenetic, protein, and cell cycle alterations in bladder cancer associated with *Sh* infection when compared to *Sh*-unassociated bladder cancer. These multiple alterations link *Sh* infection to an environment conducive to malignant growth. It is also possible that the noted shifts in the microbiome during *Sh* infection, particularly in urine, impact bladder carcinogenesis by affecting the immune response or by inducing changes

in gene expression in the urothelium. Given the emerging possibility of manipulating microbiota to prevent other cancers [244], this is another important area for future research.

The variation of reported findings among the microbiome studies highlights the challenges inherent in these types of analyses. Further adding to the complexity, microbiota can impact the immune system and gene expression [245,246] and the microbiome is highly diverse, with different body sites hosting distinct microbial communities that are impacted by multiple factors, beginning from birth. The microbiome is also dynamic, with variations not only throughout the individual's lifespan, but sometimes daily within the same individual [17,20,247]. Geographic location also impacts microbiome composition. The six microbiota-focused studies in this review investigated different compartments and spanned four different countries, with age groups ranging from 13-month-year-olds to 65-year-old adults. This could explain why the specific changes seen in one bodily compartment do not completely overlap with the findings of another study, despite investigating the same compartment. What is clear from this review is that *Sh* was associated with alterations in the microbiome in all three of these compartments. The scarcity of these studies highlights the need for additional microbiome analyses in *Sh* infection across geography, age, and assessing associations of microbiota with both baseline and provoked immune responses.

Our synthesis of articles suggests that anti-schistosome treatment results in the partial, but often incomplete, restoration of *Sh*-associated changes in immune phenotype, gene expression, and microbiota. The single study investigating genetic changes post- versus pre-PZQ treatment showed a reduction, but not complete normalization, of genetic alterations. The two studies investigating microbiome changes before versus three months after successful treatment did not observe any significant changes. It is possible that the microbiome may require longer time to recover after *Sh* infection than the three month follow up periods in these studies. Together, the study results discussed in this review argue that PZQ could be beneficial, at least in reverting some immune alterations associated with *Sh* infection back to a pre-infection state, but that PZQ may be less able to reverse the effects of *Sh* infection on gene expression and microbiota. For these reasons, it will be important to investigate whether resolution of *Sh* infection truly diminishes the risk of viral acquisition and the onset of bladder cancer.

This review has strengths and limitations. We included 94 studies that documented active schistosome infection by microscopy, antigen testing, PCR, serologies, or clinical exam. Serologies to diagnose *Sh* infection were included despite antibodies remaining positive after clearance of infection, and the possibility that some cases might not reflect active infection. Nevertheless, we included these studies which identified *Sh* infection based on positive serologies to ensure we did not underestimate effects of *Sh* infection, and in order to assess possible long-term effects of past *Sh* infections. The included studies assessed a broad range of human populations, sometimes leading to heterogenous results. Beyond age, sex, co-infections, and areas of residence with different *Sh* endemicity, a large array of other factors may influence the individual's immune response to *Sh* infection. These include environmental, sociodemographic, and economic aspects of *Sh* infection, which could not be ascertained. Furthermore, the effects of timing, duration, and infectious burden of schistosome infection likely affect the manifestation of schistosome-associated immune, gene and microbiome changes. Finally, publication bias could have led to the non-publication of studies that showed no impact of *Sh* on immune system, gene expression, and microbiome and thereby would not have been included.

Despite these limitations, our rigorous analysis provides, to our knowledge, the first comprehensive synthesis of immune, genetic, and microbial factors associated with *Sh* infection. Our synthesis can help to guide understanding of potential supplementary therapeutic approaches to reverse consequences of *Sh* infection. This body of data may also pave the way

for novel immunoregulatory, genetic, or microbiome interventions and guide global strategies related to prevention of infections and cancer.

## Methods

We conducted a systematic review to assess the effects of *Sh* infection has on the immune system, gene expression and microbiome. This study was reported in accordance with the Preferred Reporting Items for Systematic Reviews and Meta-Analyses (PRISMA) guidelines and was registered prospectively in PROSPERO (CRD42022372607).

### Search strategy

Comprehensive literature searches were developed and performed by a medical librarian [DW]. The initial search was performed December 22nd, 2022 via OVID MEDLINE ALL (December 22nd, 2022). The Cochrane Library (Cochrane Database of Systematic Reviews, Cochrane Central Register of Controlled Trials (CENTRAL), Cochrane Methodology Register, Technology Assessments (HTA)), LILACS (Latin American and Caribbean Health Sciences Literature), and OVID EMBASE (1974-March 28, 2019) were searched on December 22nd, 2022; Scopus (Elsevier) on December 22nd, 2022. Follow-up searches were performed on August 9th, 2023. Search terms included all subject headings and/or keywords associated with our research question, clustered as:

1. Parasite or proxy for parasitic infection (e.g. *Schistosoma haematobium*, Schistosomiasis, Bilharzia, Bilharziasis)

2. Immune-related headings (e.g. Immunity, Immunomodulation, Immune response, Mucosal immunity, Immune changes, Cellular immunity)

3. Microbiome related headings (Microbiome, Microbiota, Cervicovaginal microbiota, Genitourinary microbiota)

4. Gene expression-related headings (e.g. Gene expression, Mucosal gene expression, Transcriptomics)

Boolean operators 'OR' and 'AND' were used as appropriate. Grey literature and bibliographies of included articles were also searched; see full search strategy in the Supplementary Appendix (S1 Text). Publication date restrictions were not imposed. Only papers written in the English language were included. Databases were searched from inception. Authors were not contacted for further information. Studies were uploaded to the Covidence platform (Melbourne, Australia) for conduct of the screening and extraction phases. Included study types were randomized clinical trials, cohort, cross-sectional, and case-control studies. Both human, animal and experimental *ex vivo* studies were included. S1 Text summarizes our study's predefined search terms, keywords, and study types included. S2 Text summarizes the decision rules for data selection and extraction.

### Study selection and evaluation process

1. Study selection
   After excluding duplicates, two investigators independently screened titles and abstracts to identify studies eligible for inclusion using the above criteria, and a third investigator independently resolved discrepancies. Discrepant analyses of study quality were resolved by discussion and author consensus. Due to the heterogeneity of study designs, including human, animal and *ex vivo* studies, a meta-analysis was not possible. We included studies describing

the effects of *Sh* on the immune system, gene expression or microbiome. Pre-defined exclusion criteria included: not available in English, and study design of case study or case review, conference proceedings/meeting abstracts, dissertations/theses, newspaper/magazine articles and research in progress. We prespecified that studies must test for *Sh* infection by serologies, *Sh* egg detection by tissue or urine microscopy, or antigen testing. For the scope of this review, we prespecified that we would exclude studies describing humoral immune responses and solely focus on cellular immunity, cytokine and chemokine signaling and immune cell interactions. Full-text review followed the initial title and abstract screening phase, with data extracted into Microsoft Excel.

2. Data extraction
   Data extracted included study type; country of origin; sample size; immunology, gene, or microbiome outcomes; key findings; effects of anthelmintic therapy on outcomes, if applicable; and study limitations.

3. Quality assessment
   Quality assessment of data was performed using the Downs and Black checklist [27]. Studies were categorized as "poor," "fair," "good," or "excellent" based on a summative score derived from this checklist, and only those of "fair" quality or above were included in the final analysis.

## Software

The displayed graph was generated using GraphPad Prism software. Displayed figures were created with BioRender.

## Supporting information

**S1 Table. List of excluded studies during quality assessment using the Downs and Black Checklist.**
(DOCX)

**S2 Table. Extracted data of included studies.**
(XLSX)

**S1 PRISMA Checklist. For more information, visit: http://www.prismastatement.org/.**
(DOCX)

**S1 Text. Full search strategy.**
(DOCX)

**S2 Text. Decision Rules for Data Selection and Extraction Processes.**
(DOCX)

## Acknowledgments

We thank the librarian team at Weill Cornell Medicine for their assistance with searching and retrieving study records for this systematic review. Furthermore, we want to thank Drs. Mary Charleston and Carol Mancuso who lead the Master's program Clinical Epidemiology & Health Services Research, and Dr. Kyu Rhee, who leads to Burroughs Wellcome Physician Scientist Program at Weill Cornell, who all have been continued supporters and offered instrumental guidance for this review.

## Author Contributions

**Conceptualization:** Anna M. Mertelsmann, Drew Wright, Jennifer A. Downs.

**Data curation:** Anna M. Mertelsmann, Sheridan F. Bowers, Drew Wright, Jennifer A. Downs.

**Formal analysis:** Anna M. Mertelsmann, Sheridan F. Bowers, Jennifer A. Downs.

**Funding acquisition:** Anna M. Mertelsmann, Jennifer A. Downs.

**Investigation:** Anna M. Mertelsmann, Jennifer A. Downs.

**Methodology:** Anna M. Mertelsmann, Drew Wright, Jennifer A. Downs.

**Project administration:** Drew Wright.

**Resources:** Drew Wright.

**Software:** Drew Wright.

**Supervision:** Jennifer A. Downs.

**Validation:** Jane K. Maganga, Humphrey D. Mazigo, Lishomwa C. Ndhlovu, John M. Changalucha, Jennifer A. Downs.

**Visualization:** Anna M. Mertelsmann, Jennifer A. Downs.

**Writing – original draft:** Anna M. Mertelsmann, Sheridan F. Bowers, Jennifer A. Downs.

**Writing – review & editing:** Anna M. Mertelsmann, Sheridan F. Bowers, Drew Wright, Jane K. Maganga, Humphrey D. Mazigo, Lishomwa C. Ndhlovu, John M. Changalucha, Jennifer A. Downs.

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
