## [Decision Letter · Decision Letter 0]

7 Jun 2024

Dear Dr. Mertelsmann,

Thank you very much for submitting your manuscript "Effects of Schistosoma haematobium infection and treatment on the systemic and mucosal immune phenotype, gene expression and microbiome: A systematic review" for consideration at PLOS Neglected Tropical Diseases. As with all papers reviewed by the journal, your manuscript was reviewed by members of the editorial board and by several independent reviewers. In light of the reviews (below this email), we would like to invite the resubmission of a significantly-revised version that takes into account the reviewers' comments. 

We cannot make any decision about publication until we have seen the revised manuscript and your response to the reviewers' comments. Your revised manuscript is also likely to be sent to reviewers for further evaluation.

Sincerely,

Hamed Kalani

Academic Editor

jong-Yil Chai

Section Editor

Reviewer #1: 

Major comments: 

1. The main objective of this manuscript is to systematically review the effects of S. haematobium infection on the immune system, gene expression, and microbiome of humans and experimental models. However, the current structure of the "results" section makes the audience easily lost due to numerous subsections and further subsections. Additionally, these sections are sometimes interconnected, making it challenging for readers to grasp the content. For example, sections starting from line 375, 703, 823, 942, 1043, 1132, 1206, and 1270 provide brief summaries of previous literature reviews. Presenting them as separate sections disrupts the reading flow. I suggest the authors either restructure the content or include a figure/table to illustrate the organization of results section.

2. The authors should provide a more detailed explanation of the antigens studied in the literature they referenced. In Box 2, schistosome antigens include "soluble egg antigen," "adult worm antigen," "soluble adult worm antigen preparation," and "schistosome phosphatidylserine”. The authors frequently refer to "schistosome antigens" throughout the text (e.g., lines 267, 269, etc.). Are they exactly the same molecules? Additionally, the authors also talked about egg-specific, cercaria-specific, and adult worm-specific antigens (lines 320-331). What are the differences between stage-specific antigens and schistosome antigens? Clear and complete information about the antigens is crucial for readers to compare and evaluate the results. 

3. For the section "S. haematobium infection and host gene expression," it is unclear whether the observed changes were at the RNA or protein level. The authors should clarify this, for example, by using italics to indicate genes.

4. The authors should explicitly list in the tables and/or mention in the main text the studies that involve patients with not only schistosome infections but also multiple infections. This is crucial, as multiple infections can significantly confound our understanding of schistosome infections.

Minor comments: 

1. The current way of referencing studies in tables makes it difficult to locate the original study in the reference section. Please consider using the numerical indication that matches the reference style. 

2. Line 205-207, not sure what “significantly different between the groups studied” refers to. Please revise. 

3. Line 285, I suggest the authors use “purified protein derivate” rather than PDD. The term explanation is only in Box 2 and this term does not appear many times in the main text. 

4. Line 346-346, talking about HIV here feels out of place. Please consider revising. 

5. Line 439, what does "encounter" mean here? Does it refer to the first responsive immune cells or the first cell to come into contact with the parasite?

6. Line 703, the section title should be revised to match the rest section title style. Same for line 1270. 

7. Line 754-747, this sentence is very long and hard to understand. 

8. Line 876-879, I suggest moving this sentence to the beginning of the section. 

9. Line 997, please use the full term of SCC as it first appears in the text. 

10. Line 987, “….Sh-associated..” to ““….Sh-associated..” Same for line 1076. 

11. Ensure consistent styling for "ex vivo" and "in vivo. These words are italic in lines 36, 661, and 948, but not in lien 868 and 1007. 

12. Line 1146-1151, what are “the specific organ” and “most organs”?

Reviewer #2: This manuscript was written by researchers who have contributed substantially to the topic at hand. Through a systematic review, the authors attempted to compile the effects of S haematobium on the host immune, genetic and microbial profiles. A large search from public databases and unspecified sources yielded 3796 individual studies that could be narrowed down to 94 studies after manual curating for relevance and quality of the report.

Relevance was set for studies dealing with core themes for which the authors made a great deal in assessing and compiling all relevant literature and providing comprehensive figures summarizing the overall observations. The impressive amount of references included (229), and the depth of the analytical reports made for each sub-theme (host immune response, gene expression, microbiome) are quite compelling and convincingly uplift the quality of the report making this review a much welcome addition to the literature for the community.

 Some points are nevertheless to be considered:

1-Line 160: The subdivision of cytokine profiles of CD4+ T cells into simply Th1 and Th2 is quite anachronic for our present days (https://doi.org/10.3389/fimmu.2023.1284178), the T cell polarization landscape has greatly evolved and should be at least incorporated here. In fact, studies on Sh and the host immune responses do report on such updated T cell types (https://doi.org/10.1093/infdis/jis524; doi: 10.1093/infdis/jis654) and are counterintuitively reported in this review.

2-Several references mentioned in Box 1 do not appear adequate to support the points made eg. IL-2 can have pro- or anti-inflammatory properties depending on the immune milieu (38,39) where ref 38 = Lyke K, Dabo A, Sangare L, Arama C, Daou M, Diarra I, et al. Effects of concomitant Schistosoma haematobium infection on the serum cytokine levels elicited by acute Plasmodium falciparum malaria infection in Malian children. Infect Immun. 2006;74(10):5718–24; ref 39 = Ateba-Ngoa U, Adegnika A, Zinsou J, Kassa Kassa R, Smits H, Massinga-Loembe M, et al. Cytokine and chemokine profile of the innate and adaptive immune response of Schistosoma haematobium and Plasmodium falciparum single and co-infected school-1584 aged children from an endemic area of Lambaréné, Gabon. Malar J. 2015;14:94. Re-check throughout the manuscript and correct.

Reviewer #3: The most pressing issue for me would be reviewing the number of PubMed publications found as this seems concerning. Otherwise it is a very well written, well thought out and well executed review.

1- There are some instances of misspelling of Schistosoma e.g line 22

2-In the abstract 'Results' section, some more details on the findings would help the reader gain more insight.

3-Line 73: Misspelling of 'predominately'

4-To me it is surprising that only 2 studies were identified in PubMed, searching the same query on PubMed yields 1,320 results. Some clarification would be necessary here. There is also a typo in the supplementary materials, where the PubMed search string is missing at the start.

5-Table 1 - misspelling of 'inflammatory' as 'inflamatory'

6-It might be worth including a column for antigens used for stimulation to improve the readability of the table. This can also be more specific in the text e.g. line 369 and 1167, references to schistosome antigen should be more specific as to which antigen was used.

7-Similarly description of the sample types used for measurements would aid in the understanding of the studies, this is provided in some instances but is not comprehensive.

8-Figure 2 legend - 'Sh-infected depending' - there is a word missing here

9-Line 823 - should be 'local tissue' not 'systemic'

10-Line 877 - perhaps a comment should be added on mouse models using abdominal exposure to cercariae, do these not recapitulate natural Sh infection? The included number of experimental animal studies is quite low, perhaps it would be beneficial to comment on the reasons for this

11- Line 1126 - rephrase sentence e.g. 'were tested using whole blood ELISA, finding...'

12-Discussion Line 1297 - Here it would be good to comment on the lack of differences observed for CD8 T cells, this is an interesting point that is not highlighted sufficiently.

13- Line 1445 - should be antigen detection or testing etc.

14-Though I would be cautious that the final figures (3 and 4) do not overstate the findings of one paper or another. I would suggest that the findings included in the summary figures be those that only have some kind of consensus from the reviewed papers (e.g. 2 or more), if this is not already the case.
---

## [Editor Report · Decision Letter 1]

9 Jul 2024

Dear Dr. Mertelsmann,

Thank you very much for submitting your manuscript "Effects of Schistosoma haematobium infection and treatment on the systemic and mucosal immune phenotype, gene expression and microbiome: A systematic review" for consideration at PLOS Neglected Tropical Diseases. As with all papers reviewed by the journal, your manuscript was reviewed by members of the editorial board and by several independent reviewers. In light of the reviews (below this email), we would like to invite the resubmission of a significantly-revised version that takes into account the reviewers' comments. 

Reviewers' comments:

The article has been revised to address the requested corrections; however, there are remaining ambiguities and flaws that require further resolution:

1- In Figure 1, a total of 6,973 studies are obtained from various databases/registers, while the sum of studies obtained from Embase, PubMed, MEDLINE, and Unspecified databases is 8,275?

2- In the case of Figures, the legend should be given below the Figures (Figure 1)

3- In Figure 1, write the reason for excluding 293 articles in Figure 1. For example, 20 articles due to X; 50 articles due to Y (no need to write the verb)

4- Scopus has been examined as a search database, however, it is not mentioned in Figure 1. Conversely, it is imperative to clearly identify unspecified databases in order to avoid any ambiguities.

5- The "selection study" section should be divided into three separate sections: 1) Study selection 2)Quality assessment 3) Data extraction

6- It is imperative to carefully read the entire manuscript for any potential writing issues. For example, in the "supplementary materials" section, the word "Pbmed" ought to be replaced with "PubMed."

We cannot make any decision about publication until we have seen the revised manuscript and your response to the reviewers' comments. Your revised manuscript is also likely to be sent to reviewers for further evaluation.

Sincerely,

Hamed Kalani

Academic Editor

Jong-Yil Chai

Section Editor
---

## [Editor Report · Decision Letter 2]

20 Jul 2024

Dear Dr. Mertelsmann,

Thank you very much for submitting your manuscript "Effects of Schistosoma haematobium infection and treatment on the systemic and mucosal immune phenotype, gene expression and microbiome: A systematic review" for consideration at PLOS Neglected Tropical Diseases. As with all papers reviewed by the journal, your manuscript was reviewed by members of the editorial board and by several independent reviewers. The reviewers appreciated the attention to an important topic. Based on the reviews, we are likely to accept this manuscript for publication, providing that you modify the manuscript according to the review recommendations. 

The requested modifications by the authors have been made. The text appears to be well-written, however, unfortunately, there are still typographical errors in the text that need to be carefully read and corrected before any decision is made regarding this manuscript.

For examples:

Line 22, comma (,) should be removed. 

Line 39, comma (,) should be removed. 

Line 40, ....with (a) predominant type 2 ....is correct.

Line 57, ........infections(,) which (cause) urogenital.........is correct.

Line 1495, ......using Graph(P)ad Prism (software)....is correct.

The text before and after line 1495 has not been examined, and it is necessary for the authors to carefully do it.

Sincerely,

Hamed Kalani

Academic Editor

Jong-Yil Chai

Section Editor

Figure Files:

Data Requirements:

Reproducibility:

References

---

## [Editor Report · Decision Letter 3]

29 Jul 2024

Dear Dr. Mertelsmann,

Thank you very much for submitting your manuscript "Effects of Schistosoma haematobium infection and treatment on the systemic and mucosal immune phenotype, gene expression and microbiome: A systematic review" for consideration at PLOS Neglected Tropical Diseases. As with all papers reviewed by the journal, your manuscript was reviewed by members of the editorial board and by several independent reviewers. The reviewers appreciated the attention to an important topic. Based on the reviews, we are likely to accept this manuscript for publication, providing that you modify the manuscript according to the review recommendations. 

The scientific aspects of this manuscript have been properly revised, however, there are still grammatical and writing issues that need to be addressed before a decision is made regarding this manuscript (i.e. Figure 2: the title of the figures should be written below it (i.e. legend). Additionally, incorrect words such as "Graphpad" should be avoided in Figure 2. etc). The entire manuscript must be thoroughly examined and not limited to the mentioned points only.

Sincerely,

Hamed Kalani

Academic Editor

Jong-Yil Chai

Section Editor

Figure Files:

Data Requirements:

Reproducibility:

References

---

## [Editor Report · Decision Letter 4]

13 Aug 2024

Dear Dr. Mertelsmann,

We are pleased to inform you that your manuscript 'Effects of Schistosoma haematobium infection and treatment on the systemic and mucosal immune phenotype, gene expression and microbiome: A systematic review' has been provisionally accepted for publication in PLOS Neglected Tropical Diseases.

Best regards,

Hamed Kalani

Academic Editor

Jong-Yil Chai

Section Editor

The authors have addressed the errors present in the manuscript, and it appears that there are no remaining issues related to either the scientific content or the writing quality. In my assessment, the manuscript is suitable for publication.

---

## [Editor Report · Acceptance letter]

27 Aug 2024

Dear Dr. Mertelsmann,

We are delighted to inform you that your manuscript, "Effects of Schistosoma haematobium infection and treatment on the systemic and mucosal immune phenotype, gene expression and microbiome: A systematic review," has been formally accepted for publication in PLOS Neglected Tropical Diseases.

Best regards,

Shaden Kamhawi

co-Editor-in-Chief

Paul Brindley

co-Editor-in-Chief
